# Metabolic Pathway Analysis in the Presence of Biological Constraints

**Philippe Dague** 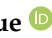

Université Paris-Saclay, CNRS, ENS Paris-Saclay, Inria, Laboratoire Méthodes Formelles, 91190 Gif-sur-Yvette, France; philippe.dague@universite-paris-saclay.fr

**Abstract:** Metabolic pathway analysis is a key method to study a metabolism in its steady state, and the concept of elementary fluxes (EFs) plays a major role in the analysis of a network in terms of non-decomposable pathways. The supports of the EFs contain in particular those of the elementary flux modes (EFMs), which are the support-minimal pathways, and EFs coincide with EFMs when the only flux constraints are given by the irreversibility of certain reactions. Practical use of both EFMs and EFs has been hampered by the combinatorial explosion of their number in large, genome-scale systems. The EFs give the possible pathways in a steady state but the real pathways are limited by biological constraints, such as thermodynamic or, more generally, kinetic constraints and regulatory constraints from the genetic network. We provide results on the mathematical structure and geometrical characterization of the solution space in the presence of such biological constraints (which is no longer a convex polyhedral cone or a convex polyhedron) and revisit the concept of EFMs and EFs in this framework. We show that most of the results depend only on very general properties of compatibility of constraints with vector signs: either sign-invariance, satisfied by regulatory constraints, or sign-monotonicity (a stronger property), satisfied by thermodynamic and kinetic constraints. We show in particular that the solution space for sign-monotone constraints is a union of particular faces of the original polyhedral cone or polyhedron and that EFs still coincide with EFMs and are just those of the original EFs that satisfy the constraint, and we show how to integrate their computation efficiently in the double description method, the most widely used method in the tools dedicated to EFs computation. We show that, for sign-invariant constraints, the situation is more complex: the solution space is a disjoint union of particular semi-open faces (i.e., without some of their own faces of lesser dimension) of the original polyhedral cone or polyhedron and, if EFs are still those of the original EFs that satisfy the constraint, their computation cannot be incrementally integrated into the double description method, and the result is not true for EFMs, that are in general strictly more numerous than those of the original EFs that satisfy the constraint.

**Keywords:** metabolic pathway analysis; steady state; flux cone; flux polyhedron; elementary flux modes; elementary flux vectors and points; extreme vectors and points; double description method; thermodynamic constraints; kinetic constraints; regulatory constraints; sign-monotonicity; sign-invariance; support-minimality; conformal non-decomposability; conformal support-wise non-strict-decomposability

## 1. Introduction

### 1.1. Metabolic Networks

In order to ensure this paper is self-contained and has no prerequisite to be read, we summarize in this introduction the state-of-the-art related to the subject and fix the notations adopted throughout the paper. The results quoted being known, are thus given without proof and the reader is invited to refer to the works in [1–7], in addition to the references in the text, for the proofs or surveys.

A metabolic network is made up of a set of $r$ biochemical enzymatic reactions. Each reaction consumes certain metabolites (called substrates of the reaction) and produces other metabolites (called products of the reaction). Each metabolite is assigned a coefficient in the

reaction, its stoichiometric coefficient (counted negatively for substrates and positively for products). We distinguish internal from external metabolites w.r.t. the system under study (e.g., a bacterium, an eukaryotic cell), the reactions involving both internal and external metabolites being transfer reactions. If $m$ is the number of internal metabolites ($r > m$), the network is thus given by its stoichiometric matrix $\mathbf{S} \in \mathbb{R}^{m \times r}$, where coefficient $\mathbf{S}_{ji}$ is the stoichiometric coefficient of internal metabolite $j$ in reaction $i$ (positive if $j$ is a product of reaction $i$ and negative if it is a substrate). A state of the network at a given time $t$ is given by the net rates (or fluxes) in each of its reactions at $t$, i.e., by a flux vector (or rate vector, or flux distribution) $\mathbf{v}(t) \in \mathbb{R}^r$. Denoting by $\underline{\mathbf{M}}(t) \in \mathbb{R}_+^{*m}$ the vector of the concentrations of internal metabolites at $t$, the time evolution of the network is thus given by

$$\frac{d\underline{\mathbf{M}}(t)}{dt} = \mathbf{Sv}. \tag{1}$$

### 1.2. Steady-State Behavior and Flux Subspace

We are interested in the steady-state behavior of the network. The steady-state assumption means that the concentrations of internal metabolites remain constant along time (no accumulation or reduction of internal metabolites inside the system, an approximation which is valid for short time periods, e.g., a few minutes) and leads thus to the fundamental equation

$$\mathbf{Sv} = \mathbf{0}. \tag{2}$$

With only this assumption, the solution space $Sol_{\mathbf{S}}$, i.e., the space of all admissible flux vectors $\mathbf{v}$, is thus the linear subspace $FS$ of $\mathbb{R}^r$ given by the kernel, or nullspace [8], of $\mathbf{S}$:

$$Sol_{\mathbf{S}} \overset{\triangle}{=} FS \overset{\triangle}{=} \{\mathbf{v} \in \mathbb{R}^r \mid \mathbf{Sv} = \mathbf{0}\}. \tag{3}$$

The dimension of the flux subspace is given by $dim(FS) = r - rank(\mathbf{S}) \geq r - m$. Often, possibly after a preprocessing to eliminate its linearly dependent rows, $\mathbf{S}$ is assumed to be of full rank and then $dim(FS) = r - m$.

The support of a vector $\mathbf{x} \in \mathbb{R}^r$ is defined by

$$supp(\mathbf{x}) \overset{\triangle}{=} \{i \mid \mathbf{x}_i \neq 0\}. \tag{4}$$

The support of a flux vector $\mathbf{v}$ has thus an important biological signification as it represents the reactions involved in the subnetwork (that we shall call pathway) defined by $\mathbf{v}$ (i.e., those through which the flux given by $\mathbf{v}$ is not null).

### 1.3. Irreversible Reactions, Flux Cones and Polyhedral Cones

In addition to the homogeneous equality constraints provided by the steady-state assumption, the flux vectors, in general, must also satisfy homogeneous inequality constraints corresponding to reactions known as irreversible, whose set will be noted **Irr**:

$$\mathbf{v}_i \geq 0 \text{ for } i \in \mathbf{Irr}. \tag{5}$$

This means that fluxes in irreversible reactions are constrained to be non-negative (in reversible reactions, fluxes may be either positive, or negative or null and the direction fixed as positive is arbitrary, the role between substrates and products being able to switch). If $r_I = |\mathbf{Irr}|$, with $0 \leq r_I \leq r$, is the number of irreversible reactions, the solution space $Sol_{\mathbf{S},\mathbf{Irr}}$ is the intersection of the linear subspace $FS$ with $r_I$ non-negative half-spaces, it is thus a particular case of a convex polyhedral cone, called s-cone (subspace cone or special cone) or flux cone, noted $FC$:

$$Sol_{\mathbf{S},\mathbf{Irr}} \overset{\triangle}{=} FC \overset{\triangle}{=} \{\mathbf{v} \in \mathbb{R}^r \mid \mathbf{Sv} = \mathbf{0}, \mathbf{v}_i \geq 0 \text{ for } i \in \mathbf{Irr}\}. \tag{6}$$

A (general) convex polyhedral cone is defined implicitly (or by intension) by finitely many homogeneous linear inequalities:

$$C \stackrel{\triangle}{=} \{\mathbf{x} \in \mathbb{R}^r \mid \mathbf{Ax} \geq \mathbf{0}\} \tag{7}$$

with a suitable matrix $\mathbf{A} \in \mathbb{R}^{n \times r}$ (called a representation matrix of $C$) and is thus the intersection of $n$ half-spaces whose frontiers contain the origin. The dimension of $C$, noted $dim(C)$, is defined as the dimension of its affine span. A flux cone corresponds thus to a particular matrix $\mathbf{A} \in \mathbb{R}^{(2m+r_I) \times r}$ given by

$$\mathbf{A} = \begin{pmatrix} \mathbf{S} \\ -\mathbf{S} \\ \mathbf{I_{Irr}} \end{pmatrix} \tag{8}$$

where $\mathbf{I_{Irr}} \in \mathbb{R}^{r_I \times r}$ is the extension of the $(r_I \times r_I)$ identity matrix by columns of zeros corresponding to reversible reactions. This means that, for a flux cone, the homogeneous linear inequalities are of a special type: part of these ($m$) are actually equalities defining a lower-dimensional subspace given by the nullspace of the stoichiometric matrix $\mathbf{S}$, the others ($r_I$) being non-negativity constraints regarding some single coordinate variables (given by the irreversible reactions) corresponding thus to particular half-spaces defined by such positive coordinate axes.

Conversely, to any convex polyhedral cone $C$ in $\mathbb{R}^r$ defined by a representation matrix $\mathbf{A} \in \mathbb{R}^{n \times r}$, we can associate a flux cone $FC_C$ of the same dimension in $\mathbb{R}^{r+n}$ defined by

$$FC_C \stackrel{\triangle}{=} \{\begin{pmatrix} \mathbf{x} \\ \mathbf{Ax} \end{pmatrix} \in \mathbb{R}^{r+n} \mid \mathbf{x} \in C\}$$
$$= \{\mathbf{v} \in \mathbb{R}^{r+n} \mid (\mathbf{A} \ -\mathbf{I})\mathbf{v} = \mathbf{0}, \mathbf{v}_i \geq 0 \text{ for } r+1 \leq i \leq r+n\}. \tag{9}$$

Using this correspondence (which defines a bijection of $C$ onto $FC_C$), several properties, proven for flux cones, can actually be lifted to general convex polyhedral cones.

From the definition of a cone, for every nonzero element $\mathbf{x}$ of $C$, the whole half-line $\{\alpha \mathbf{x} \mid \alpha \geq 0\}$ is contained in $C$. This is called a ray of $C$. Thus a flux vector is defined up to a positive scalar multiplication. The lineality space of $C$ is the union of all lines of $C$, i.e., $\{\mathbf{x} \in C \mid -\mathbf{x} \in C\}$. If $C$ is defined by a representation matrix $\mathbf{A}$, its lineality space thus equals the nullspace of $\mathbf{A}$, i.e., $\{\mathbf{x} \in \mathbb{R}^r \mid \mathbf{Ax} = \mathbf{0}\}$. For a flux cone $FC$ given by Equation (6), its lineality space is thus constituted by the flux vectors $\mathbf{v}$ such that $\mathbf{v}_i = 0$ for $i \in \mathbf{Irr}$, i.e., flux vectors involving only reversible reactions (and thus the global flux can go in either one direction or the other). The cone $C$ is called pointed if it does not contain a line, i.e., if its lineality space is reduced to $\{\mathbf{0}\}$. For example, if $C$ is contained in a closed orthant, it is pointed (where the $3^r$ closed orthants are defined by $\{\mathbf{x} \in \mathbb{R}^r \mid \mathbf{x}_i \ \mathbf{op}_i \ 0 \text{ for } i = 1, \dots, r\}$ for an operator vector $\mathbf{op} \in \{\leq, =, \geq\}^r$). In particular, a flux cone $FC$ with only irreversible reactions ($r_I = r$) is necessarily pointed as it is included in the positive $r$-orthant (i.e., of dimension $r$). Actually, reversible flux vectors very rarely occur in metabolic networks, which therefore often give rise to pointed flux cones.

### 1.4. Extreme Vectors and Generating Sets

We are interested in finding an explicit (by extension) representation of $C$ in the form of a (minimal) set of generators. A nonzero vector $\mathbf{x} \in C$ is called extreme (or extreme pathway [9,10] if $\mathbf{x}$ is a flux vector), if

$$\mathbf{x} = \mathbf{x}^1 + \mathbf{x}^2, \text{ with nonzero } \mathbf{x}^1, \mathbf{x}^2 \in C, \text{ implies } \mathbf{x}^1 = \lambda \mathbf{x}^2 \text{ with } \lambda > 0. \tag{10}$$

If $\mathbf{x} \in C$ is extreme, then $\{\alpha \mathbf{x} \mid \alpha \geq 0\}$ is called an extreme ray of $C$ as all its nonzero elements are extreme (and thus for simplifying notations, we will not distinguish extreme vectors and extreme rays when it does not create confusion). In fact, $C$ has an extreme ray if

and only if $C$ is pointed and, in this case, the extreme rays are the edges (faces of dimension one) of $C$ and, according to Minkowski's theorem, constitute the unique minimal (finite) set of generators of $C$ for conical (i.e., non-negative linear) combination:

$$C = \{\textstyle\sum_{k \in K} \beta_k \mathbf{y}^k \mid \beta_k \geq 0\} \triangleq cone(\{\mathbf{y}^k\}) \tag{11}$$

where the $\mathbf{y}^k$'s, $k \in K$ (finite index set), are representatives of the extreme rays (unique up to positive scalar multiplication) and *cone* is the conical hull. More precisely, we get an upper bound for the number of extreme vectors that are sufficient to decompose any given nonzero vector $\mathbf{x} \in C$: $\mathbf{x} = \sum_{k \in K_\mathbf{x}} \beta_k \mathbf{y}^k$ with $|K_\mathbf{x}| \leq min(dim(C), |supp(\mathbf{x})| + |supp(\mathbf{Ax})|)$. This result can be demonstrated first for a flux vector $\mathbf{v}$ of a pointed flux cone $FC$ with an upper bound given by $|K_\mathbf{v}| \leq min(dim(C), |supp(\mathbf{v})|)$ and then for a vector $\mathbf{x}$ of a general pointed polyhedral cone $C$ by using the correspondence (9) between $C$ and $FC_C$ which maps the extreme vectors of $C$ onto the extreme vectors of $FC_C$. Extreme vectors of a pointed cone $C$ will be noted ExVs.

The Double Description (DD) method [11,12], known as Fourier–Motzkin, is an incremental algorithm (which processes one by one each linear inequality $(\mathbf{Ax})_j \geq 0$) to build an explicit description of a pointed cone $C$, as a minimal generating matrix (whose columns are in 1-to-1 correspondence with the extreme rays), from an implicit description of $C$ by a representation matrix, i.e., to enumerate its extreme rays.

If $C$ is not pointed, it is still finitely generated:

$$C = \{\textstyle\sum_{k \in K} \beta_k \mathbf{y}^k + \sum_{l \in L} \gamma_l \mathbf{z}^l \mid \beta_k \geq 0, \gamma_l \in \mathbb{R}\} = cone(\{\mathbf{y}^k, \mathbf{z}^l, -\mathbf{z}^l\}) \tag{12}$$

with (not unique this time) minimal set of generators consisting of basis vectors $\mathbf{z}^l$ of the lineality space and suitable vectors $\mathbf{y}^k$ not in the lineality space (e.g., the extreme vectors of the pointed cone obtained by intersecting $C$ with the orthogonal complement of its lineality space). Actually, Minkowski–Weyl theorem for cones states that it is equivalent for a set $C$ to being a polyhedral cone (7) or to being a finitely generated cone, i.e., the conical hull of a finite set of vectors (as the $\mathbf{y}^k$'s, $\mathbf{z}^l$'s and $-\mathbf{z}^l$'s).

### 1.5. Elementary Vectors and Conformal Generating Sets

Now, if the existence and uniqueness of a minimal set of generators for conical decomposition in a pointed polyhedral cone $C$ is satisfactory for an explicit geometric description of $C$, it is not in general meaningful for a flux cone $FC$ representing the steady-state flux vectors of a metabolic network. In fact, for a metabolic pathway, only a conical decomposition without any cancellations is biochemically meaningful, as a reversible reaction cannot have a net rate in opposite directions in the contributing pathways. Indeed, the second law of thermodynamics states that a reaction can only carry flux in the direction of negative Gibbs free energy of the reaction, which is imposed by the values of the concentrations of the metabolites. This means that, when decomposing a flux vector, only so-called conformal sums, i.e., sums without cancellations, are biochemically admissible. A sum $\mathbf{v} = \mathbf{v}^1 + \mathbf{v}^2$ of vectors is called conformal if, for all $i \in \{1, \dots, r\}$:

$$\mathbf{v}_i = 0 \text{ implies } \mathbf{v}_i^1 = \mathbf{v}_i^2 = 0, \mathbf{v}_i > 0 \text{ implies } \mathbf{v}_i^1, \mathbf{v}_i^2 \geq 0, \mathbf{v}_i < 0 \text{ implies } \mathbf{v}_i^1, \mathbf{v}_i^2 \leq 0. \tag{13}$$

An equivalent definition is to define a sum $\mathbf{v} = \mathbf{v}^1 + \mathbf{v}^2$ as conformal if

$$sign(\mathbf{v}^1), sign(\mathbf{v}^2) \leq sign(\mathbf{v}) \tag{14}$$

where the sign vector $sign(\mathbf{v}) \in \{-, 0, +\}^r$ is defined by applying the sign function component-wise, i.e., $sign(\mathbf{v})_i = sign(\mathbf{v}_i)$, and the partial order $\leq$ on $\{-, 0, +\}^r$ is defined by applying component-wise the partial order on $\{-, 0, +\}$ induced by $0 < -$ and $0 < +$. For example, there is a one-to-one mapping between closed orthants $O$ and sign vectors $\eta$, defined by $O = \{\mathbf{x} \in \mathbb{R}^r \mid sign(\mathbf{x}) \leq \eta\}$ ($O$ will be called defined by $\eta$ and noted

$O_\eta$), which induces a one-to-one mapping between closed $r$-orthants and full support (i.e., with only nonzero components) sign vectors. For $\xi, \eta$ sign vectors and $\mathbf{v}$ a vector, we say that $\xi$ conforms to $\eta$ if $\xi \le \eta$ and that $\mathbf{v}$ conforms to $\eta$ if $sign(\mathbf{v}) \le \eta$. We call two vectors $\mathbf{v}^1, \mathbf{v}^2$ conformal if $sign(\mathbf{v}^1), sign(\mathbf{v}^2) \le \eta$ for a certain sign vector $\eta$ or, equivalently, if $\mathbf{v}_i^1 \mathbf{v}_i^2 \ge 0$ for all $i$ (so, $\mathbf{v}^1 + \mathbf{v}^2$ is a conformal sum if and only if $\mathbf{v}^1$ and $\mathbf{v}^2$ are conformal). A conformal sum $\mathbf{v} = \mathbf{v}^1 + \mathbf{v}^2$, i.e., verifying Equation (14), will be noted $\mathbf{v} = \mathbf{v}^1 \oplus \mathbf{v}^2$. It is therefore natural to look for generators as conformally non-decomposable vectors, where a nonzero vector $\mathbf{x}$ of a convex polyhedral cone $C$ is called conformally non-decomposable, if

$$\mathbf{x} = \mathbf{x}^1 \oplus \mathbf{x}^2, \text{ with nonzero } \mathbf{x}^1, \mathbf{x}^2 \in C, \text{ implies } \mathbf{x}^1 = \lambda \mathbf{x}^2 \text{ with } \lambda > 0. \tag{15}$$

A vector $\mathbf{x}$ (resp., flux vector $\mathbf{v}$) of a convex polyhedral cone $C$ (resp., a flux cone $FC$) is called elementary [7] if it is conformally non-decomposable. All nonzero elements of the ray defined by an elementary vector are elementary, i.e., elementary vectors are unique up to positive scalar multiplication (and thus we will often not distinguish elementary vectors and elementary rays). Elementary vectors (resp., flux vectors) will be noted EVs (resp., EFVs).

The elementary rays constitute the unique minimal (finite) set of conformal generators of $C$, i.e., generators for conformal conical sum:

$$C = \{\bigoplus_{g \in VG} \beta_g \mathbf{e}^g \mid \beta_g \ge 0\} \overset{\triangle}{=} cone_\oplus(\{\mathbf{e}^g\}) \tag{16}$$

where the $\mathbf{e}^g$'s, $g \in VG$ (finite index set), are representatives of the elementary rays (unique up to positive scalar multiplication) and $cone_\oplus$ is the conical conformal hull. More precisely, we get an upper bound for the number of elementary vectors that are sufficient to decompose any given nonzero vector $\mathbf{x} \in C$: $\mathbf{x} = \bigoplus_{g \in VG_\mathbf{x}} \beta_g \mathbf{e}^g$ with $|VG_\mathbf{x}| \le min(dim(C), |supp(\mathbf{x})| + |supp(\mathbf{A}\mathbf{x})|)$. This result can be demonstrated first for a flux vector $\mathbf{v}$ of a flux cone $FC$ with an upper bound given by $|VG_\mathbf{v}| \le min(dim(C), |supp(\mathbf{v})|)$ and then for a vector $\mathbf{x}$ of a general polyhedral cone $C$ by using the correspondence (9) between $C$ and $FC_C$ which maps the elementary vectors of $C$ onto the elementary flux vectors of $FC_C$.

It follows from (10) and (15) that an extreme vector of $C$ is elementary, i.e., ExVs $\subseteq$ EVs, but the converse is generally false as a conformally non-decomposable vector may be conically decomposable. Nevertheless, if $C$ is contained in a closed orthant, there is identity between extreme vectors and elementary vectors, i.e., ExVs = EVs. More precisely, for nonzero $\mathbf{x} \in C$ and $O$ a closed orthant with $\mathbf{x} \in O$, then $\mathbf{x}$ is elementary in $C$ if and only if it is extreme in $C \cap O$. It results that the elementary vectors of $C$ are the extreme vectors of intersections of $C$ with any closed orthant:

$$EVs(C) = \bigcup_{O \text{ orthant}} ExVs(C \cap O). \tag{17}$$

Elementary vectors of $C$ can thus be obtained by using algorithms, such as DD, to compute extreme vectors ExVs of the pointed polyhedral cones $C \cap O$. By doing this, it is convenient to select in Equation (17) only a minimal subset of closed orthants $O$ in order to avoid equality or inclusion between the $C \cap O$'s (nonempty intersection can obviously not be avoided as orthants are closed). It is clearly enough to consider only the $2^r$ closed $r$-orthants of maximal dimension $r$, but this does not avoid equality or inclusion. Let $\{\eta^i\}$ be the maximal (for the partial order defined above on $\{-, 0, +\}^r$) sign vectors of $sign(C) = \{sign(\mathbf{x}) \mid \mathbf{x} \in C\}$. It is then enough in Equation (17) to consider the closed orthants $O_i = \{\mathbf{x} \in \mathbb{R}^r \mid sign(\mathbf{x}) \le \eta^i\}$, and there is no equality or inclusion between the $C \cap O_i$'s, where $C \cap O_i = C_{\le \eta^i} = \{\mathbf{x} \in C \mid sign(\mathbf{x}) \le \eta^i\}$. The $C_{\le \eta^i}$'s are called topes, noted Ts (flux topes [13] noted FTs for a flux cone $FC$). $C$ is thus decomposed into topes and Equation (17) can be rewritten as

$$EVs(C) = \bigcup_{\eta^i \text{ maximal in } sign(C)} ExVs(C_{\le \eta^i}). \tag{18}$$

Note that, for a flux cone *FC* Equation (6), a FT is defined by specifying a maximal subset of reactions with fixed directions (thus fixing the directions of reversible reactions), the others having necessarily a zero flux. This simplifies if *FC* is consistent, i.e., without unused reaction, which means that every reaction, in every possible direction for reversible reactions, is supported by a flux vector: $\forall i \in \{1,\ldots,r\}\ \exists \mathbf{v} \in FC\ \mathbf{v}_i > 0$ and $\forall i \in \{1,\ldots,r\}\backslash\mathbf{Irr}$ $\exists \mathbf{v} \in FC\ \mathbf{v}_i < 0$. We can always assume *FC* consistent after a preprocessing step (practically, this can be achieved by using flux variability analysis [14]) that removes all reactions that cannot carry nonzero steady-state flux and changes all reversible reactions that cannot carry flux in both directions into irreversible ones. In this case, every remaining reaction in every possible direction is supported by a flux vector with full support (i.e., with nonzero flux in any reaction) and all FTs $FC_{\leq \boldsymbol{\eta}^i}$ have full support, i.e., the $\boldsymbol{\eta}^i$'s have full support or, equivalently, the $O_i$'s are *r*-orthants. An obvious upper bound for the number of FTs is thus $2^{r-r_I}$.

Now, for a flux cone *FC*, another commonly used method is, at the extreme opposite, to have it included into a single (positive) orthant in a higher dimension by splitting each reversible reaction *i* into a forward $i^+$ and a backward $i^-$ irreversible reaction. This means decomposing a flux in *i* as $\mathbf{v}_i = \mathbf{v}_i^+ - \mathbf{v}_i^-$ with $\mathbf{v}_i^+ = \mathbf{v}_{i^+} \geq 0$ and $\mathbf{v}_i^- = \mathbf{v}_{i^-} \geq 0$. Columns of the stoichiometric matrix $\mathbf{S}$ corresponding to reversible reactions *i* are negated (which means exchanging the roles of substrates and products in *i*) and appended to $\mathbf{S}$ as new columns to form the new stoichiometric matrix $\widetilde{\mathbf{S}} \in \mathbb{R}^{m \times \tilde{r}}$, where $\tilde{r} = 2r - r_I$ is the new number of reactions and all $\tilde{r}$ reactions are now irreversible, $\widetilde{\mathbf{Irr}} = \{1,\ldots,\tilde{r}\}$. *FC* is in one-to-one correspondence with vectors $\mathbf{v}$ of $\widetilde{FC}$ such that $\mathbf{v}_{i^+}.\mathbf{v}_{i^-} = 0$ for *i* reversible. In particular, the fluxes of the form $\mathbf{v}_{i^+} = \mathbf{v}_{i^-} > 0$ with all other components being null are obtained as extreme vectors of $\widetilde{FC}$ but represent futile cycles (involving reactions $i^+$ and $i^-$) without biological reality and must be eliminated. Finally, EFVs (*FC*) are in one-to-one correspondence with ExVs ($\widetilde{FC}$) \{futile cycles} (called at the origin extreme currents in stoichiometric network analysis [15]). $\widetilde{FC}$ is included in the positive $\tilde{r}$-orthant and has thus only one FT.

We therefore have two opposite ways of dealing with reversible reactions for computing EFVs of a flux cone *FC*: either splitting each reversible reaction into two irreversible ones, such that *FC* is reduced to a single FT at the price of an increase in the space dimension by $r - r_I$ (which can cause serious efficiency problems to algorithms such as DD) or keeping the reversible reactions unchanged and decomposing *FC* into FTs, in each of which the directions of reversible reactions are fixed, at the price of a potentially exponential (in terms of $r - r_I$) number of FTs to consider. All intermediate cases, where only a subset of reversible reactions are split into irreversible ones and the others are processed by decomposition into FTs, are obviously possible. Independently of the solution adopted, we will work most of the time in a given FT for *FC*, defined by a (full support if *FC* is consistent) sign vector $\boldsymbol{\eta}$, and the EFVs of *FC* which conform to $\boldsymbol{\eta}$ are thus given by the ExVs of this FT $FC_{\leq \boldsymbol{\eta}}$.

### 1.6. Elementary Modes

The null value 0 plays a component-wise crucial role in definitions of the support of a vector (4), of a flux cone (6) and of a conformal sum Equations (13) and (14). A close relationship results between support-minimal vectors and elementary vectors in a flux cone. A nonzero vector $\mathbf{x} \in C$ is called support-minimal, if

$$supp(\mathbf{x}') \subseteq supp(\mathbf{x}), \text{ with nonzero } \mathbf{x}' \in C, \text{ implies } supp(\mathbf{x}') = supp(\mathbf{x}). \tag{19}$$

A nonzero vector $\mathbf{x}$ (resp., flux vector $\mathbf{v}$) of a convex polyhedral cone *C* (resp., a flux cone *FC*) is called elementary mode (resp., elementary flux mode) if it is support-minimal (the concept of elementary flux mode was first introduced, under the name of elementary vector [16], for a subspace of $\mathbb{R}^r$, i.e., for a flux linear subspace *FS* Equation (3) and then [17] for a flux cone *FC* with a definition actually closer to that of a support-wise

non-decomposable vector). All nonzero elements of the ray defined by an elementary mode are elementary modes having the same support, i.e., elementary modes are unique up to positive scalar multiplication (and we will therefore in general identify two positively proportional elementary modes). Elementary modes (resp., flux modes) will be noted EMs (resp., EFMs). The concept of EFM in a flux cone *FC* is biologically significant as it represents a minimal pathway operating in a steady state, i.e., with all reactions involved necessarily active (with a nonzero net rate), which means that no proper sub-pathway can operate in a steady state.

Note that if $\mathbf{x}$ is an EFM, then $sign(\mathbf{x})$ is a minimal (for the partial order defined above on $\{-, 0, +\}^r$) nonzero element of $sign(FC)$ and, conversely, it is shown that a minimal nonzero sign vector $\boldsymbol{\sigma} \in sign(FC)$ determines an EFM $\mathbf{x}$ with $sign(\mathbf{x}) = \boldsymbol{\sigma}$ by $FC_{\leq \boldsymbol{\sigma}} = \{\mathbf{v} \in FC \mid sign(\mathbf{v}) \leq \boldsymbol{\sigma}\} = \{\mathbf{v} \in FC \mid sign(\mathbf{v}) = \boldsymbol{\sigma}\} \cup \{\mathbf{0}\} = \{\lambda \mathbf{x} \mid \lambda \geq 0\}$. There is thus a one-to-one mapping between EFMs and minimal nonzero sign vectors of $sign(FC)$. Comparing with the one-to-one mapping between FTs and maximal sign vectors of $sign(FC)$, we see that EFMs and FTs are dual concepts.

Now, for a flux cone *FC*, support-minimality and conformal non-decomposability are equivalent properties, i.e., there is identity between elementary flux modes and elementary flux vectors: EFMs = EFVs. For metabolic pathways in a flux cone *FC*, there is therefore identity between minimal (for support inclusion) pathways and non-decomposable (for conformal sum) pathways. From Equation (16), the EFMs constitute a conformal generating set (i.e., generating set for conformal sum) for *FC*, and in fact the unique minimal such set (for that matter one way of proving Equation (16) for flux cones *FC* is to prove it with EFMs as a conformal generating set and to prove that EFMs = EFVs). From Equation (18), for any maximal sign vector $\boldsymbol{\eta}$ of $sign(FC)$, the EFMs of *FC* which conform to $\boldsymbol{\eta}$ are the EFMs of the FT $FC_{\leq \boldsymbol{\eta}}$ and coincide with the ExVs of the said FT, this result being the basis of methods for computing EFMs [18,19]. EFMs are thus decomposed into subsets according to the decomposition of *FC* into flux topes [13]: the EFMs of the FT $FC_{\leq \boldsymbol{\eta}}$ correspond to the $FC_{\leq \boldsymbol{\sigma}}$'s, where the $\boldsymbol{\sigma}$'s are the minimal nonzero sign vectors of $sign(FC)$ such that $\boldsymbol{\sigma} \leq \boldsymbol{\eta}$.

For a general polyhedral cone *C*, there is no direct relationship between elementary modes and elementary vectors. Nevertheless, from Equation (16), it follows that, for any EM in *C*, there is an EV with the same support. Thus, all minimal support patterns of vectors appear in the set of supports of elementary vectors and are actually the minimal elements in this set for subset inclusion:

$$supp(EMs) = Min_{\subseteq}\{supp(EVs)\}. \tag{20}$$

In addition, for a general polyhedral cone *C*, the correspondence (9) between *C* and the higher dimensional flux cone $FC_C$ maps the elementary vectors of *C* onto the elementary flux vectors of $FC_C$, i.e., the elementary flux modes of $FC_C$:

$$EVs(C) = \left\{\mathbf{x} \in \mathbb{R}^r \mid \begin{pmatrix} \mathbf{x} \\ \mathbf{y} \end{pmatrix} \in EFMs(FC_C)\right\} \tag{21}$$

with $FC_C$ given by Equation (9).

Moreover, it remains true that any vector which is support-minimal in a given $C_{\leq \boldsymbol{\eta}}$ is actually support minimal in *C*, as it depends only on the convexity of *C*: if $\mathbf{x}$ and $\mathbf{x}'$ are vectors in a convex set with $supp(\mathbf{x}') \subset supp(\mathbf{x})$, then a vector $\mathbf{x}''$ exists in this convex set with $sign(\mathbf{x}'') \leq sign(\mathbf{x})$ and $supp(\mathbf{x}'') \subset supp(\mathbf{x})$ (we take $\mathbf{x}'' = \lambda \mathbf{x}' + (1 - \lambda)\mathbf{x}$ with $\lambda$ minimal in $(0, 1]$ such that $\mathbf{x}''_i = 0$ for a certain *i* with $\mathbf{x}_i \neq 0$). Thus, EMs of *C* can be computed tope by tope:

$$EMs(C) = \bigcup_{\boldsymbol{\eta}^i \text{ maximal in } sign(C)} EMs(C_{\leq \boldsymbol{\eta}^i}). \tag{22}$$

See Figure 1 for an illustration of the concepts of extreme vectors, elementary vectors and elementary modes in a convex polyhedral cone.

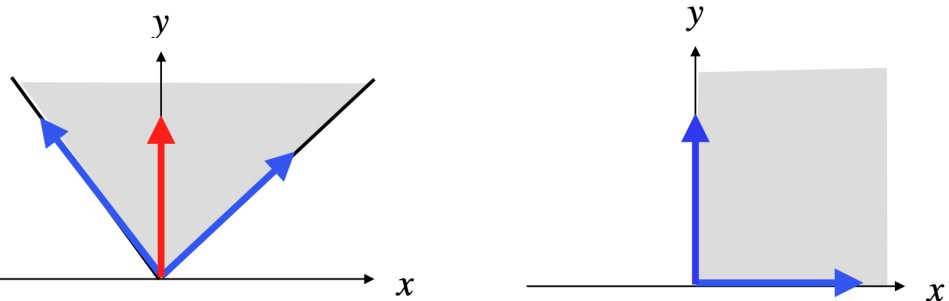

**Figure 1.** On the left, we consider the convex polyhedral cone shaded in gray. The extreme vectors are the blue vectors. The elementary vectors are all the blue or red vectors and coincide with the extreme vectors for all topes (here we have two topes: the subcone with $x \geq 0$ and the subcone with $x \leq 0$). The elementary mode is the red vector (common to both topes). Flux cones being defined by linear inequalities of the form $\mathbf{v}_i \geq 0$, elementary modes coincide with elementary vectors in this case (at right).

*1.7. Inhomogeneous Linear Constraints and Polyhedra*

Additionally, in this standard setting, fluxes may be constrained by other constraints, typically lower and upper bounds regarding reaction rates:

$$\mathbf{v}_i^- \leq \mathbf{v}_i \leq \mathbf{v}_i^+ \tag{23}$$

or, more generally, any set of inhomogeneous linear constraints, noted **ILC**, that can be written in the general form

$$\mathbf{Gv} \geq \mathbf{h} \tag{24}$$

where $\mathbf{G} \in \mathbb{R}^{l \times r}$ is a matrix and $\mathbf{h} \in \mathbb{R}^l$ a vector with nonzero components, defining a general inhomogeneous convex polyhedron $P_{\mathbf{ILC}} = \{\mathbf{v} \in \mathbb{R}^r \mid \mathbf{Gv} \geq \mathbf{h}\}$. The solution space $Sol_{\mathbf{S,Irr,ILC}}$ is thus now a s-polyhedron or flux polyhedron noted $FP$ and defined by

$$Sol_{\mathbf{S,Irr,ILC}} \triangleq FP \triangleq \{\mathbf{v} \in \mathbb{R}^r \mid \mathbf{Sv} = \mathbf{0}, \mathbf{v}_i \geq 0 \text{ for } i \in \mathbf{Irr}, \mathbf{Gv} \geq \mathbf{h}\} = FC \cap P_{\mathbf{ILC}}. \tag{25}$$

This is a particular case of (general) convex polyhedron that is defined implicitly by finitely many linear inequalities:

$$P \triangleq \{\mathbf{x} \in \mathbb{R}^r \mid \mathbf{Ax} \geq \mathbf{b}\} \tag{26}$$

with a suitable matrix $\mathbf{A} \in \mathbb{R}^{n \times r}$ and vector $\mathbf{b} \in \mathbb{R}^n$ and is thus the intersection of $n$ (affine) half-spaces ($\mathbb{R}^r$ is equipped with both its structure of affine space, with origin $\mathbf{0}$, and its underlying structure of vector space and we will identify a point in the affine space with the corresponding vector in the vector space). Its dimension is defined as the dimension of its affine span. In this way, a flux polyhedron $FP$ corresponds to a particular matrix $\mathbf{A} \in \mathbb{R}^{(2m+r_I+l) \times r}$ and vector $\mathbf{b} \in \mathbb{R}^{2m+r_I+l}$ given by

$$\mathbf{A} = \begin{pmatrix} \mathbf{S} \\ -\mathbf{S} \\ \mathbf{I_{Irr}} \\ \mathbf{G} \end{pmatrix} \qquad \mathbf{b} = \begin{pmatrix} \mathbf{0} \\ \mathbf{0} \\ \mathbf{0} \\ \mathbf{h} \end{pmatrix} \tag{27}$$

meaning that the inequalities that are homogeneous actually divide into $m$ equalities defining a lower-dimensional subspace given by the nullspace of the stoichiometric matrix $\mathbf{S}$ and into $r_I$ non-negativity constraints regarding certain single coordinate variables (given by the irreversible reactions). A polyhedral cone (resp., flux cone) is a special case of polyhedron (resp., flux polyhedron) where $\mathbf{b} = \mathbf{0}$ (resp., $\mathbf{ILC} = \varnothing$).

To any nonempty polyhedron $P$ given by Equation (26) is associated its so-called recession cone $C_P = \{\mathbf{x} \in \mathbb{R}^r \mid \mathbf{Ax} \geq \mathbf{0}\}$, which is the polyhedral cone containing all unbounded directions (rays) of $P$ (if $P$ is a polyhedral cone, then $P = C_P$). A bounded polyhedron is called a polytope and thus $P$ is a polytope if and only if its recession cone is trivial: $C_P = \{\mathbf{0}\}$. $P$ is called pointed if its recession cone $C_P$ is pointed, i.e., if its lineality space $\{\mathbf{x} \in \mathbb{R}^r \mid \mathbf{Ax} = \mathbf{0}\}$, also called the lineality space of $P$ (as it contains all unbounded lines of $P$), is trivial. For a flux polyhedron $FP$, we have: $C_{FP} = FC \cap C_{P_{\mathbf{ILC}}}$ (thus $FP$ can be a polytope without $P_{\mathbf{ILC}}$ being so and be pointed without either $FC$ or $P_{\mathbf{ILC}}$ being so). Note that $C_{FP}$ is not in general a flux cone.

### 1.7.1. Extreme Points and Vectors and Generating Sets

A vector $\mathbf{x} \in P$ is called an extreme point, if it cannot be written as a convex combination of two distinct vectors of $P$:

$$\mathbf{x} = \lambda \mathbf{x}^1 + (1 - \lambda)\mathbf{x}^2, \text{ with } \mathbf{x}^1, \mathbf{x}^2 \in P \text{ and } 0 < \lambda < 1, \text{ implies } \mathbf{x}^1 = \mathbf{x}^2. \quad (28)$$

Note that, as $P$ is convex, it is enough to consider the midpoint of $\mathbf{x}^1$ and $\mathbf{x}^2$, i.e., to take $\lambda = 0.5$. Extreme points coincide with vertices of $P$, where a vertex of $P$ is defined as a face of dimension 0. $P$ is pointed if and only if it has a vertex and in this case, according to Minkowski's theorem, the vertices of $P$ and the extreme rays of $C_P$ constitute the unique minimal (finite) set of "bounded" and "unbounded" generators of $P$ for convex and conical combination, respectively:

$$P = \{\textstyle\sum_{j \in J} \alpha_j \mathbf{p}^j + \sum_{k \in K} \beta_k \mathbf{y}^k \mid \alpha_j, \beta_k \geq 0, \sum_{j \in J} \alpha_j = 1\} \stackrel{\triangle}{=} conv(\{\mathbf{p}^j\}) + cone(\{\mathbf{y}^k\}) \quad (29)$$

where the $\mathbf{p}^j$'s, $j \in J$ (finite index set), are the extreme points (vertices) of $P$, noted ExPs, and the $\mathbf{y}^k$'s, $k \in K$ (finite index set), are the extreme vectors of $C_P$ (unique up to positive scalar multiplication), noted ExVs, and *conv* is the convex hull (if $P$ is a pointed polyhedral cone, then it has only one vertex, which is the zero vector, and the Equation (29) reduces to Equation (11)). More precisely, we get an upper bound for the number of extreme points and vectors that are sufficient to decompose any given vector $\mathbf{x} \in P$: $\mathbf{x} = \sum_{j \in J_\mathbf{x}} \alpha_j \mathbf{p}^j + \sum_{k \in K_\mathbf{x}} \beta_k \mathbf{y}^k$ with $|J_\mathbf{x}| + |K_\mathbf{x}| \leq min(dim(P) + 1, |supp(\mathbf{x})| + |supp(\mathbf{Ax} - \mathbf{b})| + 1)$. This result can be deduced from result (11) for a pointed flux cone $FC$ by using the following correspondence between $P$ and such a flux cone.

In fact, to any convex polyhedron $P$ in $\mathbb{R}^r$ defined by a matrix $\mathbf{A} \in \mathbb{R}^{n \times r}$ and vector $\mathbf{b} \in \mathbb{R}^n$ Equation (26), we can associate a flux cone $FC_P$ in a higher dimension $\mathbb{R}^{r+1+n}$ defined by

$$
\begin{aligned}
FC_P &\stackrel{\triangle}{=} \left\{ \begin{pmatrix} \mathbf{x} \\ \xi \\ \mathbf{Ax} - \xi\mathbf{b} \end{pmatrix} \in \mathbb{R}^{r+1+n} \mid \xi \geq 0, \mathbf{Ax} - \xi\mathbf{b} \geq \mathbf{0} \right\} \\
&= \{\mathbf{v} \in \mathbb{R}^{r+1+n} \mid \begin{pmatrix} \mathbf{A} & -\mathbf{b} & -\mathbf{I} \end{pmatrix}\mathbf{v} = \mathbf{0}, \mathbf{v}_i \geq 0 \text{ for } r+1 \leq i \leq r+1+n\}.
\end{aligned}
\quad (30)
$$

This introduces a correspondence between vectors $\mathbf{x}$ of $P$ and vectors $\begin{pmatrix} \mathbf{x} \\ 1 \\ \mathbf{Ax} - \mathbf{b} \end{pmatrix}$ of $FC_P$, which maps vertices of $P$ onto extreme vectors of $FC_P$ with component $\xi = 1$, and between vectors $\mathbf{x}$ of $C_P$ and vectors $\begin{pmatrix} \mathbf{x} \\ 0 \\ \mathbf{Ax} \end{pmatrix}$ of $FC_P$, which maps extreme vectors of $C_P$ onto extreme vectors of $FC_P$ with component $\xi = 0$. Thanks to this correspondence, several properties, proven for flux cones, can be lifted to general convex polyhedra.

For the particular case where $P$ is a flux polyhedron $FP$ given by Equation (25), the correspondence (30) simplifies by associating to $FP$ the flux cone $FC_{FP}$ in dimension $\mathbb{R}^{r+1+l}$ defined by

$$
\begin{aligned}
FC_{FP} &\triangleq \{ \begin{pmatrix} \mathbf{v} \\ \xi \\ \mathbf{Gv} - \xi\mathbf{h} \end{pmatrix} \in \mathbb{R}^{r+1+l} \mid \mathbf{Sv} = \mathbf{0}, \mathbf{v}_i \geq 0 \text{ for } i \in \mathbf{Irr}, \xi \geq 0, \mathbf{Gv} - \xi\mathbf{h} \geq \mathbf{0} \} \\
&= \{ \mathbf{v}' \in \mathbb{R}^{r+1+l} \mid \begin{pmatrix} \mathbf{S} & \mathbf{0} & \mathbf{0} \\ \mathbf{G} & -\mathbf{h} & -\mathbf{I} \end{pmatrix} \mathbf{v}' = \mathbf{0}, \\
&\qquad \mathbf{v}'_i \geq 0 \text{ for } i \in \mathbf{Irr} \text{ and for } r+1 \leq i \leq r+1+l \}.
\end{aligned}
\tag{31}
$$

The DD method builds an explicit description of a pointed polyhedron $P$, in the form of two generating matrices whose columns are respectively the $\mathbf{p}^j$'s and the $\mathbf{y}^k$'s, from an implicit description of $P$ as in Equation (26), i.e., enumerates its vertices ExPs and extreme vectors ExVs.

If $P$ is not pointed, it is still finitely generated:

$$
\begin{aligned}
P &= \{ \textstyle\sum_{j \in J} \alpha_j \mathbf{p}^j + \sum_{k \in K} \beta_k \mathbf{y}^k + \sum_{l \in L} \gamma_l \mathbf{z}^l \mid \alpha_j, \beta_k \geq 0, \gamma_l \in \mathbb{R}, \sum_{j \in J} \alpha_j = 1 \} \\
&= conv(\{\mathbf{p}^j\}) + cone(\{\mathbf{y}^k, \mathbf{z}^l, -\mathbf{z}^l\})
\end{aligned}
\tag{32}
$$

with a (not unique this time) minimal set of generators consisting of basis vectors $\mathbf{z}^l$ of the linearity space and suitable vectors $\mathbf{p}^j$ and $\mathbf{y}^k$ (e.g., the vertices and extreme vectors of the pointed polyhedron obtained by intersecting $P$ with the orthogonal complement of its lineality space; if $P$ is a non-pointed polyhedral cone, there is no nonzero $\mathbf{p}^j$ and Equation (32) reduces to Equation (12)). In fact, Minkowski–Weyl theorem for polyhedra states that it is equivalent for a set $P$ to be a polyhedron (26) or to be finitely generated, i.e., to be the Minkowski sum of the convex hull of a finite set of vectors (as the $\mathbf{p}^j$'s) and of the conical hull of a finite set of vectors (as the $\mathbf{y}^k$'s, $\mathbf{z}^l$'s and $-\mathbf{z}^l$'s).

### 1.7.2. Elementary Points and Vectors and Conformal Generating Sets

However, such a decomposition into a finite set of generators is not in general satisfactory for a flux polyhedron as only a decomposition without any cancellations is biochemically meaningful, as was stipulated for a flux cone in Section 1.5. In the same way as we replaced, as generators for a polyhedral cone, extreme vectors by conformally non-decomposable vectors, we will replace, as generators for a polyhedron $P$, extreme points (vertices) by convex-conformally non-decomposable vectors (and, for its recession cone $C_P$, extreme vectors by conformally non-decomposable vectors). A vector $\mathbf{x}$ of a polyhedron $P$ is called convex-conformally non-decomposable, if

$$
\mathbf{x} = \lambda\mathbf{x}^1 \oplus (1-\lambda)\mathbf{x}^2, \text{ with } \mathbf{x}^1, \mathbf{x}^2 \in P \text{ and } 0 < \lambda < 1, \text{ implies } \mathbf{x}^1 = \mathbf{x}^2.
\tag{33}
$$

Given a polyhedron $P$ (resp., flux polyhedron $FP$), a vector (resp., flux vector) $\mathbf{x}$ is called an elementary point (resp., elementary flux point)—also called "bounded" elementary vector—of $P$ if $\mathbf{x} \in P$ is convex-conformally non-decomposable and is called an elementary vector (resp., elementary flux vector)—also called "unbounded" elementary vector—of $P$ if $\mathbf{x} \in C_P$ is conformally non-decomposable (it is unique only up to positive scalar multiplication) [7]. Elementary points (resp., flux points) will be noted EPs (resp., EFPs) and elementary vectors (resp., flux vectors) will be noted EVs (resp., EFVs), which is consistent with the same notation for polyhedral cones and flux cones. We will note Es = EPs ∪ EVs (resp., EFs = EFPs ∪ EFVs) the elementary elements (resp., elementary fluxes) of $P$ (resp., $FP$).

The elementary points and the elementary rays constitute the unique minimal (finite) set of conformal generators of $P$, i.e., generators for convex-conformal (for elementary points) and conformal (for elementary vectors) sum:

$$P = \{\bigoplus_{g \in PG} \alpha_g \mathbf{e}^g \oplus \bigoplus_{g \in VG} \beta_g \mathbf{e}^g \mid \alpha_g, \beta_g \geq 0, \sum_{g \in PG} \alpha_g = 1\}$$
$$\triangleq conv_{\oplus}(\{\mathbf{e}^g \mid g \in PG\}) \oplus cone_{\oplus}(\{\mathbf{e}^g \mid g \in VG\}) \tag{34}$$

where the $\mathbf{e}^g$'s, $g \in PG$ (finite index set), are the elementary points and the $\mathbf{e}^g$'s, $g \in VG$ (finite index set), are the elementary vectors (unique up to positive scalar multiplication), and $conv_{\oplus}$ is the convex conformal hull. More precisely, we get an upper bound for the number of elementary points and vectors that are sufficient to decompose any given vector $\mathbf{x} \in P$: $\mathbf{x} = \bigoplus_{g \in PG_{\mathbf{x}}} \alpha_g \mathbf{e}^g \oplus \bigoplus_{g \in VG_{\mathbf{x}}} \beta_g \mathbf{e}^g$ with $|PG_{\mathbf{x}}| + |VG_{\mathbf{x}}| \leq min(dim(P)+1, |supp(\mathbf{x})| + |supp(\mathbf{Ax} - \mathbf{b})| + 1)$. This result can be demonstrated from result (16) for a flux cone by using the correspondence (30) between $P$ and $FC_P$ which maps the elementary points of $P$ onto the elementary flux vectors of $FC_P$ with component $\xi = 1$ and the elementary vectors of $P$ onto the elementary flux vectors of $FC_P$ with component $\xi = 0$.

We already know that an extreme vector of $C_P$ is elementary, and is therefore an elementary vector of $P$, i.e., ExVs $\subseteq$ EVs. It follows from Equations (28) and (33) that an extreme point (vertex) of $P$ is an elementary point of $P$, i.e., ExPs $\subseteq$ EPs, but the converse is generally false as a convex-conformally non-decomposable vector may be convex decomposable. Nevertheless, if $P$ is contained in a closed orthant (and thus $C_P$ too), any sum of vectors of $P$ (resp., $C_P$) is conformal and thus there is identity between extreme points (vertices) and elementary points (resp., between extreme vectors and elementary vectors), i.e., ExPs = EPs and ExVs = EVs. More precisely, for $\mathbf{x} \in P$ (resp., $\mathbf{x} \in C_P$ and nonzero) and $O$ a closed orthant with $\mathbf{x} \in O$, then $\mathbf{x}$ is an elementary point (resp., elementary vector) of $P$ if and only if it is a vertex in $P \cap O$ (resp., an extreme vector in $C_P \cap O = C_{P \cap O}$). It follows that the elementary points (resp., elementary vectors) of $P$ are the vertices (resp., extreme vectors) of intersections of $P$ (resp., $C_P$) with any closed orthant, which are pointed subpolyhedra (resp., pointed subcones):

$$EPs(P) = \bigcup_{O\ orthant} ExPs(P \cap O) \qquad EVs(P) = \bigcup_{O\ orthant} ExVs(C_P \cap O). \tag{35}$$

Note in particular that, if $\mathbf{0} \in P$, then $\mathbf{0} \in$ EPs. Elementary points and vectors can therefore be obtained by using algorithms, such as DD, to compute vertices ExPs and extreme vectors ExVs of the pointed polyhedra $P \cap O$. It is obvious that considering only the $2^r$ closed $r$-orthants is enough. Now, as for polyhedral cones, decomposing the polyhedron into topes is better:

$$EPs(P) = \bigcup_{\boldsymbol{\eta}^i\ maximal\ in\ sign(P)} ExPs(P_{\leq \boldsymbol{\eta}^i})$$
$$EVs(P) = \bigcup_{\boldsymbol{\eta}^j\ maximal\ in\ sign(C_P)} ExVs(C_{P_{\leq \boldsymbol{\eta}^j}}). \tag{36}$$

If $O_i$ is the closed orthant defined by $\boldsymbol{\eta}^i$, then the corresponding tope for $P$ is $P \cap O_i = P_{\leq \boldsymbol{\eta}^i} = \{\mathbf{x} \in P \mid sign(\mathbf{x}) \leq \boldsymbol{\eta}^i\}$ and, as seen for polyhedral cones, $C_P \cap O_j = C_{P_{\leq \boldsymbol{\eta}^j}}$ is a tope for the recession cone $C_P$. Examine how the equality $C_{P_{\leq \boldsymbol{\eta}}} = C_{P_{\leq \boldsymbol{\eta}}}$, for $\boldsymbol{\eta}$ an arbitrary sign vector, can be expressed in terms of topes. Note first that any $\boldsymbol{\eta}^j$ is dominated by at least one $\boldsymbol{\eta}^i$ for the partial order $\leq$ on $\{-, 0, +\}^r$: $\forall \boldsymbol{\eta}^j \exists \boldsymbol{\eta}^i \boldsymbol{\eta}^j \leq \boldsymbol{\eta}^i$, which means that $O_j$ is a sub-orthant of $O_i$, and we have $C_{P_{\leq \boldsymbol{\eta}^j}} = C_{P_{\leq \boldsymbol{\eta}^i}}$, expressing the relation between the topes for the recession cone of the polyhedron and the recession cones of certain of the polyhedron topes (precisely those topes $P_{\leq \boldsymbol{\eta}^i}$ for which $\boldsymbol{\eta}^i$ dominates an $\boldsymbol{\eta}^j$, necessarily unique). More generally, the recession cone of any tope $P_{\leq \boldsymbol{\eta}^i}$ for $P$ can be expressed as a subcone of a tope for the recession cone of $P$ by $C_{P_{\leq \boldsymbol{\eta}^i}} = C_{P_{\leq \mathbf{c}(\boldsymbol{\eta}^i)}}$, where $\mathbf{c}(\boldsymbol{\eta}^i) = max\{\boldsymbol{\eta} \in sign(C_P) \mid \boldsymbol{\eta} \leq \boldsymbol{\eta}^i\}$ is the greatest (it is necessarily unique) sign vector in $sign(C_P)$ dominated by $\boldsymbol{\eta}^i$ (thus, if $\boldsymbol{\eta}^i$ does not dominate any $\boldsymbol{\eta}^j$, $C_{P_{\leq \mathbf{c}(\boldsymbol{\eta}^i)}}$ is not a tope for

$C_P$; this is the case for example if $P$ is not a polytope but $P_{\leq \eta^i}$ is, implying that $\mathbf{c}(\eta^i) = \mathbf{0}$ and that $C_{P_{\leq \mathbf{c}(\eta^i)}} = \{\mathbf{0}\}$ is not a tope for $C_P \neq \{\mathbf{0}\}$).

For the particular case of a flux polyhedron $FP$ (25), we have $FP_{\leq \eta} = FC_{\leq \eta} \cap P_{\mathbf{ILC}}$ and $C_{FP_{\leq \eta}} = C_{FP_{\leq \eta}} = FC_{\leq \eta} \cap C_{P_{\mathbf{ILC}}}$, for any sign vector $\eta$. Any flux tope $FP_{\leq \eta^i}$ for $FP$ can thus be expressed as $FP_{\leq \eta^i} = FC_{\leq \eta^k} \cap P_{\mathbf{ILC}}$, for a certain flux tope $FC_{\leq \eta^k}$ for FC, i.e., by applying the constraints **ILC** to a flux tope for $FC$. The same holds for the recession cone: $C_{FP_{\leq \eta^i}} = C_{FP_{\leq \mathbf{c}(\eta^i)}} = FC_{\leq \eta^k} \cap C_{P_{\mathbf{ILC}}}$, i.e., by applying the homogeneous counterparts of constraints **ILC** to a flux tope for $FC$. If $FP$ is assumed to be consistent (same definition as for a flux cone, i.e., without unused reaction, always with a zero flux), all FTs $FP_{\leq \eta^i}$ then have full support, i.e., the $\eta^i$'s have full support or, equivalently, the $O_i$'s are $r$-orthants. An obvious upper bound for the number of FTs is thus $2^{r-r_I}$. Note that $C_{FP}$ to be consistent is a sufficient (but not necessary) condition for $FP$ to be consistent, and that, if $FP$ is consistent, so is $FC$ (but $FP$ can be inconsistent even if both $FC$ and $P_{\mathbf{ILC}}$ are consistent).

The method used to include a flux cone into a single (positive) orthant in higher dimension, by splitting each reversible reaction $i$ into a forward $i^+$ and a backward $i^-$ irreversible reaction, applies as well to a flux polyhedron $FP$. Matrix $\mathbf{G}$ is extended into matrix $\widetilde{\mathbf{G}} \in \mathbb{R}^{l \times \tilde{r}}$ in the same way that $\mathbf{S}$ is extended into $\widetilde{\mathbf{S}} \in \mathbb{R}^{m \times \tilde{r}}$, where $\tilde{r} = 2r - r_I$ is the new number of reactions and all $\tilde{r}$ reactions are now irreversible, $\widetilde{\mathbf{Irr}} = \{1, \ldots, \tilde{r}\}$. $FP$ is in one-to-one correspondence with vectors $\mathbf{v}$ of $\widetilde{FP}$ such that $\mathbf{v}_{i^+}.\mathbf{v}_{i^-} = 0$ for $i$ reversible. In particular, the fluxes with $\mathbf{v}_{i^+}, \mathbf{v}_{i^-} > 0$ for a certain $i$, which are obtained as vertices or extreme vectors of $\widetilde{FP}$, are futile (involving a net rate in opposite directions in reactions $i^+$ and $i^-$) without biological reality and must be eliminated (they are not generally limited to futile cycles as in flux cones). Finally, the set of elementary fluxes EFs $(FP)$ is in one-to-one correspondence with (ExPs $(\widetilde{FP}) \cup$ ExVs $(\widetilde{FP})) \setminus \{$futile fluxes$\}$. As $\widetilde{FP}$ is included in the positive $\tilde{r}$-orthant, it has only one FT.

As for flux cones, there are two opposite ways of dealing with reversible reactions for computing EFs = EFPs $\cup$ EFVs of a flux polyhedron $FP$: either splitting each reversible reaction into two irreversible ones, reducing $FP$ to a single FT in higher dimension, or keeping the reversible reactions unchanged and decomposing $FP$ into FTs without increasing the dimension; all intermediate cases are possible. Whatever the solution adopted, EFs are obtained as the union of ExPs and ExVs for each flux tope for $FP$ and thus we will generally work in a given FT for $FP$, defined by a (full support if $FP$ is consistent) sign vector $\eta$, and the EFs of $FP$ which conform to $\eta$ are thus given by the ExPs and ExVs of this FT $FP_{\leq \eta}$.

### 1.7.3. Elementary Modes

For a general polyhedron $P$ (and even for a flux polyhedron $FP$), there is no direct relationship between support-minimal Equation (19) vectors and elementary elements as it was the case for flux cones $FC$. Nevertheless, from Equation (34), it follows that, for any support-minimal nonzero vector in $P$ (still called elementary mode EM), there is an elementary element with the same support. Thus, Equation (20) generalizes to the case of polyhedra and all minimal support patterns of vectors appear in the set of supports of elementary elements and are actually the minimal nonempty elements in this set for subset inclusion:

$$supp(EMs) = Min_{\subseteq}(\{supp(Es)\} \setminus \{\varnothing\}). \tag{37}$$

In particular, for a metabolic network, this means that any set of reactions involved in a minimal pathway (EFM) appears as the set of reactions involved in a certain elementary flux (EF).

In addition, for a general polyhedron $P$, the correspondence (30) between $P$ and the higher dimensional flux cone $FC_P$ maps the elementary points (resp., elementary vectors) of

*P* onto the elementary flux vectors, i.e., the elementary flux modes, of $FC_P$ with component $\xi = 1$ (resp., with component $\xi = 0$):

$$EPs(P) = \{\mathbf{x} \in \mathbb{R}^r \mid \begin{pmatrix} \mathbf{x} \\ 1 \\ \mathbf{y} \end{pmatrix} \in EFMs(FC_P)\}$$

$$EVs(P) = \{\mathbf{x} \in \mathbb{R}^r \mid \begin{pmatrix} \mathbf{x} \\ 0 \\ \mathbf{y} \end{pmatrix} \in EFMs(FC_P)\}$$

(38)

with $FC_P$ given by Equation (30). The same formula holds for the particular case of a flux polyhedron *FP* (25) with $FC_{FP}$ given by Equation (31).

Moreover, as seen in the proof of Equation (22), any vector which is support-minimal in a certain $P_{\leq \boldsymbol{\eta}}$ is actually support minimal in *P*, thus EMs of *P* can be computed tope by tope:

$$EMs(P) = \bigcup_{\boldsymbol{\eta}^i \ maximal \ in \ sign(P)} EMs(P_{\leq \boldsymbol{\eta}^i}). \tag{39}$$

See Figure 2 for an illustration of the concepts of extreme points and vectors, elementary points and vectors and elementary modes in a convex polyhedron.

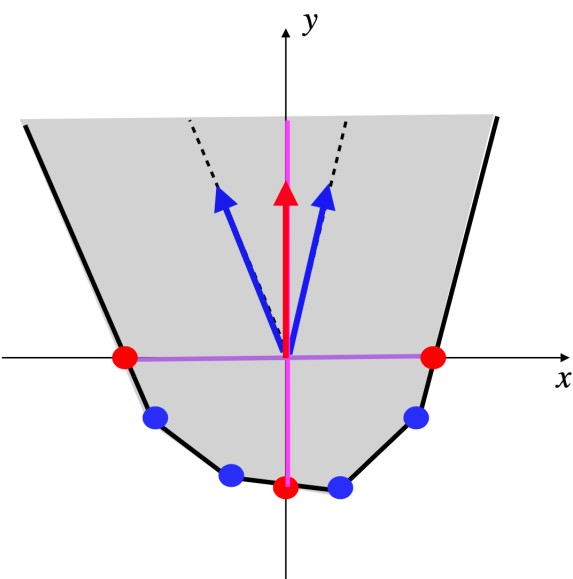

**Figure 2.** We consider the convex polyhedron in gray and its associated recession cone delimited by dashed lines. The extreme points are the vertices in blue and the extreme vectors are the blue vectors. The elementary points are all the blue or red points, and the elementary vectors are all the blue or red vectors, and they coincide, respectively, with the extreme points and extreme vectors for all topes (here we have four topes, corresponding to the subpolyhedra in the four quadrants defined by *x* and *y*). The elementary modes are the purple segment and half-line and they do not have a direct relationship with elementary elements, but their two possible supports, $\{x\}$ and $\{y\}$, are the minimal nonempty supports of the elementary elements.

*1.8. Complexity Results*

Enumerating the vertices of an unbounded polyhedron has been proven to be an NP-hard enumeration problem [20]. However, the hardness of vertex generation for bounded polyhedra, and also the complexity of enumerating together vertices and extreme rays of polyhedra, are open problems. Consequently, the complexity of enumerating all EFs or EFMs of a metabolic network remains an open problem [21]. Nevertheless, enumerating all extreme rays of a flux cone that contain a given reaction in their support has been proven not to be in polynomial total time unless P = NP [21], which means that the output

cannot be generated in time polynomial in the combined size of the input and the output. The problem of enumerating the extreme vectors of a polyhedral cone or the vertices of a polytope with a polynomial delay (i.e., the time between one output item and the next is bounded by a polynomial function in terms of the input size), or the even weaker question whether this enumeration can be done in polynomial total time, is an open problem too. Therefore, the possibility of enumerating EFs or EFMs of a metabolic network with a polynomial delay is an open problem [22] (except for a flux cone with all reactions being reversible, i.e., a flux linear subspace (3), as the EFMs are then the circuits of a linear matroid [16]). Given a flux cone and two reactions, deciding if there exists an extreme ray of the cone that has both reactions in its support is NP-complete [21]. Thus, given a metabolic network, deciding if an EF or EFM exists with a given support of size at least two is an NP-complete problem. The same result holds for deciding if an EF or EFM exists whose support size is bounded above by a given positive integer [22].

Regarding now the number of EFs or EFMs, counting the extreme rays of a polyhedral cone or the vertices of a polytope has been proven to be a #P-complete problem [23]. Thus, counting the number of EFs or EFMs of a metabolic network is also a #P-complete problem [22]. The McMullen's upper bound theorem [24] states that, for any fixed positive integers $d$ and $n$, the maximum number of $j$-faces of a $d$-polytope with $n$ facets (i.e., faces of dimension $d - 1$) is attained by the dual cyclic polytope $c^\star(d, n)$ for all $j = 0, 1, \ldots, d - 2$. A consequence is that the maximum number of vertices of a $d$-polytope with $n$ facets is given by $\binom{n - \lceil d/2 \rceil}{n - d} + \binom{n - \lfloor d/2 \rfloor - 1}{n - d} \sim 2\binom{n - \lfloor d/2 \rfloor}{n - d}$. We thus obtain that the number of EFs or EFMs in a metabolic network (after having split each reversible reaction into two irreversible ones) is bounded above by a quantity approximately equal to $2\binom{\lfloor (\tilde{r} + m)/2 \rfloor}{m}$ with $\tilde{r} = 2r - r_I$ (so $\tilde{r}$ varies between $r$ and $2r$). If the number $m$ of internal metabolites is small compared to the total number $\tilde{r}$ of reactions (after having split reversible reactions), the number of EFs or EFMs is then bounded above by a quantity of order $\Theta((\tilde{r}/2)^m)$. If $m$ is close to $\tilde{r}$, this number is bounded above by a quantity of order $\Theta(\tilde{r}^{(\tilde{r}-m)/2})$. The worst case occurs when $m$ is close to $\tilde{r}/3$ with an upper bound approximately equal to $2\binom{2m}{m}$.

We obtain the same results with $r$ instead of $\tilde{r}$ if we do not split reversible reactions but fix their signs arbitrarily, i.e., if we consider EFs or EFMs in an arbitrary closed $r$-orthant $O$, i.e., in an arbitrary flux tope for $FC$ (or $FP$). However, note that in practice the actual number of EFs or EFMs of a metabolic network is likely to be much smaller than these upper bounds.

## 2. Metabolic Pathways in the Presence of Biological Constraints

Although the current improved implementations of the DD method [25,26] allow the computation of millions, even billions, of EFMs or EFs, tackling genome-scale metabolic models (GSMMs) is still beyond our reach. Moreover, most of the computed EFMs or EFs are not biologically valid, because only stoichiometry and certain flux constraints, such as irreversibility of reactions or bounds on reaction rates, are taken into account. For both scaling up to large systems and limiting the number of biologically invalid solutions, it is necessary to consider additional biological constraints, such as thermodynamic, kinetic or regulatory constraints.

### 2.1. Biological Constraints

#### 2.1.1. Thermodynamic Constraints

Assuming constant pressure and a closed system, according to the second law of thermodynamics, a reaction $i$ proceeds spontaneously only in the direction of its negative Gibbs free energy $\Delta_r G_i$ [27], given by

$$\Delta_r G_i = \Delta_r G_i'^0 + RT \ln(\prod_j \mathbf{M}_j^{\overline{\mathbf{S}}_{ji}}) \tag{40}$$

where $\Delta_r G_i^{\prime 0}$ is the standard Gibbs free energy of reaction $i$, $R$ the molar gas constant, $T$ the absolute temperature, $\mathbf{M}_j$ the (positive) concentration of metabolite $j$ and $\overline{\mathbf{S}}_{ji}$ is the stoichiometric coefficient of metabolite $j$ in reaction $i$ (i.e., $\overline{\mathbf{S}} \in \mathbb{R}^{(m+\overline{m}) \times r}$ is the extension of the stoichiometric matrix $\mathbf{S}$ to the $\overline{m}$ external metabolites). This means that $\Delta_r G_i < 0$, (resp., $\Delta_r G_i > 0$) is a necessary condition to get $\mathbf{v}_i > 0$ (resp., $\mathbf{v}_i < 0$), which can be expressed by the constraint $sign(\mathbf{v}_i) \leq -sign(\Delta_r G_i)$, and that a flux vector $\mathbf{v}$ is thermodynamically feasible [28] if and only if all its components $\mathbf{v}_i$ satisfy such a constraint (it is enough to consider those $i \in supp(\mathbf{v})$ as the constraint is trivially satisfied when $\mathbf{v}_i = 0$). The thermodynamic constraint for $\mathbf{v}$, that depends on the vector $\mathbf{M}$ of metabolite concentrations, can thus be defined as

$$\mathbf{TC_M}(\mathbf{v}) \triangleq \forall i \in supp(\mathbf{v}), sign(\mathbf{v}_i) = -sign(\Delta_r G_i^{\prime 0} + RT \ln(\prod_j \mathbf{M}_j^{\overline{\mathbf{S}}_{ji}})). \qquad (41)$$

As at equilibrium $\Delta_r G_i$ is null, we obtain $\Delta_r G_i^{\prime 0} = -RT \ln(K_{eq}^i)$, where $K_{eq}^i = \prod_j M_{j_{eq}}^{\overline{\mathbf{S}}_{ji}}$ is the equilibrium constant of reaction $i$. Thus, $\Delta_r G_i$ can be rewritten as $\Delta_r G_i = RT \ln(\prod_j \mathbf{M}_j^{\overline{\mathbf{S}}_{ji}} / K_{eq}^i)$ and the thermodynamic constraint $\mathbf{TC_M}(\mathbf{v})$ as

$$\mathbf{TC_M}(\mathbf{v}) \triangleq \forall i \in supp(\mathbf{v}), sign(\mathbf{v}_i) = -sign(\prod_j \mathbf{M}_j^{\overline{\mathbf{S}}_{ji}} - K_{eq}^i). \qquad (42)$$

Often, the concentrations of external metabolites can be measured and included in the constraint as known parameters, keeping only a dependency of the constraint on the concentrations of internal metabolites. The formula $\Delta_r G_i = RT \ln(\prod_j \mathbf{M}_j^{\overline{\mathbf{S}}_{ji}} / K_{eq}^i)$ can be rewritten, by dividing the numerator and denominator of the fraction by the terms dealing with external metabolites, as $\Delta_r G_i = RT \ln(\prod_j \underline{\mathbf{M}}_j^{\mathbf{S}_{ji}} / \hat{K}_{eq}^i)$, where $\hat{K}_{eq}^i = K_{eq}^i / \prod_j \overline{\mathbf{M}}_j^{\overline{\mathbf{S}}_{ji}}$ is the apparent equilibrium constant of the reaction $i$ and $\underline{\mathbf{M}}$ (resp., $\overline{\mathbf{M}}$) the vector of internal (resp., external) metabolite concentrations. The thermodynamic constraint (42) can thus be rewritten as

$$\underline{\mathbf{TC_M}}(\mathbf{v}) \triangleq \forall i \in supp(\mathbf{v}), sign(\mathbf{v}_i) = -sign(\prod_j \underline{\mathbf{M}}_j^{\mathbf{S}_{ji}} - \hat{K}_{eq}^i). \qquad (43)$$

For given metabolite concentrations vector $\mathbf{M}$ (resp., internal metabolite concentrations vector $\underline{\mathbf{M}}$), let $\mathbf{ts_M} \in \{-, 0, +\}^r$ (resp., $\mathbf{ts_{\underline{M}}}$) be the fixed thermodynamic sign vector defined by:

$$\begin{aligned} (\mathbf{ts_M})_i &= -sign(\Delta_r G_i^{\prime 0} + RT \ln(\prod_j \mathbf{M}_j^{\overline{\mathbf{S}}_{ji}})) = -sign(\prod_j \mathbf{M}_j^{\overline{\mathbf{S}}_{ji}} - K_{eq}^i) \\ (\mathbf{ts_{\underline{M}}})_i &= -sign(\prod_j \underline{\mathbf{M}}_j^{\mathbf{S}_{ji}} - \hat{K}_{eq}^i) \qquad \text{for } 1 \leq i \leq r. \end{aligned} \qquad (44)$$

Then, the thermodynamic constraint $\mathbf{TC_M}$ (resp., $\underline{\mathbf{TC_M}}$) can be rewritten as

$$\mathbf{TC_M}(\mathbf{v}) \triangleq sign(\mathbf{v}) \leq \mathbf{ts_M} \qquad \underline{\mathbf{TC_M}}(\mathbf{v}) \triangleq sign(\mathbf{v}) \leq \mathbf{ts_{\underline{M}}}. \qquad (45)$$

Thus, the set $Sol_{\mathbf{TC_M}}$ (resp., $Sol_{\underline{\mathbf{TC_M}}}$) of vectors $\mathbf{v}$ satisfying the constraint $\mathbf{TC_M}(\mathbf{v})$ (resp., $\underline{\mathbf{TC_M}}(\mathbf{v})$), given by $\{\mathbf{v} \in \mathbb{R}^r \mid sign(\mathbf{v}) \leq \mathbf{ts_M}\}$ (resp., $\{\mathbf{v} \in \mathbb{R}^r \mid sign(\mathbf{v}) \leq \mathbf{ts_{\underline{M}}}\}$), is the closed orthant $O_{\mathbf{ts_M}}$ (resp., $O_{\mathbf{ts_{\underline{M}}}}$) defined by $\mathbf{ts_M}$ (resp., $\mathbf{ts_{\underline{M}}}$), of dimension $r$ if $\mathbf{ts_M}$ (resp., $\mathbf{ts_{\underline{M}}}$) has full support, i.e., $(\mathbf{ts_M})_i \neq 0$ (resp., $(\mathbf{ts_{\underline{M}}})_i \neq 0$) for all $i$, of lesser dimension otherwise.

**Lemma 1.** *Given metabolite concentrations $M$ (resp., internal metabolite concentrations $\underline{M}$), the set $Sol_{\mathbf{TC_M}}$ (resp., $Sol_{\underline{\mathbf{TC_M}}}$) of vectors in $\mathbb{R}^r$ satisfying the thermodynamic constraint $\mathbf{TC_M}$ (resp., $\underline{\mathbf{TC_M}}$) is the set of vectors that conform to the thermodynamic sign vector $ts_M$ (resp., $ts_{\underline{M}}$) Equation (44), i.e., the closed orthant $O_{ts_M}$ (resp., $O_{ts_{\underline{M}}}$) defined by $ts_M$ (resp., $ts_{\underline{M}}$).*

### 2.1.2. Kinetic Constraints

Metabolic reactions are catalyzed by enzymes. The catalytic mechanisms of key enzymes have been investigated in great detail and described by mathematical formulas. However, many kinetic equations are still unknown and have to be substituted by standard rate laws such as mass-action kinetics, power laws, reversible Hill kinetics, lin-log kinetics, convenience kinetics, generic rate equations or TKM rate laws [29]. What is important for our study is that these modular rate laws share the general form

$$\mathbf{v}_i = \mathbf{E}_i \kappa_i(\mathbf{M}) \quad \text{with} \quad \kappa_i = f_i^{reg} \frac{T_i}{D_i + D_i^{reg}} \tag{46}$$

where $\mathbf{E}_i$ is the (non-negative) level of the enzyme catalyzing the reaction $i$ (given either as an amount or as a concentration, in which case the rate law is pre-multiplied by the compartment volume) and $\kappa_i$ depends on the concentrations of the metabolites occurring in $i$ (reactants of $i$) and on reaction $i$ stoichiometry, rate law considered, allosteric regulation and parameters. In the general form of $\kappa_i$, $T_i$ is the thermodynamic numerator (which can be written in the compact form $k_i^+ \theta_i^+ - k_i^- \theta_i^-$ with turnover rate parameters $k_i^{\pm}$) that gives its sign to $\kappa_i$ (and thus to $\mathbf{v}_i$) and reflects the relationship between chemical potentials and reaction directions and ensures that the rate vanishes at chemical equilibrium, $f_i^{reg}$ and $D_i^{reg}$, both positive, implement enzyme regulation (partial or complete for the first one, specific for the second) and $D_i$ is the (positive) kinetic denominator, a polynomial of scaled reactant concentrations whose terms correspond to different binding states of the enzyme (reducing the enzyme amount available for catalysis), which depends on the rate law considered.

The kinetic constraint for $\mathbf{v}$ depends both on the vector $\mathbf{E}$ of enzyme concentrations and on the vector $\mathbf{M}$ of metabolite concentrations and can thus be defined as

$$\mathbf{KC_{E,M}}(\mathbf{v}) \overset{\triangle}{=} \mathbf{v} = \mathbf{E} \circ \boldsymbol{\kappa}(\mathbf{M}) \tag{47}$$

where $\circ$ is the component-wise product of vectors: $(\mathbf{E} \circ \boldsymbol{\kappa})_i = \mathbf{E}_i \kappa_i$. This means that the flux vector is a component-wise linear function of the vector of enzyme concentrations (and a nonlinear function of the metabolite concentrations vector). For nonzero $\mathbf{E}_i$, the sign of $T_i$ gives the direction in which reaction $i$ proceeds, so in this sense the kinetic constraint includes the thermodynamic constraint.

This can be highlighted on a widely-used rate law, the reversible Michaelis–Menten kinetics [30]. In the simple case of a reaction $i$ where the enzyme can only exist in one of three distinct states: free, all substrates bound, or all products bound, it can be written as

$$\kappa_i = \frac{k_i^+ \prod_{j|\overline{\mathbf{S}}_{ji}<0}(\mathbf{M}_j/K_{ij}^M)^{-\overline{\mathbf{S}}_{ji}} - k_i^- \prod_{j|\overline{\mathbf{S}}_{ji}>0}(\mathbf{M}_j/K_{ij}^M)^{\overline{\mathbf{S}}_{ji}}}{1 + \prod_{j|\overline{\mathbf{S}}_{ji}<0}(\mathbf{M}_j/K_{ij}^M)^{-\overline{\mathbf{S}}_{ji}} + \prod_{j|\overline{\mathbf{S}}_{ji}>0}(\mathbf{M}_j/K_{ij}^M)^{\overline{\mathbf{S}}_{ji}}} \tag{48}$$

where $k_i^+$ and $k_i^-$ are the maximal forward and backward rates in reaction $i$ per unit of enzyme and the $K_{ij}^M$'s are the Michaelis constants. Equating the numerator to zero at equilibrium, we obtain $\frac{k_i^+}{k_i^-} \prod_j (K_{ij}^M)^{\overline{\mathbf{S}}_{ji}} = K_{eq}^i$. This gives

$$\kappa_i = k_i^+ \left( \frac{\prod_{j|\overline{\mathbf{S}}_{ji}<0}(\mathbf{M}_j/K_{ij}^M)^{-\overline{\mathbf{S}}_{ji}}}{1 + \prod_{j|\overline{\mathbf{S}}_{ji}<0}(\mathbf{M}_j/K_{ij}^M)^{-\overline{\mathbf{S}}_{ji}} + \prod_{j|\overline{\mathbf{S}}_{ji}>0}(\mathbf{M}_j/K_{ij}^M)^{\overline{\mathbf{S}}_{ji}}} \right) \left( 1 - \frac{\prod_j \mathbf{M}_j^{\overline{\mathbf{S}}_{ji}}}{K_{eq}^i} \right) \tag{49}$$

that is, the product of three terms: the positive capacity term per unit of enzyme, the positive (smaller than one) fractional saturation term depending on $\mathbf{M}$ and the thermodynamic term, which can be rewritten as $1 - e^{\Delta_r G_i / RT}$ and gives the sign of $\kappa_i$ (and thus the sign of

$\mathbf{v}_i$): $\kappa_i > 0 \Leftrightarrow \Delta_r G_i < 0$, i.e., the thermodynamic constraint. Thus, $sign(\kappa(\mathbf{M})) = \mathbf{ts_M}$ and we will assume that this equality holds for all kinetic laws we consider.

Note that, for given metabolite and enzyme concentrations $\mathbf{M}$ and $\mathbf{E}$, the kinetic constraint $\mathbf{KC_{E,M}}$ defines completely and uniquely the only vector that satisfies it: $Sol_{\mathbf{KC_{E,M}}} = \{\mathbf{E} \circ \kappa(\mathbf{M})\}$.

### 2.1.3. Regulatory Constraints

Coupling metabolic networks with Boolean transcriptional regulatory networks allows us to express the additional constraints imposed by gene regulatory information on a metabolic network and to take them into account when computing EFMs [31–33]. In all generality, such a constraint may be given by an arbitrary Boolean formula in terms of the reactions $i$, viewed as propositional symbols, i.e., the positive literal $i$ meaning that reaction $i$ is active (nonzero flux) and the negative literal $\neg i$ meaning it is inactive (zero flux). Thus, when applying this Boolean constraint, noted $Bc$, to a flux vector $\mathbf{v}$, the positive literal $i$ is interpreted as $\mathbf{v}_i \neq 0$, i.e., $i \in supp(\mathbf{v})$ (4), and the negative literal $\neg i$ as $\mathbf{v}_i = 0$, i.e., $i \notin supp(\mathbf{v})$.

The regulatory constraint $\mathbf{RC}_{Bc}$ for $\mathbf{v}$ induced by $Bc$ can thus be defined as

$$\mathbf{RC}_{Bc}(\mathbf{v}) \overset{\triangle}{=} Bc(\mathbf{v}_i). \tag{50}$$

Note that coupled reactions as used by Flux Coupling Analysis (FCA) [34–36] can be easily represented by such constraints. For example, $i$ directionally coupled to $j$, meaning that zero flux through $i$ implies zero flux through $j$, is expressed by $Bc = i \vee \neg j$, and $i$ partially coupled to $j$, meaning that zero flux through $i$ is equivalent to zero flux through $j$, is expressed by $Bc = (i \wedge j) \vee (\neg i \wedge \neg j)$.

By rewriting the Boolean constraint $Bc$ in DNF (Disjunctive Normal Form), $Bc = \bigvee_k D_k$, the set $Sol_{\mathbf{RC}_{Bc}} = \{\mathbf{v} \in \mathbb{R}^r \mid \mathbf{RC}_{Bc}(\mathbf{v})\}$ of vectors $\mathbf{v}$ satisfying the constraint $\mathbf{RC}_{Bc}$ is a union of the solution spaces for each disjunct $D_k$ (and this union can be assumed to be disjoint by taking the disjuncts $D_k$ two by two inconsistent). Now a disjunct is a conjunction of literals, where a negative literal $\neg i$ corresponds to the constraint $\mathbf{v}_i = 0$ and a positive literal $i$ to the constraint $\mathbf{v}_i \neq 0$, which can be rewritten as the disjunctive constraint $(\mathbf{v}_i < 0) \vee (\mathbf{v}_i > 0)$. Finally, a propositional symbol $i$ that does not appear in the disjunct corresponds to an absence of constraint on $\mathbf{v}_i$, which can be rewritten as the disjunction $(\mathbf{v}_i < 0) \vee (\mathbf{v}_i = 0) \vee (\mathbf{v}_i > 0)$. Thus, the solution space for a disjunct is itself the disjoint union of subspaces each one defined by constraints of type $\mathbf{v}_i \, op_i \, 0$ for all $i$, with $op_i \in \{<, =, >\}$, that is, defined by $\{\mathbf{v} \in \mathbb{R}^r \mid sign(\mathbf{v}) = \mathbf{rs}\}$ for a given sign vector $\mathbf{rs} \in \{-, 0, +\}^r$, i.e., an open orthant $\mathring{O}_{\mathbf{rs}}$ (which, for $\mathbf{rs} \neq \mathbf{0}$, is topologically open in the vector subspace it spans and is the interior in this subspace of the closed orthant $O_{\mathbf{rs}} = \{\mathbf{v} \in \mathbb{R}^r \mid sign(\mathbf{v}) \leq \mathbf{rs}\}$, which is the closure of $\mathring{O}_{\mathbf{rs}}$ in $\mathbb{R}^r$). In summary, $Sol_{\mathbf{RC}_{Bc}}$ is thus the disjoint union of such open orthants. Now, instead of keeping this partition of $Sol_{\mathbf{RC}_{Bc}}$ in open orthants, it can be more practical to generalize this concept and deal with what we will call semi-open orthants $O^\circ$, i.e., orthants $O$ without some of their faces of lesser dimension (open orthant is thus a particular case of semi-open orthant, without any facet thus without any face of lesser dimension). To do this, we can group together with any given open orthant $\mathring{O}_{\mathbf{rs}_j}$ in $Sol_{\mathbf{RC}_{Bc}}$, in a same cluster $\mathcal{C}$, all other open orthants $\mathring{O}_{\mathbf{rs}_k}$ in $Sol_{\mathbf{RC}_{Bc}}$ such that $O_{\mathbf{rs}_k} \subset O_{\mathbf{rs}_j}$, i.e., $O_{\mathbf{rs}_k}$ is a face of $O_{\mathbf{rs}_j}$, and, for any arbitrary orthant $O_{\mathbf{rs}_l}$ with $O_{\mathbf{rs}_k} \subset O_{\mathbf{rs}_l} \subset O_{\mathbf{rs}_j}$ (which is equivalent to $\mathbf{rs}_k < \mathbf{rs}_l < \mathbf{rs}_j$), then $\mathring{O}_{\mathbf{rs}_l} \in \mathcal{C}$. In this case $\bigcup_{\mathring{O} \in \mathcal{C}} \mathring{O}$ is a semi-open orthant $O^\circ$. We can iteratively process like this by considering each time the open orthants in $Sol_{\mathbf{RC}_{Bc}}$ that have not yet been selected in any cluster, which guarantees that the semi-open orthants built in this way are disjoint. In addition, by choosing as open orthant, $\mathring{O}_{\mathbf{rs}_j}$, to start with at each iteration, a maximal one among those that remain, we are sure that no two of the semi-open orthants thus built can be grouped together to constitute a bigger semi-open orthant, i.e., the collection obtained is minimal in this sense. We finally obtain that $Sol_{\mathbf{RC}_{Bc}}$ can be written as a disjoint union of semi-open orthants, with no merging possible between any two of them. However, note that such a

decomposition is not unique and that an inclusion can still exist between the closures of two such semi-open orthants. If we consider the particular case of a Boolean constraint which is an arbitrary disjunct $D$, then for any closed $r$-orthant $O_\eta$, i.e., for any full support sign vector $\eta$, $Sol_{RC_D \leq \eta}$ is a semi-open orthant, which is the face of $O_\eta$ defined by $\mathbf{v}_i = 0$ for all $i$ such that $\neg i$ is a literal of $D$, without its facets of equation $\mathbf{v}_i = 0$ for all $i$ such that $i$ is a literal of $D$.

**Lemma 2.** *The set $Sol_{RC_{Bc}}$ of vectors in $\mathbb{R}^r$ satisfying the regulatory constraint $\mathbf{RC}_{Bc}$ is a disjoint union of open orthants $\mathring{O}$. These can be grouped together such that $Sol_{RC_{Bc}}$ is a disjoint union of semi-open orthants $O^\circ$, i.e., orthants $O$ without some of their faces, such that no two of them can be grouped together to constitute a bigger semi-open orthant. If $Bc$ is a conjunction of literals and $\eta$ an arbitrary full support sign vector, then $Sol_{RC_{Bc} \leq \eta}$ is itself a semi-open orthant.*

### 2.2. Characterizing the Solution Space

We start from a flux cone (6) (or more generally a flux polyhedron (25)) representing the flux vectors of a metabolic network in a steady state, satisfying the stoichiometric equations, the inequalities regarding irreversibility of reactions (and possibly some inhomogeneous linear constraints). Then, we consider additional biological constraints, such as those described in Section 2.1. In all generality, these constraints will be noted $\mathbf{C_x}(\mathbf{v})$, where $\mathbf{v}$ represents a flux vector in $\mathbb{R}^r$ and $\mathbf{x}$ a vector of biochemical quantities involved in the constraint (typically, metabolite concentrations and enzyme concentrations). Some parameters (such as stoichiometric coefficients), that are not made explicit in the notation for sake of simplicity, are also present. In the following, the solution space in $\mathbb{R}^r$ of an arbitrary constraint $\mathbf{C}$ will be noted $Sol_\mathbf{C}$, defined as $Sol_\mathbf{C} = \{\mathbf{v} \in \mathbb{R}^r \mid \mathbf{C}(\mathbf{v})\}$. For given values of quantities $\mathbf{x}$, the solution space is thus the constrained flux cone subset (not necessarily a cone, depending on $\mathbf{C_x}$):

$$Sol_{\mathbf{S,Irr,C}}(\mathbf{x}) \triangleq CFC_\mathbf{C}(\mathbf{x}) \triangleq \{\mathbf{v} \in \mathbb{R}^r \mid \mathbf{Sv} = \mathbf{0}, \mathbf{v}_i \geq 0 \text{ for } i \in \mathbf{Irr}, \mathbf{C_x}(\mathbf{v})\}$$
$$= Sol_{\mathbf{S,Irr}} \cap Sol_{\mathbf{C_x}} = FC \cap Sol_{\mathbf{C_x}} \tag{51}$$

or, more generally, the constrained flux polyhedron subset (not necessarily a polyhedron, depending on $\mathbf{C_x}$):

$$Sol_{\mathbf{S,Irr,ILC,C}}(\mathbf{x}) \triangleq CFP_\mathbf{C}(\mathbf{x})$$
$$\triangleq \{\mathbf{v} \in \mathbb{R}^r \mid \mathbf{Sv} = \mathbf{0}, \mathbf{v}_i \geq 0 \text{ for } i \in \mathbf{Irr}, \mathbf{Gv} \geq \mathbf{h}, \mathbf{C_x}(\mathbf{v})\} \tag{52}$$
$$= Sol_{\mathbf{S,Irr,ILC}} \cap Sol_{\mathbf{C_x}} = FP \cap Sol_{\mathbf{C_x}} = CFC_\mathbf{C}(\mathbf{x}) \cap P_{\mathbf{ILC}}.$$

Now, very often, most of the values of quantities $\mathbf{x}$, if not all, are unknown. We then only require consistency of the set of constraints $\mathbf{C_x}$, i.e., the existence of values for variables $\mathbf{x}$ such that the constraints are satisfied. This means that we replace the constraint $\mathbf{C_x}$ by the constraint $\exists \mathbf{x C_x}$ whose solution space is $Sol_{\exists \mathbf{x C_x}} = \{\mathbf{v} \in \mathbb{R}^r \mid \{\mathbf{x} \mid \mathbf{C_x}(\mathbf{v})\} \neq \varnothing\} = \bigcup_\mathbf{x} Sol_{\mathbf{C_x}}$. The solution space becomes

$$Sol_{\mathbf{S,Irr,C}} \triangleq CFC_\mathbf{C} \triangleq \{\mathbf{v} \in \mathbb{R}^r \mid \mathbf{Sv} = \mathbf{0}, \mathbf{v}_i \geq 0 \text{ for } i \in \mathbf{Irr}, \exists \mathbf{x C_x}(\mathbf{v})\}$$
$$= Sol_{\mathbf{S,Irr}} \cap Sol_{\exists \mathbf{x C_x}} = \bigcup_\mathbf{x} Sol_{\mathbf{S,Irr,C}}(\mathbf{x}) = FC \cap \bigcup_\mathbf{x} Sol_{\mathbf{C_x}} \tag{53}$$

or

$$Sol_{\mathbf{S,Irr,ILC,C}} \triangleq CFP_\mathbf{C} \triangleq \{\mathbf{v} \in \mathbb{R}^r \mid \mathbf{Sv} = \mathbf{0}, \mathbf{v}_i \geq 0 \text{ for } i \in \mathbf{Irr}, \mathbf{Gv} \geq \mathbf{h}, \exists \mathbf{x C_x}(\mathbf{v})\}$$
$$= Sol_{\mathbf{S,Irr,ILC}} \cap Sol_{\exists \mathbf{x C_x}} = \bigcup_\mathbf{x} Sol_{\mathbf{S,Irr,ILC,C}}(\mathbf{x}) = FP \cap \bigcup_\mathbf{x} Sol_{\mathbf{C_x}} = CFC_\mathbf{C} \cap P_{\mathbf{ILC}}. \tag{54}$$

If some but not all of the values of quantities $\mathbf{x}$ are known, $\exists \mathbf{x}$ concerns only those $\mathbf{x}$ whose value is unknown, the known values being integrated in $\mathbf{C}$ as parameters to simplify the notations.

We will obtain examples of such constraints $\exists \mathbf{x} \mathbf{C}_{\mathbf{x}}(\mathbf{v})$ from the cases of thermodynamic constraint **TC** and kinetic constraint **KC**.

Depending on the constraint considered, the solution space is in general no longer a cone or polyhedron and no longer convex. Nevertheless, the definitions of extreme vectors and extreme points, of elementary flux vectors and elementary flux points, and of elementary flux modes are still valid, as, respectively, non-decomposable and convex non-decomposable vectors, conformally non-decomposable and convex-conformally non-decomposable vectors, and support-minimal vectors, where decomposition and support minimality have to be understood w.r.t. vectors of the solution space. It is then clear that if a vector of $FC$ (resp., $FP$) satisfies one of those properties in $FC$ (resp., $FP$) and satisfies the given constraint, then it satisfies the same property in the solution space, i.e.,

$$
\begin{aligned}
ExVs(FC) \cap Sol_{\mathbf{C_x}} &\subseteq ExVs(CFC_{\mathbf{C}}(\mathbf{x})) & ExVs(FC) \cap Sol_{\exists \mathbf{x} \mathbf{C_x}} &\subseteq ExVs(CFC_{\mathbf{C}}) \\
ExVs(C_{FP}) \cap Sol_{\mathbf{C_x}} &\subseteq ExVs(CFP_{\mathbf{C}}(\mathbf{x})) & ExVs(C_{FP}) \cap Sol_{\exists \mathbf{x} \mathbf{C_x}} &\subseteq ExVs(CFP_{\mathbf{C}}) \\
ExPs(FP) \cap Sol_{\mathbf{C_x}} &\subseteq ExPs(CFP_{\mathbf{C}}(\mathbf{x})) & ExPs(FP) \cap Sol_{\exists \mathbf{x} \mathbf{C_x}} &\subseteq ExPs(CFP_{\mathbf{C}}) \\
EFVs(FC) \cap Sol_{\mathbf{C_x}} &\subseteq EFVs(CFC_{\mathbf{C}}(\mathbf{x})) & EFVs(FC) \cap Sol_{\exists \mathbf{x} \mathbf{C_x}} &\subseteq EFVs(CFC_{\mathbf{C}}) \\
EFVs(C_{FP}) \cap Sol_{\mathbf{C_x}} &\subseteq EFVs(CFP_{\mathbf{C}}(\mathbf{x})) & EFVs(C_{FP}) \cap Sol_{\exists \mathbf{x} \mathbf{C_x}} &\subseteq EFVs(CFP_{\mathbf{C}}) \\
EFPs(FP) \cap Sol_{\mathbf{C_x}} &\subseteq EFPs(CFP_{\mathbf{C}}(\mathbf{x})) & EFPs(FP) \cap Sol_{\exists \mathbf{x} \mathbf{C_x}} &\subseteq EFPs(CFP_{\mathbf{C}}) \\
EFMs(FC) \cap Sol_{\mathbf{C_x}} &\subseteq EFMs(CFC_{\mathbf{C}}(\mathbf{x})) & EFMs(FC) \cap Sol_{\exists \mathbf{x} \mathbf{C_x}} &\subseteq EFMs(CFC_{\mathbf{C}}) \\
EFMs(FP) \cap Sol_{\mathbf{C_x}} &\subseteq EFMs(CFP_{\mathbf{C}}(\mathbf{x})) & EFMs(FP) \cap Sol_{\exists \mathbf{x} \mathbf{C_x}} &\subseteq EFMs(CFP_{\mathbf{C}})
\end{aligned}
\tag{55}
$$

For EFMs, the above formulas are valid whatever the structure of $Sol_{\mathbf{C}}$. It is also the case for ExPs and EFPs but, in practice, only really meaningful if $CFP_{\mathbf{C}}(\mathbf{x})$ (resp., $CFP_{\mathbf{C}}$) is a convex polyhedron. Finally, for ExVs and EFVs, this is only meaningful if $CFC_{\mathbf{C}}(\mathbf{x})$ (resp., $CFC_{\mathbf{C}}$) is a convex polyhedral cone and $CFP_{\mathbf{C}}(\mathbf{x})$ (resp., $CFP_{\mathbf{C}}$) is a convex polyhedron. This is the case when $Sol_{\mathbf{C_x}}$ (resp., $Sol_{\exists \mathbf{x} \mathbf{C_x}}$) is itself a convex polyhedral cone and we will see that, for almost all common biological constraints, this solution space is actually a union of such cones or of semi-open cones and thus the formulas will apply for each conical component. In this case we have $C_{CFP_{\mathbf{C}}} = CFC_{\mathbf{C}} \cap C_{P_{\text{ILC}}} = C_{FP} \cap Sol_{\mathbf{C}}$. This can be generalized when $Sol_{\mathbf{C}}$ is a convex polyhedron, and thus $CFC_{\mathbf{C}}$ and $CFP_{\mathbf{C}}$ too, with $C_{CFC_{\mathbf{C}}} = FC \cap C_{Sol_{\mathbf{C}}}$ and $C_{CFP_{\mathbf{C}}} = C_{CFC_{\mathbf{C}}} \cap C_{P_{\text{ILC}}} = C_{FP} \cap C_{Sol_{\mathbf{C}}}$, by replacing $Sol_{\mathbf{C}}$ by $C_{Sol_{\mathbf{C}}}$ in the formulas above regarding ExVs and EFVs.

However, in all the above cases, we generally do not have the reciprocal subset inclusion. This is what we will study for particular constraints.

### 2.2.1. Application to Thermodynamics

Assume first that the concentrations of the metabolites (resp., of the internal metabolites) are known and given. From Equations (51) and (52) and Lemma 1, we obtain

$$
CFC_{\mathbf{TC}}(\mathbf{M}) = FC_{\leq \mathbf{ts_M}} \qquad CFC_{\underline{\mathbf{TC}}}(\underline{\mathbf{M}}) = FC_{\leq \mathbf{ts_{\underline{M}}}}
\tag{56}
$$

$$
CFP_{\mathbf{TC}}(\mathbf{M}) = FP_{\leq \mathbf{ts_M}} \qquad CFP_{\underline{\mathbf{TC}}}(\underline{\mathbf{M}}) = FP_{\leq \mathbf{ts_{\underline{M}}}}.
\tag{57}
$$

$CFC_{\mathbf{TC}}(\mathbf{M})$, $CFC_{\underline{\mathbf{TC}}}(\underline{\mathbf{M}})$ are thus flux cones and $CFP_{\mathbf{TC}}(\mathbf{M})$, $CFP_{\underline{\mathbf{TC}}}(\underline{\mathbf{M}})$ flux polyhedra (possibly equal to $\{\mathbf{0}\}$ or empty (for polyhedra) if the flux directions imposed by the concentrations of the metabolites and by the second law of thermodynamics are incompatible with the steady-state assumption). When nonempty, they consist of a single flux tope. From Equation (18), (36) and (39), it follows that

$$
\begin{aligned}
EFVs(CFC_{\mathbf{TC}}(\mathbf{M})) = EFVs(FC)_{\leq \mathbf{ts_M}} = \\
EFMs(CFC_{\mathbf{TC}}(\mathbf{M})) = EFMs(FC)_{\leq \mathbf{ts_M}} = ExVs(FC_{\leq \mathbf{ts_M}})
\end{aligned}
\tag{58}
$$

$$EFPs(CFP_{\mathbf{TC}}(\mathbf{M})) = EFPs(FP)_{\leq \mathbf{ts_M}} = ExPs(FP_{\leq \mathbf{ts_M}})$$
$$EFVs(CFP_{\mathbf{TC}}(\mathbf{M})) = EFVs(C_{FP})_{\leq \mathbf{ts_M}} = ExVs(C_{FP_{\leq \mathbf{ts_M}}})  \tag{59}$$
$$EFMs(CFP_{\mathbf{TC}}(\mathbf{M})) = EFMs(FP)_{\leq \mathbf{ts_M}}$$

and the same for $CFC_{\mathbf{TC}}(\mathbf{M})$ and $CFP_{\mathbf{TC}}(\underline{\mathbf{M}})$. Now, considering the decomposition of *FC* (resp., *FP*) into flux topes, we can proceed flux tope by flux tope for the computation of the sets above. Starting from a FT $FC_{\leq \eta}$ (resp., $FP_{\leq \eta}$), we have $(FC_{\leq \eta})_{\leq \mathbf{ts_M}} = FC_{\leq \eta'}$ (resp., $(FP_{\leq \eta})_{\leq \mathbf{ts_M}} = FP_{\leq \eta'}$) with $\eta' = inf(\eta, \mathbf{ts_M})$ (where *inf* is defined component-wise with $inf(-, +) = 0$) and the four sets above are equal to ExVs ($FC_{\leq \eta'}$), ExPs ($FP_{\leq \eta'}$), ExVs ($C_{FP_{\leq \eta'}}$) and EFMs ($FP_{\leq \eta'}$) respectively. This is in particular the case if we split each reversible reaction into two irreversible ones as, in higher dimension, $\widetilde{FC}$ (resp., $\widetilde{FP}$) is reduced to a single flux tope defined by $\eta = +$, the all-positive sign vector, and thus $\eta'$ is obtained from $\mathbf{ts_M}$ by changing each $-$ into 0. Note also that *FC* (resp., *FP*) consistent does not imply that $CFC_{\mathbf{TC}}(\mathbf{M})$ (resp., $CFP_{\mathbf{TC}}(\mathbf{M})$) is consistent ($\eta$ being full support does not imply that $\eta'$ is full support).

**Proposition 1.** *Given metabolite concentrations* $\mathbf{M}$, *the space of flux vectors in FC (resp., FP) satisfying the thermodynamic constraint* $\mathbf{TC_M}$ *is a flux cone (resp., a flux polyhedron), made up of vectors of FC (resp., FP) which conform to the thermodynamic sign vector* $\mathbf{ts_M}$ *given in Equation* (44). *Its elementary flux vectors, identical to elementary flux modes, (resp., elementary flux points, elementary flux vectors and elementary flux modes) are exactly those of FC (resp., FP) that satisfy the constraint, i.e., that conform to* $\mathbf{ts_M}$. *The same result holds for internal metabolite concentrations* $\underline{\mathbf{M}}$ *and thermodynamic constraint* $\underline{\mathbf{TC_M}}$ *with* $\mathbf{ts_{\underline{M}}}$.

Let us now consider the more usual case where the concentrations of the metabolites (at least those of internal metabolites) are unknown. From Equation (45), we obtain one or the other form for the thermodynamic constraint (existentially quantified on metabolite concentrations):

$$\mathbf{TC}(\mathbf{v}) \triangleq \exists \mathbf{M} \; \mathbf{TC_M}(\mathbf{v}) \triangleq \exists \mathbf{M} \; sign(\mathbf{v}) \leq \mathbf{ts_M}$$
$$\underline{\mathbf{TC}}(\mathbf{v}) \triangleq \exists \underline{\mathbf{M}} \; \underline{\mathbf{TC_M}}(\mathbf{v}) \triangleq \exists \underline{\mathbf{M}} \; sign(\mathbf{v}) \leq \mathbf{ts_{\underline{M}}}.  \tag{60}$$

Though the metabolite concentrations $\mathbf{M}_j$ are unknown, some lower bounds $\mathbf{M}_j^-$ and upper bounds $\mathbf{M}_j^+$ on these concentrations are often known. They are thus added as additional constraints:

$$\mathbf{TC}^b(\mathbf{v}) \triangleq \exists \mathbf{M} \; (sign(\mathbf{v}) \leq \mathbf{ts_M} \wedge \mathbf{M}^- \leq \mathbf{M} \leq \mathbf{M}^+)$$
$$\underline{\mathbf{TC}}^b(\mathbf{v}) \triangleq \exists \underline{\mathbf{M}} \; (sign(\mathbf{v}) \leq \mathbf{ts_{\underline{M}}} \wedge \underline{\mathbf{M}}^- \leq \underline{\mathbf{M}} \leq \underline{\mathbf{M}}^+).  \tag{61}$$

The solution space in $\mathbb{R}^r$ of these constraints is thus $Sol_{\mathbf{TC}} = \bigcup_{\mathbf{M}} Sol_{\mathbf{TC_M}}$ (idem for $\underline{\mathbf{TC}}$) and $Sol_{\mathbf{TC}^b} = \bigcup_{\mathbf{M} | \mathbf{M}^- \leq \mathbf{M} \leq \mathbf{M}^+} Sol_{\mathbf{TC_M}}$ (idem for $\underline{\mathbf{TC}}^b$). From the result of Lemma 1, it follows that $Sol_{\mathbf{TC}} = \bigcup_{\mathbf{M}} O_{\mathbf{ts_M}}$ and $Sol_{\mathbf{TC}^b} = \bigcup_{\mathbf{M} | \mathbf{M}^- \leq \mathbf{M} \leq \mathbf{M}^+} O_{\mathbf{ts_M}}$ (idem for $\underline{\mathbf{TC}}$ and $\underline{\mathbf{TC}}^b$ with $O_{\mathbf{ts_{\underline{M}}}}$). This is obviously enough to take the union on the maximal $\mathbf{ts_M}$'s (resp., $\mathbf{ts_{\underline{M}}}$'s) when $\mathbf{M}$ varies. As these are at most $2^r$, the union is finite.

**Lemma 3.** *The set* $Sol_{\mathbf{TC}}$ *(resp.,* $Sol_{\mathbf{TC}^b}$*) of vectors in* $\mathbb{R}^r$ *satisfying the thermodynamic constraint* $\mathbf{TC}$ *(resp.,* $\mathbf{TC}^b$*) is a union of closed orthants. More precisely, it is the (finite) union for all* $\mathbf{M}$ *(resp., all bounded* $\mathbf{M}$*) of the sets of vectors that conform to* $\mathbf{ts_M}$, *i.e.,* $O_{\mathbf{ts_M}}$*'s. The same result holds for* $\underline{\mathbf{TC}}$ *and* $\underline{\mathbf{TC}}^b$ *with* $\underline{\mathbf{M}}$ *and* $O_{\mathbf{ts_{\underline{M}}}}$.

It follows from Equations (53), (54), (56) and (57) that

$$CFC_{\mathbf{TC}} = \bigcup_{\mathbf{M}} FC_{\leq \mathbf{ts_M}} \qquad CFC_{\mathbf{TC}^b} = \bigcup_{\mathbf{M} | \mathbf{M}^- \leq \mathbf{M} \leq \mathbf{M}^+} FC_{\leq \mathbf{ts_M}}  \tag{62}$$

$$CFP_{\mathbf{TC}} = \bigcup_{\mathbf{M}} FP_{\leq \mathbf{ts_M}} \qquad CFP_{\mathbf{TC}^b} = \bigcup_{\mathbf{M}|\mathbf{M}^- \leq \mathbf{M} \leq \mathbf{M}^+} FP_{\leq \mathbf{ts_M}} \tag{63}$$

and the same for $\underline{\mathbf{TC}}$ and $\underline{\mathbf{TC}}^b$ with $\underline{\mathbf{M}}$ and $\mathbf{ts_{\underline{M}}}$, where all unions are finite.

From this and Equations (18), (36) and (39), we get

$$EFVs(CFC_{\mathbf{TC}}) = \bigcup_{\mathbf{M}} EFVs(FC_{\leq \mathbf{ts_M}}) = \bigcup_{\mathbf{M}} EFVs(FC)_{\leq \mathbf{ts_M}} = $$
$$EFMs(CFC_{\mathbf{TC}}) = \bigcup_{\mathbf{M}} EFMs(FC_{\leq \mathbf{ts_M}}) = \bigcup_{\mathbf{M}} EFMs(FC)_{\leq \mathbf{ts_M}} = \bigcup_{\mathbf{M}} ExVs(FC_{\leq \mathbf{ts_M}}) \tag{64}$$

$$EFPs(CFP_{\mathbf{TC}}) = \bigcup_{\mathbf{M}} EFPs(FP_{\leq \mathbf{ts_M}}) = \bigcup_{\mathbf{M}} EFPs(FP)_{\leq \mathbf{ts_M}} = \bigcup_{\mathbf{M}} ExPs(FP_{\leq \mathbf{ts_M}})$$
$$EFVs(CFP_{\mathbf{TC}}) = \bigcup_{\mathbf{M}} EFVs(C_{FP_{\leq \mathbf{ts_M}}}) = \bigcup_{\mathbf{M}} EFVs(C_{FP})_{\leq \mathbf{ts_M}} = \bigcup_{\mathbf{M}} ExVs(C_{FP_{\leq \mathbf{ts_M}}}) \tag{65}$$
$$EFMs(CFP_{\mathbf{TC}}) = \bigcup_{\mathbf{M}} EFMs(FP_{\leq \mathbf{ts_M}}) = \bigcup_{\mathbf{M}} EFMs(FP)_{\leq \mathbf{ts_M}}$$

and the analog for $\mathbf{TC}^b$, $\underline{\mathbf{TC}}$ and $\underline{\mathbf{TC}}^b$. Now, considering the decomposition of *FC* (resp., *FP*) into flux topes, we can proceed flux tope by flux tope for the computation of the sets above. Starting from a FT $FC_{\leq \eta}$ (resp., $FP_{\leq \eta}$), we have $\bigcup_{\mathbf{M}}(FC_{\leq \eta})_{\leq \mathbf{ts_M}} = \bigcup_i FC_{\leq \eta^i}$ (resp., $\bigcup_{\mathbf{M}}(FP_{\leq \eta})_{\leq \mathbf{ts_M}} = \bigcup_i FP_{\leq \eta^i}$) where $\{\eta^i\}$ are the maximal sign vectors in $\{inf(\eta, \mathbf{ts_M}) \mid \mathbf{M} \in \mathbb{R}_+^{*(m+\overline{m})}\}$ and the four sets above are equal to $\bigcup_i ExVs(FC_{\leq \eta^i})$, $\bigcup_i ExPs(FP_{\leq \eta^i})$, $\bigcup_i ExVs(C_{FP_{\leq \eta^i}})$ and $\bigcup_i EFMs(FP_{\leq \eta^i})$, respectively. This is in particular the case if we split each reversible reaction into two irreversible ones as, in higher dimension, $\widetilde{FC}$ (resp., $\widetilde{FP}$) is reduced to a single flux tope defined by $\eta = +$. The analog holds for $\mathbf{TC}^b$, $\underline{\mathbf{TC}}$ and $\underline{\mathbf{TC}}^b$.

**Proposition 2.** *The space of flux vectors in FC (resp., FP) satisfying the thermodynamic constraint TC, or $TC^b$, is a finite union of flux cones (resp., flux polyhedra), obtained as those vectors of FC (resp., FP) which conform to $\mathbf{ts_M}$, for a certain $\mathbf{M}$, or bounded $\mathbf{M}$. It is thus no longer convex but made up of particular faces $FC_{\leq \eta^i}$ (resp., $FP_{\leq \eta^i}$) of each flux tope $FC_{\leq \eta}$ for FC (resp., $FP_{\leq \eta}$ for FP), with $\eta^i \leq \eta$. Its elementary flux vectors, identical to elementary flux modes, (resp., elementary flux points, elementary flux vectors and elementary flux modes) are exactly those of FC (resp., FP) that satisfy the constraint, i.e., that conform to a certain $\mathbf{ts_M}$, and coincide thus with the extreme vectors (resp., the extreme points, extreme vectors and elementary modes) of the $FC_{\leq \eta^i}$'s (resp., $FP_{\leq \eta^i}$'s). The same result holds for $\underline{TC}$, or $\underline{TC}^b$, with $\underline{\mathbf{M}}$ and $\mathbf{ts_{\underline{M}}}$.*

We will now refine this result in order to characterize the faces involved. Consider the case of a flux cone where the concentrations of external metabolites are given and assume first there are no bounds on the concentrations of the internal metabolites [37]. As $CFC_{\mathbf{TC}} = FC \cap \bigcup_{\mathbf{M}} O_{\mathbf{ts_M}}$ and $CFP_{\mathbf{TC}} = FP \cap \bigcup_{\mathbf{M}} O_{\mathbf{ts_M}}$, let us begin by studying $\bigcup_{\mathbf{M}} O_{\mathbf{ts_M}}$. We have $\mathbf{x} \in \bigcup_{\mathbf{M}} O_{\mathbf{ts_M}} \Leftrightarrow \exists \mathbf{M} \in \mathbb{R}_+^{*m} (sign(\mathbf{x}) \leq \mathbf{ts_M}) \Leftrightarrow \exists \mathbf{M} \in \mathbb{R}_+^{*m} \forall i, 1 \leq i \leq r, (\overline{sign}(\mathbf{x}_i) \leq -sign(\prod_j \underline{\mathbf{M}}_j^{\mathbf{S}_{ji}} - \hat{K}_{eq}^i))$ from Equation (44). By applying the monotonic logarithm function and noting $\mathbf{LM} \in \mathbb{R}^m$ the vector whose components are given by $\mathbf{LM}_j = ln(\underline{\mathbf{M}}_j)$, we obtain $\mathbf{x} \in \bigcup_{\underline{\mathbf{M}}} O_{\mathbf{ts_M}} \Leftrightarrow \exists \mathbf{LM} \in \mathbb{R}^m \forall i, 1 \leq i \leq r, (sign(\mathbf{x}_i) \leq -sign(\sum_j \mathbf{S}_{ji} \mathbf{LM}_j - ln(\hat{K}_{eq}^i))) \Leftrightarrow \exists \mathbf{LM} \in \mathbb{R}^m \forall i \in supp(\mathbf{x}) (sign(\mathbf{x}_i) \sum_j \mathbf{S}_{ji} \mathbf{LM}_j < sign(\mathbf{x}_i) ln(\hat{K}_{eq}^i))$. We will now make use of Gale's theorem (or Kuhn–Fourier theorem), which is a form of Farkas' duality lemma (for two vectors $\mathbf{x}, \mathbf{y}, \mathbf{x} < \mathbf{y}$ means $\mathbf{x}_i < \mathbf{y}_i$ for all $i$):

**Gale's theorem** (a form of Farkas' lemma). *For any $A \in \mathbb{R}^{p \times q}$ and $b \in \mathbb{R}^p$, exactly one of the following statements holds*:

(a)     *there exists $y \in \mathbb{R}^q$ such that $Ay < b$;*

(b)     *there exists $z \in \mathbb{R}^p \setminus \{0\}$ such that $z \geq 0$, $z^T A = 0$ and $z^T b \leq 0$.*

Applying this theorem in the formula above to $A \in \mathbb{R}^{supp(\mathbf{x}) \times m}$ (where we note $\mathbb{R}^{supp(\mathbf{x})} \triangleq \mathbb{R}^r \cap \bigcap_{i|\mathbf{x}_i=0} \{\mathbf{z}_i = 0\}$ the subspace of vectors having a null component outside the support of $\mathbf{x}$), given by $A_{ij} = sign(\mathbf{x}_i)\mathbf{S}_{ji}$, and $b \in \mathbb{R}^{supp(\mathbf{x})}$, given by $b_i = sign(\mathbf{x}_i)ln(\hat{K}_{eq}^i)$,

provides $\mathbf{x} \in \bigcup_{\underline{\mathbf{M}}} O_{\mathrm{ts}_{\underline{\mathbf{M}}}} \Leftrightarrow \forall \mathbf{z} \in \mathbb{R}^{supp(\mathbf{x})} \backslash \{\mathbf{0}\} (\mathbf{z} \geq \mathbf{0}, \forall j, 1 \leq j \leq m, \sum_i \mathbf{S}_{ji} sign(\mathbf{x}_i) \mathbf{z}_i = 0 \Rightarrow \sum_i sign(\mathbf{x}_i) \mathbf{z}_i ln(\hat{K}^i_{eq}) > 0) \Leftrightarrow \forall \mathbf{z} \in \mathbb{R}^r \backslash \{\mathbf{0}\} (sign(\mathbf{z}) \leq sign(\mathbf{x}), \mathbf{Sz} = \mathbf{0} \Rightarrow \mathbf{z}^T \mathbf{ln\hat{K}_{eq}} > 0)$ where $\mathbf{ln\hat{K}_{eq}} \in \mathbb{R}^r$ is defined by: $(\mathbf{ln\hat{K}_{eq}})_i = ln(\hat{K}^i_{eq})$.

We thus obtain $CFC_{\underline{\mathbf{TC}}} = \{\mathbf{v} \in FC \mid \forall \mathbf{z} \in FC \backslash \{\mathbf{0}\} (sign(\mathbf{z}) \leq sign(\mathbf{v}) \Rightarrow \mathbf{z}^T \mathbf{ln\hat{K}_{eq}} > 0)\}$. This can be rewritten as $CFC_{\underline{\mathbf{TC}}} = \{\mathbf{v} \in FC \mid FC_{\leq sign(\mathbf{v})} \backslash \{\mathbf{0}\} \subseteq H^+_{\mathbf{ln\hat{K}_{eq}}}\}$, where $H^+_{\mathbf{ln\hat{K}_{eq}}} \overset{\triangle}{=} \{\mathbf{z} \in \mathbb{R}^r \mid \mathbf{z}^T \mathbf{ln\hat{K}_{eq}} > 0\}$ is an open half-space with the hyperplane $H_{\mathbf{ln\hat{K}_{eq}}} \overset{\triangle}{=} \{\mathbf{z} \in \mathbb{R}^r \mid \mathbf{z}^T \mathbf{ln\hat{K}_{eq}} = 0\}$ as a frontier. Considering the decomposition of $FC$ into FTs $FC_{\leq \eta}$, this is equivalent to $CFC_{\underline{\mathbf{TC}} \leq \eta} = \{\mathbf{v} \in FC_{\leq \eta} \mid FC_{\leq sign(\mathbf{v})} \backslash \{\mathbf{0}\} \subseteq H^+_{\mathbf{ln\hat{K}_{eq}}}\}$ for all $\eta$'s maximal sign vectors in $sign(FC)$. Now, note that, in this formula, $FC_{\leq sign(\mathbf{v})}$ is the minimal face (for set inclusion) of $FC_{\leq \eta}$ containing $\mathbf{v}$. We finally obtain finally:

$$CFC_{\underline{\mathbf{TC}} \leq \eta} = \bigcup_i FC_{\leq \eta^i} \text{ for all maximal } \eta^i \leq \eta \text{ s.t. } FC_{\leq \eta^i} \backslash \{\mathbf{0}\} \subseteq H^+_{\mathbf{ln\hat{K}_{eq}}}. \tag{66}$$

That is to say, $CFC_{\underline{\mathbf{TC}}}$ is the union of all the maximal faces $FC_{\leq \eta^i}$ of the FTs for $FC$ such that $FC_{\leq \eta^i} \backslash \{\mathbf{0}\} \subseteq H^+_{\mathbf{ln\hat{K}_{eq}}}$ or, equivalently, the union of all maximal faces $F_i$ of the $FC \cap O$'s for all $r$-orthants $O$ such that $F_i \backslash \{\mathbf{0}\} \subseteq H^+_{\mathbf{ln\hat{K}_{eq}}}$ (see Figure 3).

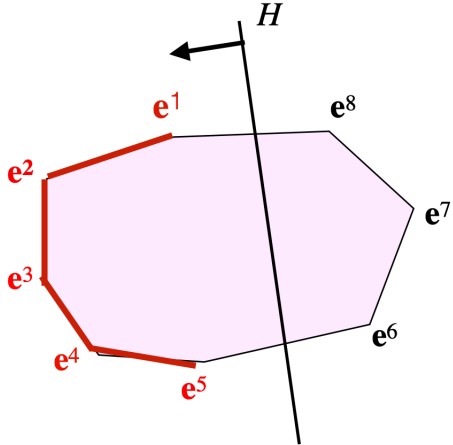

**Figure 3.** Structure of the solution space in the presence of the thermodynamic constraint (without bounds on the metabolite concentrations). We consider a given flux tope and represent the section of the flux cone with an affine hyperplane (polytope shaded in pink). Vertices $\mathbf{e}^1$ to $\mathbf{e}^8$ thus represent the EFMs. When we add the thermodynamic constraint, the solution space is the union of all the maximal faces of the flux cone that are entirely contained in the open half-space $\{\mathbf{z} \in \mathbb{R}^r \mid \mathbf{z}^T \mathbf{ln\hat{K}_{eq}} > 0\}$ (left side of frontier $H$ in the section), represented in red. In particular, the thermodynamically feasible EFMs are given by the vertices $\mathbf{e}^1$ to $\mathbf{e}^5$ that belong to this half-space. The set of these EFMs is decomposed into LTCSs, made up of the EFMs of the maximal faces above (in the 2D shape, these LTCSs are thus given by $\{\{\mathbf{e}^1, \mathbf{e}^2\}, \{\mathbf{e}^2, \mathbf{e}^3\}, \{\mathbf{e}^3, \mathbf{e}^4\}, \{\mathbf{e}^4, \mathbf{e}^5\}\}$).

A topological consequence of this characterization is that $CFC_{\underline{\mathbf{TC}}} \backslash \{\mathbf{0}\}$ is a connected set (actually arc-connected). Another consequence, regarding EFMs (or, equivalently, EFVs), is:

$$EFMs(CFC_{\underline{\mathbf{TC}}}) = EFMs(FC) \cap Sol_{\underline{\mathbf{TC}}} = EFMs(FC) \cap H^+_{\mathbf{ln\hat{K}_{eq}}} \tag{67}$$

i.e., the elementary flux modes in the (non-convex) cone of those flux vectors in *FC* satisfying the thermodynamic constraint **TC** are exactly those elementary flux modes in *FC* that satisfy **TC**, i.e., that belong to the open half-space $H^+_{\ln \hat{K}_{eq}}$. We can equivalently write

$$
\begin{aligned}
EFMs(CFC_{\underline{TC}}) &= EFMs(FC^+_{\ln \hat{K}_{eq}}) \backslash H_{\ln \hat{K}_{eq}} \\
\text{with } FC^+_{\ln \hat{K}_{eq}} &\triangleq \{\mathbf{v} \in \mathbb{R}^r \mid \mathbf{Sv} = \mathbf{0}, \mathbf{v}_i \geq 0 \text{ for } i \in \mathbf{Irr}, \mathbf{v}^T \ln \hat{K}_{eq} \geq 0\}
\end{aligned}
\tag{68}
$$

where $FC^+_{\ln \hat{K}_{eq}}$ is the polyhedral cone obtained from *FC* merely by adding the homogeneous linear inequality $\mathbf{v}^T \ln \hat{K}_{eq} \geq 0$ [38]. Each flux vector of $CFC_{\underline{TC}}$ is a conformal conical sum of these EFMs, but the converse is false as $CFC_{\underline{TC}}$ is not convex. Thus the set of EFMs, considered globally, does not characterize the solution space $CFC_{\underline{TC}}$. More precisely, we have

$$
CFC_{\underline{TC} \leq \boldsymbol{\eta}} = \bigcup_i cone_{\oplus}(E_i) \text{ with } E_i = EFMs(FC_{\leq \boldsymbol{\eta}^i})
\tag{69}
$$

where $FC_{\leq \boldsymbol{\eta}^i}$ is as in Equation (66), i.e., $CFC_{\underline{TC}}$ is characterized by the decomposition of the set of EFMs into the (non-disjoint) subsets $E_i$. Now, the $E_i$'s are exactly the maximal subsets of EFMs included in a given flux tope for *FC* (i.e., in a given *r*-orthant *O*) and whose conical hull is included in $CFC_{\underline{TC}}$, i.e., all vectors in this hull must satisfy the constraint **TC**: $E_i$ maximal such that $E_i \subseteq$ EFMs $(CFC_{\underline{TC}})$ and $E_i \subseteq O$ with *O* *r*-orthant and $cone_{\oplus}(E_i) \subseteq CFC_{\underline{TC}}$. Such an $E_i$ is called a largest thermodynamically consistent set (LTCS) of EFMs [39] in *O* (or, equivalently, in the flux tope $FC_{\leq \boldsymbol{\eta}} = FC \cap O$ defined by *O*).

We could want to estimate the ratio of thermodynamically feasible EFMs on all EFMs, i.e., the ratio of EFMs $(CFC_{\underline{TC}})$ on EFMs $(FC)$. If all reactions are reversible ($r_I = 0$), then the function $\mathbf{v} \mapsto -\mathbf{v}$ maps the EFMs on one side of hyperplane $H_{\ln \hat{K}_{eq}}$ onto the EFMs on the other side. Thus, if we neglect the EFMs that might belong to this hyperplane, it means that 50% of the EFMs are thermodynamically feasible (and thus only 50% eliminated). Now, intuitively, irreversible reactions given in **Irr** come from an expert knowledge that can be seen as a form of compiled thermodynamic knowledge, as we saw that it is thermodynamics which imposes the direction in which a reaction may proceed. Therefore, we can assume, provided the adequacy of the model for some given environment (such as the concentrations of external metabolites, supposed here to be known), that any thermodynamically feasible EFM satisfies these irreversibility constraints. Under this assumption, the irreversibility constraints rule out only thermodynamically unfeasible EFMs so if we then apply the thermodynamic constraint **TC** we only eliminate at most than 50% of the remaining EFMs (from 50% when all reactions are reversible to 0% when all are irreversible, without splitting any reversible reaction into two irreversible ones).

If inhomogeneous linear constraints are added, we have $CFP_{\underline{TC}} = CFC_{\underline{TC}} \cap P_{\mathbf{ILC}}$, with $P_{\mathbf{ILC}} = \{\mathbf{v} \in \mathbb{R}^r \mid \mathbf{Gv} \geq \mathbf{h}\}$ Equation (25) and we obtain in the same way, for all $\boldsymbol{\eta}$'s maximal sign vectors in *sign*(*FC*)

$$
CFP_{\underline{TC} \leq \boldsymbol{\eta}} = \bigcup_i FP_{\leq \boldsymbol{\eta}^i} \text{ for all maximal } \boldsymbol{\eta}^i \leq \boldsymbol{\eta} \text{ s.t. } FC_{\leq \boldsymbol{\eta}^i} \backslash \{\mathbf{0}\} \subseteq H^+_{\ln \hat{K}_{eq}}.
\tag{70}
$$

That is to say, $CFP_{\underline{TC}}$ is the union of all the $FP_{\leq \boldsymbol{\eta}^i} = FC_{\leq \boldsymbol{\eta}^i} \cap P_{\mathbf{ILC}}$ such that $FC_{\leq \boldsymbol{\eta}^i}$ is a maximal face of a FT for *FC* verifying $FC_{\leq \boldsymbol{\eta}^i} \backslash \{\mathbf{0}\} \subseteq H^+_{\ln \hat{K}_{eq}}$ or, equivalently, the union of the $F_i \cap P_{\mathbf{ILC}}$ for all maximal faces $F_i$ of the $FC \cap O$'s for all *r*-orthants *O* such that $F_i \backslash \{\mathbf{0}\} \subseteq H^+_{\ln \hat{K}_{eq}}$. Take care because the $FP_{\leq \boldsymbol{\eta}^i}$'s involved are faces of $FP_{\leq \boldsymbol{\eta}}$ included in $H^+_{\ln \hat{K}_{eq}} \cup \{\mathbf{0}\}$, but not any face of $FP_{\leq \boldsymbol{\eta}}$ included in $H^+_{\ln \hat{K}_{eq}} \cup \{\mathbf{0}\}$ is included in $CFP_{\underline{TC}}$ (as such a face may be defined by inhomogeneous equality constraints coming from **ILC** and not only by nullity constraints of the form $\mathbf{v}_i = 0$). $CFP_{\underline{TC}}$ is no longer a connected set in general.

We can sum up these results as follows.

**Proposition 3.** *The space of flux vectors in FC satisfying the thermodynamic constraint $\underline{\textbf{TC}}$ is a finite union of flux cones, obtained as all the maximal faces of all the flux topes $FC_{\leq\eta}$ that are entirely contained (except the null vector) in the open half-space $H^+_{ln\hat{K}_{eq}} = \{z \mid z^T ln\hat{K}_{eq} > 0\}$. The thermodynamically feasible EFMs are thus those EFMs which belong to this half-space and can be simply computed as the EFMs of the flux cone obtained from FC by adding to it the homogeneous linear inequality $v^T ln\hat{K}_{eq} \geq 0$ (and removing those EFMs that would belong to the frontier hyperplane of $H^+_{ln\hat{K}_{eq}}$). The set of these EFMs can be decomposed into (non-disjoint) maximal subsets of EFMs belonging to a same flux tope (i.e., a same r-orthant) and whose conical hull is made up of thermodynamically feasible vectors, each of these subsets thus representing the set of EFMs of one of the maximal faces above. At most 50% of the EFMs can thus be ruled out as thermodynamically infeasible, if we assume that no thermodynamically feasible flux vector violates given irreversibility constraints. In the presence of additional inhomogeneous linear constraints on flux vectors given by $Gv \geq h$, the space of flux vectors in FP satisfying $\underline{\textbf{TC}}$ is a finite union of flux polyhedra, obtained as intersections of the flux cones above with the polyhedron defined by $Gv \geq h$. All these results hold for constraint $\textbf{TC}$ by just replacing the $\hat{K}^i_{eq}$'s by the $K^i_{eq}$'s.*

This applies in particular to $\widetilde{FC}$ and $\widetilde{FP}$ (after splitting each reversible reaction into two irreversible ones) with the simplification, as $\widetilde{FC}$ (resp., $\widetilde{FP}$) is reduced to a single flux tope defined by $\eta = +$, that we must just consider the maximal faces of $\widetilde{FC}$ that are entirely contained (except the null vector) in the open half-space $H^+_{\mathbf{ln\hat{K}_{eq}}}$.

Consider now the case where certain bounds on the concentrations of internal metabolites are known, i.e., the case of the thermodynamic constraint $\underline{\textbf{TC}}^b$. We have $CFC_{\underline{\textbf{TC}}^b} = FC \cap \bigcup_{\underline{\textbf{M}} \mid \underline{\textbf{M}}^- \leq \underline{\textbf{M}} \leq \underline{\textbf{M}}^+} O_{ts_{\underline{\textbf{M}}}}$. From what precedes, we obtain $\mathbf{x} \in \bigcup_{\underline{\textbf{M}} \mid \underline{\textbf{M}}^- \leq \underline{\textbf{M}} \leq \underline{\textbf{M}}^+} O_{ts_{\underline{\textbf{M}}}} \Leftrightarrow \exists \underline{\textbf{LM}} \in \mathbb{R}^m (\forall i \in supp(\mathbf{x}) \, (sign(\mathbf{x}_i) \sum_j \mathbf{S}_{ji} \underline{\textbf{LM}}_j < sign(\mathbf{x}_i) ln(\hat{K}^i_{eq})) \wedge \underline{\textbf{LM}} \leq ln(\underline{\textbf{M}}^+) \wedge -\underline{\textbf{LM}} \leq -ln(\underline{\textbf{M}}^-))$. Applying Gale's theorem in this formula to $\begin{pmatrix} \mathbf{A} \\ \mathbf{I}_m \\ -\mathbf{I}_m \end{pmatrix} \in \mathbb{R}^{(supp(\mathbf{x})+2m) \times m}$, with $\mathbf{A}_{ij} = sign(\mathbf{x}_i)\mathbf{S}_{ji}$, and $\begin{pmatrix} \mathbf{b} \\ ln\underline{\textbf{M}}^+ \\ -ln\underline{\textbf{M}}^- \end{pmatrix} \in \mathbb{R}^{supp(\mathbf{x})+2m}$, with $\mathbf{b}_i = sign(\mathbf{x}_i) \, ln(\hat{K}^i_{eq})$, provides $\mathbf{x} \in \bigcup_{\underline{\textbf{M}} \mid \underline{\textbf{M}}^- \leq \underline{\textbf{M}} \leq \underline{\textbf{M}}^+} O_{ts_{\underline{\textbf{M}}}} \Leftrightarrow \forall \begin{pmatrix} \mathbf{z} \\ \underline{\mathbf{z}} \\ \bar{\mathbf{z}} \end{pmatrix} \in \mathbb{R}^{r+2m} (\mathbf{z} \neq \mathbf{0}, \underline{\mathbf{z}} \geq \mathbf{0}, \bar{\mathbf{z}} \geq \mathbf{0}, sign(\mathbf{z}) \leq sign(\mathbf{x}),$ $\mathbf{Sz} + \underline{\mathbf{z}} - \bar{\mathbf{z}} = \mathbf{0} \Rightarrow (\mathbf{z}^T \underline{\mathbf{z}}^T \bar{\mathbf{z}}^T) \mathbf{ln\hat{K}^b_{eq}} > 0)$ where $\mathbf{ln\hat{K}^b_{eq}} \in \mathbb{R}^{r+2m}$ is defined by $\mathbf{ln\hat{K}^b_{eq}} = \begin{pmatrix} \mathbf{ln\hat{K}_{eq}} \\ ln\underline{\textbf{M}}^+ \\ ln(1/\underline{\textbf{M}}^-) \end{pmatrix} = ln\begin{pmatrix} \mathbf{\hat{K}_{eq}} \\ \underline{\textbf{M}}^+ \\ 1/\underline{\textbf{M}}^- \end{pmatrix}$. Let $FC^b \triangleq \{\begin{pmatrix} \mathbf{z} \\ \underline{\mathbf{z}} \\ \bar{\mathbf{z}} \end{pmatrix} \in \mathbb{R}^{r+2m} \mid (\mathbf{S} \, \mathbf{I}_m \, -\mathbf{I}_m) \begin{pmatrix} \mathbf{z} \\ \underline{\mathbf{z}} \\ \bar{\mathbf{z}} \end{pmatrix} = \mathbf{0},$ $\mathbf{z}_i \geq 0$ for $i \in \mathbf{Irr}, \underline{\mathbf{z}} \geq \mathbf{0}, \bar{\mathbf{z}} \geq \mathbf{0}\} = \{\begin{pmatrix} \mathbf{z} \\ \mathbf{w} + (\mathbf{Sz})^- \\ \mathbf{w} + (\mathbf{Sz})^+ \end{pmatrix} \in \mathbb{R}^{r+2m} \mid \mathbf{z}_i \geq 0$ for $i \in \mathbf{Irr}, \mathbf{w} \geq \mathbf{0}\}$, where $(\mathbf{Sz})^+_j = max((\mathbf{Sz})_j, 0)$ and $(\mathbf{Sz})^-_j = max(-(\mathbf{Sz})_j, 0)$. $FC^b$ is a flux cone in $\mathbb{R}^{r+2m}$ of dimension $r + m$ and $FC^b \cap (\mathbb{R}^r \times \{\mathbf{0}\}) = FC$. We thus obtain $CFC_{\underline{\textbf{TC}}^b} = \{\mathbf{v} \in FC \mid \forall (\mathbf{z}^T \underline{\mathbf{z}}^T \bar{\mathbf{z}}^T)^T \in FC^b \, (\mathbf{z} \neq \mathbf{0}, sign(\mathbf{z}) \leq sign(\mathbf{v}) \Rightarrow (\mathbf{z}^T \underline{\mathbf{z}}^T \bar{\mathbf{z}}^T) \mathbf{ln\hat{K}^b_{eq}} > 0)\}$. This is equivalent to: $CFC_{\underline{\textbf{TC}}^b \leq \eta} = \{\mathbf{v} \in FC_{\leq\eta} \mid \forall (\mathbf{z}^T \underline{\mathbf{z}}^T \bar{\mathbf{z}}^T)^T \in FC^b \, (\mathbf{z} \in O_{sign(\mathbf{v})} \setminus \{\mathbf{0}\} \Rightarrow (\mathbf{z}^T \underline{\mathbf{z}}^T \bar{\mathbf{z}}^T) \mathbf{ln\hat{K}^b_{eq}} > 0)\} = \{\mathbf{v} \in FC_{\leq\eta} \mid \forall \mathbf{z} \in \mathbb{R}^r (\mathbf{z} \in O_{sign(\mathbf{v})} \setminus \{\mathbf{0}\} \Rightarrow (\mathbf{z}^T (\mathbf{Sz})^{-T} (\mathbf{Sz})^{+T}) \mathbf{ln\hat{K}^b_{eq}} > 0)\}$ for all flux topes $FC_{\leq\eta}$ for $FC$. This can be rewritten as $CFC_{\underline{\textbf{TC}}^b \leq \eta} = \{\mathbf{v} \in FC_{\leq\eta} \mid FC^b \cap (O_{sign(\mathbf{v})} \setminus \{\mathbf{0}\} \times \mathbb{R}^{2m}_+) \subseteq H^+_{\mathbf{ln\hat{K}^b_{eq}}}\}$, where we note $H^+_{\mathbf{ln\hat{K}^b_{eq}}} \triangleq \{\mathbf{Z} \in \mathbb{R}^{r+2m} \mid \mathbf{Z}^T \mathbf{ln\hat{K}^b_{eq}} > 0\}$ the open half-space with hyperplane $H_{\mathbf{ln\hat{K}^b_{eq}}} \triangleq \{\mathbf{Z} \in \mathbb{R}^{r+2m} \mid \mathbf{Z}^T \mathbf{ln\hat{K}^b_{eq}} = 0\}$ as a frontier.

We have $H^+_{\ln\hat{K}^b_{eq}} \cap \mathbb{R}^r = H^+_{\ln\hat{K}_{eq}}$ and $H_{\ln\hat{K}^b_{eq}} \cap \mathbb{R}^r = H_{\ln\hat{K}_{eq}}$. Now, note that, in the formula above, $FC^b \cap (O_{sign(\mathbf{v})} \times \mathbb{R}^{2m}_+)$ is a face of the cone $FC^b_{\leq\boldsymbol{\eta}+}$ (we note $\boldsymbol{\eta}+$ the sign vector of dimension $r + 2m$ obtained by concatenation of $\boldsymbol{\eta}$ of dimension $r$ and $+$ of dimension $2m$, thus $FC^b_{\leq\boldsymbol{\eta}+} = FC^b \cap (O_{\boldsymbol{\eta}} \times \mathbb{R}^{2m}_+))$ whose intersection with $\mathbb{R}^r \times \{\mathbf{0}\}$ is equal to $FC_{\leq sign(\mathbf{v})}$, the minimal face of the tope $FC_{\leq\boldsymbol{\eta}}$ containing $\mathbf{v}$. Actually, among all the faces of $FC^b_{\leq\boldsymbol{\eta}+}$ whose intersection with $\mathbb{R}^r \times \{\mathbf{0}\}$ is equal to $FC_{\leq sign(\mathbf{v})}$, this is the maximal one (without any constraint on the $\mathbb{R}^{2m}_+$ factor). Thus, finally,

$$CFC_{\underline{\mathbf{TC}}^b \leq\boldsymbol{\eta}} = \bigcup_j FC_{\leq\boldsymbol{\eta}^j} \text{ for all maximal } \boldsymbol{\eta}^j \leq \boldsymbol{\eta} \text{ s.t. } F^b_j \backslash (\{\mathbf{0}\} \times \mathbb{R}^{2m}) \subseteq H^+_{\ln\hat{K}^b_{eq}}, \qquad (71)$$

where $F^b_j$ is the maximal face of $FC^b_{\leq\boldsymbol{\eta}+}$ whose intersection with $\mathbb{R}^r \times \{\mathbf{0}\}$ is equal to $FC_{\leq\boldsymbol{\eta}^j}$.

Note that any $FC_{\leq\boldsymbol{\eta}^j}$ given by Equation (71) is a face of a certain $FC_{\leq\boldsymbol{\eta}^i}$ given by Equation (66) (i.e., $\forall\boldsymbol{\eta}^j \exists\boldsymbol{\eta}^i \, \boldsymbol{\eta}^j \leq \boldsymbol{\eta}^i$). $CFC_{\underline{\mathbf{TC}}^b} \backslash \{\mathbf{0}\}$ is no longer a connected set in general. A consequence, for EFMs (or, equivalently, EFVs), is:

$$\begin{aligned} EFMs(CFC_{\underline{\mathbf{TC}}^b \leq\boldsymbol{\eta}}) &= EFMs(FC_{\leq\boldsymbol{\eta}}) \cap Sol_{\underline{\mathbf{TC}}^b} \\ &= \{\mathbf{v} \in EFMs(FC_{\leq\boldsymbol{\eta}}) \mid F^b_{\mathbf{v}} \backslash (\{\mathbf{0}\} \times \mathbb{R}^{2m}) \subseteq H^+_{\ln\hat{K}^b_{eq}}\}, \end{aligned} \qquad (72)$$

where $F^b_{\mathbf{v}}$ is the maximal face of $FC^b_{\leq\boldsymbol{\eta}+}$ s.t. $F^b_{\mathbf{v}} \cap (\mathbb{R}^r \times \{\mathbf{0}\}) = \mathbb{R}_+\mathbf{v}$. This means that the elementary flux modes in the (non-convex) cone of those flux vectors in $FC_{\leq\boldsymbol{\eta}}$ satisfying the thermodynamic constraint $\underline{\mathbf{TC}}^b$ are exactly those elementary flux modes $\mathbf{v}$ in $FC_{\leq\boldsymbol{\eta}}$ (or in $CFC_{\underline{\mathbf{TC}} \leq\boldsymbol{\eta}}$) that satisfy $\underline{\mathbf{TC}}^b$, i.e., such that the maximal face of $FC^b_{\leq\boldsymbol{\eta}+}$ whose intersection with $\mathbb{R}^r \times \{\mathbf{0}\}$ is equal to the ray $\mathbb{R}_+\mathbf{v}$ is included in $H^+_{\ln\hat{K}_{eq}} \cup (\{\mathbf{0}\} \times \mathbb{R}^{2m})$. Note that EFMs $(CFC_{\underline{\mathbf{TC}}^b})$ is thus given by

$$\begin{aligned} \{\mathbf{v} \in EFMs(FC) \mid &\forall \mathbf{z} \in \mathbb{R}^r \\ &(sign(\mathbf{z}) \leq sign(\mathbf{v}), (\mathbf{z}^T(\mathbf{Sz})^{-T}(\mathbf{Sz})^{+T})\ln\hat{K}^b_{eq} \leq 0 \Rightarrow \mathbf{z} = \mathbf{0})\}. \end{aligned} \qquad (73)$$

Each flux vector of $CFC_{\underline{\mathbf{TC}}^b}$ is a conformal conical sum of these EFMs, but the converse is false as $CFC_{\underline{\mathbf{TC}}^b}$ is not convex. More precisely, we have

$$CFC_{\underline{\mathbf{TC}}^b \leq\boldsymbol{\eta}} = \bigcup_j cone_\oplus(E_j) \text{ with } E_j = EFMs(FC_{\leq\boldsymbol{\eta}^j}) \qquad (74)$$

where $FC_{\leq\boldsymbol{\eta}^j}$ is as in Equation (71), i.e., $CFC_{\underline{\mathbf{TC}}^b}$ is characterized by the decomposition of the set of EFMs into the (non-disjoint) subsets $E_j$. Now, the $E_j$'s are exactly the maximal subsets of EFMs included in a given flux tope for $FC$ (i.e., in a given $r$-orthant $O$) and whose conical hull is included in $CFC_{\underline{\mathbf{TC}}^b}$, i.e., all vectors in this hull must satisfy the constraint $\underline{\mathbf{TC}}^b$: $E_j$ maximal such that $E_j \subseteq$ EFMs $(CFC_{\underline{\mathbf{TC}}^b})$ and $E_j \subseteq O$ with $O$ $r$-orthant and $cone_\oplus(E_j) \subseteq CFC_{\underline{\mathbf{TC}}^b}$. Such an $E_j$ is called a largest thermodynamically consistent (for bounded concentrations of internal metabolites) set (LTCbS) of EFMs [39] in $O$ (or, equivalently, in the flux tope $FC_{\leq\boldsymbol{\eta}} = FC \cap O$ defined by $O$) and is included in a certain LTCS $E_i$ as in Equation (69).

If inhomogeneous linear constraints **ILC** are added, we have, for all $\boldsymbol{\eta}$'s maximal sign vectors in $sign(FC)$:

$$CFP_{\underline{\mathbf{TC}}^b \leq\boldsymbol{\eta}} = \bigcup_j FP_{\leq\boldsymbol{\eta}^j} \qquad (75)$$

with $\boldsymbol{\eta}^j$ as in Equation (71). That is to say, $CFP_{\underline{\mathbf{TC}}^b}$ is the union of all the $FP_{\leq\boldsymbol{\eta}^j} = FC_{\leq\boldsymbol{\eta}^j} \cap P_{\mathbf{ILC}}$ such that $FC_{\leq\boldsymbol{\eta}^j}$ is given as in Equation (71).

We can sum up these results as follows.

**Proposition 4.** *For FC a flux cone in $\mathbb{R}^r$, let us define its lift to $\mathbb{R}^{r+2m}$ as the flux cone $FC^b = \{Z \in \mathbb{R}^r \times \mathbb{R}^{2m}_+) \mid (S\, I_m\, {-}I_m)Z = 0, Z_i \geq 0 \text{ for } i \in Irr, 1 \leq i \leq r\}$. Thus, $FC^b \cap (\mathbb{R}^r \times \{0\}) = FC$. For any flux tope $FC_{\leq \eta}$ for FC and any face $FC'$ of $FC_{\leq \eta}$, its lift to $\mathbb{R}^{r+2m}$ is defined as the maximal face of $FC^b_{\leq \eta+}$ whose intersection with $\mathbb{R}^r \times \{0\}$ is equal to $FC'$. The space of flux vectors in FC satisfying the thermodynamic constraint $\underline{TC}^b$ is a finite union of flux cones, obtained as all the maximal faces of all the flux topes $FC_{\leq \eta}$ whose lifts to $\mathbb{R}^{r+2m}$ are entirely contained (except vectors from $\{0\} \times \mathbb{R}^{2m}$) in the open half-space $H^+_{ln\hat{K}^b_{eq}} = \{Z \in \mathbb{R}^{r+2m} \mid Z^T ln\hat{K}^b_{eq} > 0\}$, where $ln\hat{K}^b_{eq} = ln(\hat{K}_{eq} \underline{M}^+ 1/\underline{M}^-)^T$. The thermodynamically feasible EFMs in $FC_{\leq \eta}$ are those EFMs in $FC_{\leq \eta}$ whose lifts (as rays) are contained (except vectors from $\{0\} \times \mathbb{R}^{2m}$) in this half-space. The set of these EFMs can be decomposed into (non-disjoint) maximal subsets of EFMs belonging to a same flux tope (i.e., a same r-orthant) and whose conical hull is made up of thermodynamically feasible vectors, each of these subsets representing thus the set of EFMs of one of the maximal faces above. In the presence of additional inhomogeneous linear constraints on flux vectors given by $Gv \geq h$, the space of flux vectors in FP satisfying $\underline{TC}^b$ is a finite union of flux polyhedra, obtained as intersections of the flux cones above with the polyhedron defined by $Gv \geq h$.*

### 2.2.2. Application to Kinetics

In the same way, we obtain for the kinetic constraint in the absence of knowledge regarding values of concentrations of enzymes and metabolites:

$$\mathbf{KC}(\mathbf{v}) \; \overset{\triangle}{=} \; \exists \mathbf{E}, \mathbf{M}\; \mathbf{KC}_{\mathbf{E},\mathbf{M}}(\mathbf{v}) \; \overset{\triangle}{=} \; \exists \mathbf{E}, \mathbf{M}\; \mathbf{v} = \mathbf{E} \circ \kappa(\mathbf{M}). \tag{76}$$

Once more we can add optional lower and upper bounds $\mathbf{M}^-_j$ and $\mathbf{M}^+_j$ on metabolite concentrations and/or lower and upper bounds $\mathbf{E}^-_i$ and $\mathbf{E}^+_i$ on enzyme concentrations if they are known, and also a global enzymatic resource constraint, which is often considered, as $\mathbf{c}^T\mathbf{E} \leq W$, where $\mathbf{c}$ is a constant positive vector of size $r$ and $W$ a positive constant:

$$\mathbf{KC}^b(\mathbf{v}) \; \overset{\triangle}{=} \; \exists \mathbf{E}, \mathbf{M}\; (\mathbf{v} = \mathbf{E} \circ \kappa(\mathbf{M}) \wedge \mathbf{c}^T\mathbf{E} \leq W \wedge \mathbf{E}^- \leq \mathbf{E} \leq \mathbf{E}^+ \wedge \mathbf{M}^- \leq \mathbf{M} \leq \mathbf{M}^+). \tag{77}$$

We can also consider intermediate constraints, existentially quantified only on metabolite concentrations if enzyme concentrations are known:

$$\mathbf{KC}_{\mathbf{E}}(\mathbf{v}) \; \overset{\triangle}{=} \; \exists \mathbf{M}\; \mathbf{KC}_{\mathbf{E},\mathbf{M}}(\mathbf{v}) \tag{78}$$

possibly with bounds on metabolite concentrations in the quantification:

$$\mathbf{KC}^b_{\mathbf{E}}(\mathbf{v}) \; \overset{\triangle}{=} \; \exists \mathbf{M}\; (\mathbf{KC}_{\mathbf{E},\mathbf{M}}(\mathbf{v}) \wedge \mathbf{M}^- \leq \mathbf{M} \leq \mathbf{M}^+) \tag{79}$$

or only on enzyme concentrations if metabolite concentrations are known:

$$\mathbf{KC}_{\mathbf{M}}(\mathbf{v}) \; \overset{\triangle}{=} \; \exists \mathbf{E}\; \mathbf{KC}_{\mathbf{E},\mathbf{M}}(\mathbf{v}) \tag{80}$$

possibly with enzymatic resource constraint and bounds on enzyme concentrations in the quantification:

$$\mathbf{KC}^b_{\mathbf{M}}(\mathbf{v}) \; \overset{\triangle}{=} \; \exists \mathbf{E}\; (\mathbf{KC}_{\mathbf{E},\mathbf{M}}(\mathbf{v}) \wedge \mathbf{c}^T\mathbf{E} \leq W \wedge \mathbf{E}^- \leq \mathbf{E} \leq \mathbf{E}^+). \tag{81}$$

The solution space in $\mathbb{R}^r$ of the constraint $\mathbf{KC}_{\mathbf{M}}$ is $Sol_{\mathbf{KC}_{\mathbf{M}}} = \bigcup_{\mathbf{E}} Sol_{\mathbf{KC}_{\mathbf{E},\mathbf{M}}} = \bigcup_{\mathbf{E} \in \mathbb{R}^r_+} \{\mathbf{E} \circ \kappa(\mathbf{M})\} = \{\mathbf{v} \in \mathbb{R}^r \mid sign(\mathbf{v}) \leq sign(\kappa(\mathbf{M}))\} = \{\mathbf{v} \in \mathbb{R}^r \mid sign(\mathbf{v}) \leq \mathbf{ts}_{\mathbf{M}}\} = Sol_{\mathbf{TC}_{\mathbf{M}}}$, as the sign of $\kappa(\mathbf{M})$ is the thermodynamic sign vector $\mathbf{ts}_{\mathbf{M}}$. It is thus the same as the solution space of the thermodynamic constraint $\mathbf{TC}_{\mathbf{M}}$ and is the closed orthant $O_{\mathbf{ts}_{\mathbf{M}}}$. It follows that the solution space of the constraint $\mathbf{KC}$, given by $Sol_{\mathbf{KC}} = \bigcup_{\mathbf{M}} Sol_{\mathbf{KC}_{\mathbf{M}}} = \bigcup_{\mathbf{M}} Sol_{\mathbf{TC}_{\mathbf{M}}} = Sol_{\mathbf{TC}}$, is the same as the solution space of the thermodynamic constraint $\mathbf{TC}$ and is the finite union of

the closed orthants $O_{\mathbf{ts_M}}$. Similarly, $Sol_{\mathbf{KC}^b} = Sol_{\mathbf{TC}^b}$ if the bounds are only on the metabolite concentrations, i.e., there are no bounds on enzyme concentrations. From $Sol_{\mathbf{KC}^b_{\mathbf{M}}} = \bigcup_{\mathbf{E}^- \leq \mathbf{E} \leq \mathbf{E}^+ \wedge \mathbf{c}^T \mathbf{E} \leq W} \{\mathbf{E} \circ \boldsymbol{\kappa}(\mathbf{M})\}$, it follows that the flux vectors in $Sol_{\mathbf{KC}^b_{\mathbf{M}}}$ are defined by the linear inequalities: $\kappa_i(\mathbf{M})\mathbf{E}^-_i \leq \mathbf{v}_i \leq \kappa_i(\mathbf{M})\mathbf{E}^+_i$ for those $i$'s such that $\kappa_i(\mathbf{M}) > 0$, $\kappa_i(\mathbf{M})\mathbf{E}^+_i \leq \mathbf{v}_i \leq \kappa_i(\mathbf{M})\mathbf{E}^-_i$ for those $i$'s such that $\kappa_i(\mathbf{M}) < 0$, $\mathbf{v}_i = 0$ for those $i$'s such that $\kappa_i(\mathbf{M}) = 0$ and $\sum_{i|\kappa_i(\mathbf{M}) \neq 0} \mathbf{c}_i \kappa_i(\mathbf{M})^{-1} \mathbf{v}_i \leq W - \sum_{i|\kappa_i(\mathbf{M})=0} \mathbf{c}_i \mathbf{E}^-_i$. Thus $Sol_{\mathbf{KC}^b_{\mathbf{M}}}$ is a convex polyhedron (a parallelepiped contained in the closed orthant $O_{\mathbf{ts_M}}$, truncated by a hyperplane). Now, if $\mathbf{E}^- = \mathbf{0}$, i.e., in the absence of positive lower bounds on enzyme concentrations, this polyhedron has $\mathbf{0}$ as a vertex and is nothing else than $Sol_{\mathbf{KC_M}}$, i.e., the closed orthant $O_{\mathbf{ts_M}}$, truncated as a parallelepiped by the faces given by $\mathbf{v}_i = \kappa_i(\mathbf{M})\mathbf{E}^+_i$ for those $i$'s such that $\kappa_i(\mathbf{M}) \neq 0$ and by the hyperplane $\sum_{i|\kappa_i(\mathbf{M}) \neq 0} \mathbf{c}_i \kappa_i(\mathbf{M})^{-1} \mathbf{v}_i = W$. Consequently, $Sol_{\mathbf{KC}^b} = \bigcup_{\mathbf{M}} Sol_{\mathbf{KC}^b_{\mathbf{M}}}$ is equal to a certain truncation of $Sol_{\mathbf{KC}} = Sol_{\mathbf{TC}}$, defined as the finite union of truncations of the closed orthants $O_{\mathbf{ts_M}}$ (each $O_{\mathbf{ts}}$ becoming a parallelepiped, after being truncated by hyperplanes according to equations $\mathbf{v}_i = sup(\kappa_i(\mathbf{M}))\mathbf{E}^+_i$ if $\mathbf{ts}_i = +$ (resp., $\mathbf{v}_i = inf(\kappa_i(\mathbf{M}))\mathbf{E}^+_i$ if $\mathbf{ts}_i = -$), where $sup$ (resp., $inf$) applies to those $\mathbf{M}$ such that $sign(\boldsymbol{\kappa}(\mathbf{M})) = \mathbf{ts}$, which gives a polyhedron, but also, in the presence of an enzymatic resource constraint, by an algebraic, nonlinear, surface, which gives in this case a local solution space in $O_{\mathbf{ts}}$ that is no longer a polyhedron and is not necessarily convex).

**Proposition 5.** *Given metabolite concentrations $\mathbf{M}$, the kinetic constraint $KC_{\mathbf{M}}$ is identical to the thermodynamic constraint $TC_{\mathbf{M}}$ and thus the set $Sol_{KC_{\mathbf{M}}}$ of vectors in $\mathbb{R}^r$ satisfying $KC_{\mathbf{M}}$ is the closed orthant defined by the thermodynamic sign vector $\mathbf{ts_M}$. The kinetic constraint $KC$ is identical to the thermodynamic constraint $TC$ and thus the set $Sol_{KC}$ of vectors in $\mathbb{R}^r$ satisfying $KC$ is a union of closed orthants. In the presence of bounds only on metabolite concentrations (and not on enzyme concentrations), the kinetic constraint $KC^b$ is identical to the thermodynamic constraint $TC^b$ and thus the set $Sol_{KC^b}$ of vectors in $\mathbb{R}^r$ satisfying $KC^b$ is a union of closed orthants. Therefore, these kinetic constraints boil down to thermodynamic constraints and the results regarding the geometrical structure of the corresponding spaces of flux vectors and the characterization of elementary flux vectors (or elementary flux modes) given by Propositions 1–4 apply: in particular, $CFC_{KC}(\mathbf{M}) = CFC_{TC}(\mathbf{M})$ is a flux cone and $CFC_{KC} = CFC_{TC}$ and $CFC_{KC^b} = CFC_{TC^b}$ are finite unions of flux cones. For a given $\mathbf{M}$ and in the presence of bounds on enzyme concentrations, the set of vectors in $\mathbb{R}^r$ satisfying $KC^b_{\mathbf{M}}$ is a convex polyhedron and $CFC_{KC^b}(\mathbf{M})$ is thus a flux polyhedron. In the particular case where positive lower bounds on enzyme concentrations are absent, $CFC_{KC^b}(\mathbf{M})$ is just the parallelepiped obtained by truncating the flux cone $CFC_{KC}(\mathbf{M}) = CFC_{TC}(\mathbf{M})$ by hyperplanes originating from the upper bounds on enzyme concentrations and by a hyperplane originating from the enzymatic resource constraint and coincides thus with the said flux cone in a certain adequate neighborhood of $\mathbf{0}$, i.e., for sufficiently small values of the fluxes. In this case, $CFC_{KC^b}$ is thus a truncation (by an algebraic surface) of $CFC_{TC}$ and coincides with this union of flux cones in a certain adequate neighborhood of $\mathbf{0}$ and the characterization of elementary flux vectors remains valid in this neighborhood. These results extend in the presence of additional inhomogeneous linear constraints on flux vectors given by $\mathbf{Gv} \geq \mathbf{h}$ by intersecting the solution spaces above with the polyhedron defined by $\mathbf{Gv} \geq \mathbf{h}$, giving rise to flux polyhedra (results in a neighborhood of $\mathbf{0}$ holding only if $\mathbf{h} < \mathbf{0}$).*

However, the geometric structure of $Sol_{\mathbf{KC_E}}$, of $Sol_{\mathbf{KC}^b_E}$ and of $Sol_{\mathbf{KC}^b}$ in the presence of positive lower bounds on enzyme concentrations, depends on the kinetic function $\boldsymbol{\kappa}(\mathbf{M})$ and $CFC_{\mathbf{KC}}(\mathbf{E})$, $CFC_{\mathbf{KC}^b}(\mathbf{E})$ and $CFC_{\mathbf{KC}^b}$ are generally neither polyhedra nor convex.

Proposition 5 has important consequences on constrained enzyme allocation problems in kinetic metabolic networks. Considering a kinetic metabolic network, with possible bounds on metabolite concentrations, but not on enzyme concentrations, i.e., with solution space $CFC_{\mathbf{KC}} = CFC_{\mathbf{TC}}$, or $CFC_{\mathbf{KC}^b} = CFC_{\mathbf{TC}^b}$, which is a finite union, for $\mathbf{M}$ varying, of the flux cones $FC_{\leq \mathbf{ts_M}}$, the generic enzyme allocation problem consists in maximizing the specific flux (or specific rate, i.e., rate expressed per unit of biomass amount) of a

given reaction, say $k$, defined as $\mathbf{v}_k / E_T$, where $\mathbf{v}$ is a flux vector in $CFC_{\mathbf{KC}}$ or $CFC_{\mathbf{KC}^b}$ and $E_T$ denotes the total protein content in the system. $E_T$ is expressed in all generality as a fixed weighted sum of the enzyme concentrations, $E_T = \sum_{i=1}^{r} w_i \mathbf{E}_i$ (the $w_i$'s being given positive weights that denote the fraction of the resource used per unit of enzyme), able to encode different enzymatic constraints (such as limited cellular or membrane surface space) as well as other constraints regarding the abundance of certain enzymes. Likewise, the steady-state flux component $\mathbf{v}_k > 0$ may stand for diverse metabolic processes, ranging from the synthesis rate of a particular product within a specific pathway to the rate of overall cellular growth. The formation rate of a metabolic product expressed per gram of biomass and the specific growth rate of a cell are both examples of such specific rates. We look for maximization in the solution space by varying the metabolite concentrations (inside their bounds, if any) and the enzyme concentrations (without bounds), which gives rise to a complex non-convex optimization problem. Now, maximizing $\mathbf{v}_k / E_T$ is equivalent to fixing the rate $\mathbf{v}_k$ to a positive value, e.g., to 1, and minimizing the $E_T$ needed to attain this level of $\mathbf{v}_k$. This means minimizing the function $\sum_{i=1}^{r} (w_i / \kappa_i(\mathbf{M})) \mathbf{v}_i$ by varying $\mathbf{M}$ (with possible bounds) and, for each given $\mathbf{M}$, the flux vector $\mathbf{v}$ in $FP_{\mathbf{M}} = FC_{\leq \mathbf{ts_M}} \cap \{\mathbf{v} \mid \mathbf{v}_k = 1\}$ (without bounds on $\mathbf{v}$ as we assume there are no bounds on enzyme concentrations). If all $FP_{\mathbf{M}}$'s are empty, the problem is unsolvable, i.e., $\mathbf{v}_k > 0$ is incompatible with the kinetics. Otherwise, i.e., when the problem is solvable, we consider successively each nonempty $FP_{\mathbf{M}}$. Such an $FP_{\mathbf{M}}$ is a flux polyhedron whose elementary flux points (which are equal to the extreme points or vertices) correspond to the intersections of the hyperplane $\{\mathbf{v} \mid \mathbf{v}_k = 1\}$ with extreme rays (edges) of $FC_{\leq \mathbf{ts_M}}$, i.e., EFMs of $CFC_{\mathbf{KC}}$ or $CFC_{\mathbf{KC}^b}$, and whose elementary vectors (equal to extreme vectors), if any, correspond to the extreme rays of $FC_{\leq \mathbf{ts_M}} \cap \{\mathbf{v} \mid \mathbf{v}_k = 0\}$. As, for $\mathbf{M}$ fixed, the function to minimize is linear in $\mathbf{v}$, its minimum on $FP_{\mathbf{M}}$ is reached on a lower-dimensional face of $FP_{\mathbf{M}}$ (as $\mathbf{v}_i / \kappa_i(\mathbf{M}) \geq 0$, $w_i > 0$ and $FP_{\mathbf{M}}$ is included in a closed orthant, this face is necessarily a convex hull of certain extreme points of $FP_{\mathbf{M}}$ even if it is not a polytope, i.e., no extreme vector may occur as one of the generators of this face), and thus reached in particular at at least one extreme point, i.e., at an EFM. Now, as the total number of EFMs is finite, so is the number of those for which the minimum of the function occurs for any given $\mathbf{M}$, considering all nonempty $FP_{\mathbf{M}}$'s. Therefore, when $\mathbf{M}$ varies in its domain, we obtain the result that the maximum of the specific flux $\mathbf{v}_k / E_T$ occurs (at least) at an EFM of $CFC_{\mathbf{KC}}$ or $CFC_{\mathbf{KC}^b}$. In the case of $CFC_{\mathbf{KC}^b}$ and assuming the $\kappa_i(\mathbf{M})$'s are continuous, this maximum is attained at an EFM at finite metabolite concentrations as $\mathbf{M}$ then varies in the compact set $\prod_j [\mathbf{M}_j^-, \mathbf{M}_j^+]$. In the case of $CFC_{\mathbf{KC}}$, the maximum might not be attained at finite metabolite concentrations. This is the result already obtained in [40,41] and we followed a similar proof, but relying this time on a precise characterization of the solution space $CFC_{\mathbf{KC}}$ or $CFC_{\mathbf{KC}^b}$ and of the EFMs given by the Proposition 5, which was not the case in the above-quoted works. Finally, the enzyme allocation problem can theoretically be solved by computing all the thermodynamically feasible EFMs having $k$ in their support (i.e., satisfying the Boolean constraint $Bc = k$, see next subsection), say $\{\mathbf{e}^l\}$ (all components of which are fixed by $\mathbf{e}_k^l = 1$), and, for each one, by computing the minimum (if it exists) of $\sum_{i \in supp(\mathbf{e}^l)} (w_i \mathbf{e}_i^l) / \kappa_i(\mathbf{M})$ for $\mathbf{M}$ varying in its domain, such that $sign(\kappa_i(\mathbf{M})) = sign(\mathbf{e}_i^l)$ for all $i \in supp(\mathbf{e}^l)$, which is a much simpler optimization problem than the initial one. The global minimum, if it exists, is then the smallest of these local minima, for all $\mathbf{e}^l$'s. We then obtain the maximum value of the specific flux $\mathbf{v}_k / E_T$, an EFM where this maximum occurs and the values of the metabolite concentrations for which it occurs.

**Proposition 6.** *Given a kinetic metabolic network with possible bounds on metabolite concentrations, but not on enzyme concentrations, i.e., with solution space $CFC_{KC}$ or $CFC_{KC^b}$, if the enzyme allocation problem, which consists in maximizing the specific flux (rate per unit of biomass amount) in a given reaction $k$, i.e., $v_k / E_T$, where $v$ is a flux vector in $CFC_{KC}$ or $CFC_{KC^b}$ and $E_T = \sum_{i=1}^{r} w_i E_i$ is the total protein content in the system (the $w_i$'s being fixed positive weights),*

has an optimal solution (which is always the case for $CFC_{KC^b}$ if the problem is solvable), then this solution is reached in particular at an EFM of $CFC_{KC}$ or $\mathring{CFC}_{KC^b}$.

2.2.3. Application to Regulatory Constraints

From Lemma 2, we obtain that $CFC_{RC_{Bc}}$ (resp., $CFP_{RC_{Bc}}$) is the disjoint union of $FC \cap \mathring{O}$'s (resp., $FP \cap \mathring{O}$'s), for certain open orthants $\mathring{O}$ and, after merging, the disjoint union of $FC \cap O^\circ$'s (resp., $FP \cap O^\circ$'s), for certain semi-open orthants $O^\circ$ (orthants without some of their faces), without any further merging possible.

We will call open polyhedral cone (resp., open polyhedron) in dimension $r > 0$ an $r$-polyhedral cone $C$ (resp., $r$-polyhedron $P$) without its facets, and we will note it $\mathring{C}$ (resp., $\mathring{P}$) as it coincides with the topological interior of $C$ (resp., $P$) in the affine span of $C$ (resp., $P$). Reciprocally, $C$ (resp., $P$) is the topological closure of $\mathring{C}$ (resp., $\mathring{P}$) in $\mathbb{R}^r$. $\mathring{C}$ (resp., $\mathring{P}$) is defined as in Equation (7) (resp., in Equation (26)) but with strict inequalities (and equalities for defining its affine space). We will in particular consider open faces of a cone $C$ or polyhedron $P$ as $\mathring{F}$ for $F$ a face of $C$ or $P$. $C$ (resp., $P$) is the disjoint union of its open faces (by convention, a face of dimension 0, i.e., a vertex, is equal to its open face).

It follows from these definitions that, for any closed $r$-orthant $O$ and any open orthant $\mathring{O}' \subset O$, $FC \cap \mathring{O}'$ (resp., $FP \cap \mathring{O}'$) is an open face of $FC \cap O$ (resp., a disjoint union of open faces of $FP \cap O$, as we have to keep faces corresponding to $\mathbf{G v} = \mathbf{h}$). In all, we obtain that $CFC_{RC_{Bc}} \cap O$ (resp., $CFP_{RC_{Bc}} \cap O$) is a disjoint union of open faces of the flux cone $FC \cap O$ (resp., the flux polyhedron $FP \cap O$) or, equivalently, that, for any flux tope $FC_{\leq \eta}$ for $FC$ (resp., $FP_{\leq \eta}$ for $FP$), $CFC_{RC_{Bc} \leq \eta}$ (resp., $CFP_{RC_{Bc} \leq \eta}$) is a disjoint union of open faces of $FC_{\leq \eta}$ (resp., $FP_{\leq \eta}$). As we grouped together open orthants into semi-open orthants in Lemma 2, we can also group together with such an open face $\mathring{F}$ all those other open faces $\mathring{F}'$ in question where $F'$ is a face of $F$ to obtain thus a (minimal) disjoint union of semi-open polyhedral cones (resp., semi-open polyhedra). Here, we call semi-open polyhedral cone $C^\circ$ (resp., semi-open polyhedron $P^\circ$) a polyhedral cone $C$ (resp., polyhedron $P$) without certain (between zero and all) of its faces of lesser dimension, that can be thus expressed as a disjoint union of certain (between all and only $\mathring{C}$, resp., $\mathring{P}$) of the open faces of $C$ (resp., $P$). We have: $\mathring{C} \subseteq C^\circ \subseteq C$ (resp., $\mathring{P} \subseteq P^\circ \subseteq P$).

**Proposition 7.** *Given an arbitrary Boolean constraint Bc, the solution space $CFC_{RC_{Bc}}$ (resp., $CFP_{RC_{Bc}}$) for the regulatory constraint $\mathbf{RC}_{Bc}$ is a finite disjoint union of open polyhedral cones (resp., open polyhedra), which are certain open faces of all the flux topes $FC_{\leq \eta}$ (resp., $FP_{\leq \eta}$). Grouping together certain of these open faces according to Lemma 2, we obtain a disjoint union of semi-open (i.e., without certain of their faces of lesser dimension) faces of the $FC_{\leq \eta}$'s (resp., $FP_{\leq \eta}$'s), without any possible further merging of two of them to make up a bigger semi-open face.*

Note that the rules presented in the proof of Lemma 2 to group together open faces into semi-open faces are applied globally to all flux topes. If we choose to apply them separately for each flux tope, then the union of semi-open faces obtained is no longer disjoint in general as two such semi-open faces for two different flux topes may have a nonempty intersection (as $CFC_{RC_{Bc}} \cap O$ (resp., $CFP_{RC_{Bc}} \cap O$) and $CFC_{RC_{Bc}} \cap O'$ (resp., $CFP_{RC_{Bc}} \cap O'$), for two different closed $r$-orthants $O$ and $O'$ may have open faces in common). Note also that, even after this merging, there may exist a strict inclusion relationship between the closures of two semi-open faces.

From this geometrical structure of the solution space, we will now deduce what its EFMs and its EFVs (or EFPs) are. Let us begin with a preliminary remark. We know that, by definition of EFVs, we have, for a flux cone $FC$, EFVs $(FC) = \bigcup_{\eta \text{ maximal in } sign(FC)}$ EFVs $(FC_{\leq \eta})$ (and idem for a flux polyhedron $FP$ with EFPs and EFVs), which remains true for any constrained flux cone subset $CFC_{\mathbf{C}}$ (or constrained flux polyhedron subset $CFP_{\mathbf{C}}$) whatever the biological constraint $\mathbf{C}$ is. We saw that this equality was also satisfied by EFMs for $FC$: EFMs $(FC) = \bigcup_{\eta \text{ maximal in } sign(FC)}$ EFMs $(FC_{\leq \eta})$ (and idem for $FP$) and remained true for $CFC_{\mathbf{C}}$ for $\mathbf{C}$ any thermodynamic constraint or any kinetic constraint of the form

**KC$_\mathbf{M}$**, **KC** or **KC$^b$** (in the absence of bounds on enzyme concentrations), thus allowing in these cases to decompose the identification of EFMs flux tope by flux tope, as for EFVs. However this is no longer true for regulatory constraints. Actually, counter-examples can be found in the presence of reversible reactions, used in a given direction in an EFM local to a certain flux tope and in the other direction in an EFM local to another flux tope, choosing the Boolean constraint such that it imposes a strict set inclusion between the supports of these two local EFMs.

**Example 1.** *Consider the simple network comprising one reaction $R : A \to B$, where $A$ and $B$ are the two internal metabolites, and four transfer reactions $T1 : \to A$, $T2 : \to B$, $T3 : A \to$, $T4 : B \to$, and assume that $R$ and $T1$ are reversible (we will note par rev() the reversed reaction), the three other reactions being irreversible (see Figure 4). It is obvious that $e^1 = (1,1,0,0,1)^T$, made up of $T1, R, T4$, $e^4 = (-1,-1,1,0,0)^T$, made up of $T2, rev(R), rev(T1)$, and $e^3 = (0,0,1,0,1)^T$, made up of $T2, T4$, are EFMs of FC. Consider now the Boolean constraint $Bc = T1 \wedge T4$. Then, $e^1$ is an EFM of $CFC_{RC_{Bc}}$, belonging to $CFC_{RC_{Bc} \leq (+,+,+,+,+)^T}$. Moreover, any $e = \lambda e^3 + e^4$ with $\lambda > 0$ is an EFM of $CFC_{RC_{Bc} \leq (-,-,+,+,+)^T}$ (obviously, the sub-pathways $e^3$ and $e^4$ of $e$ do not belong to $CFC_{RC_{Bc}}$). As $supp(e^1) = \{R, T1, T4\}$ and $supp(e) = \{R, T1, T2, T4\}$, we have $supp(e^1) \subset supp(e)$ and thus $e$ is not an EFM of $CFC_{RC_{Bc}}$.*

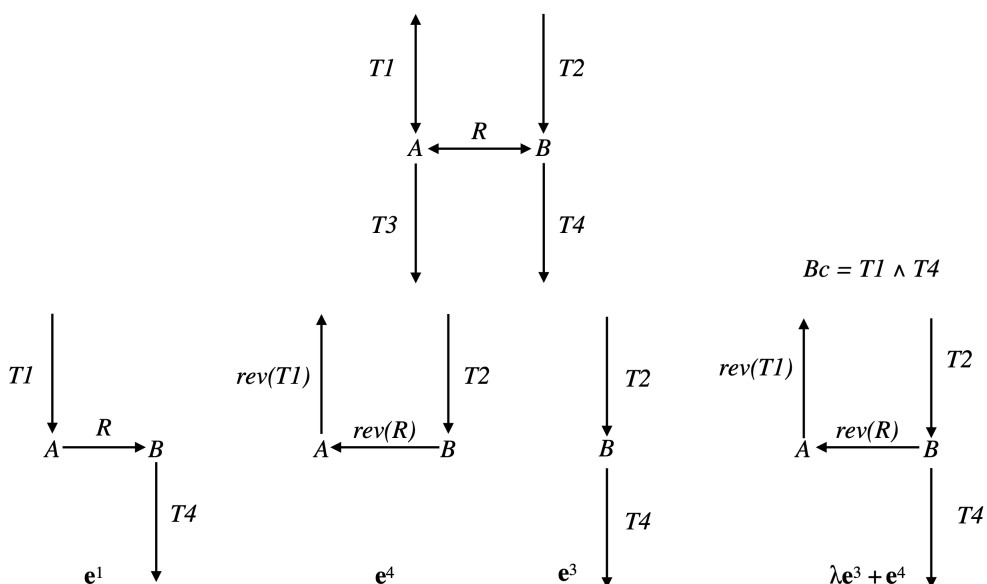

**Figure 4.** Simple network of Example 1 with $R$ and $T1$ reversible. $\mathbf{e}^1, \mathbf{e}^4, \mathbf{e}^3$ are EFMs. When we add the regulatory constraint $T1 \wedge T4$, $\mathbf{e}^1$ is still an EFM, which is not the case of any positive combination of $\mathbf{e}^3$ and $\mathbf{e}^4$, which has a larger support. Nevertheless, the latter is an EFM in the flux tope defined by reverse fluxes in $R$ and $T1$.

Now, from a biological point of view, it is not relevant to compare supports of two pathways with a certain reaction in a given direction in the first support and in the other direction in the second support (case of $R$ and $T1$ in the example). This means that the useful concept concerning minimality is not support-minimality, but sign-minimality (exactly in the same way as, concerning decomposition, we saw that the useful concept is not non-decomposability but conform non-decomposability), which is equivalent to comparing supports separately for each closed $r$-orthant, i.e., for each flux tope. We will thus identify the EFMs flux tope by flux tope (note that this is analog to distinguishing a positive flux from a negative flux in a regulatory constraint, e.g., distinguishing the constraints $T1^+ \wedge T4$ and $T1^- \wedge T4$ in the example above, which could be done by adopting a tri-valued logic instead of a Boolean logic to represent these constraints; this is obviously done automatically when splitting each reversible reaction into two irreversible ones, where only the positive $r$-orthant has to be considered).

Therefore, we will consider in the following an arbitrary closed $r$-orthant $O$ given by a full support sign vector $\boldsymbol{\eta}$ (with $\boldsymbol{\eta}_i = +$ for $i \in \mathbf{Irr}$) and consider the part of the solution space limited to this orthant, i.e., $CFC_{\mathbf{RC}_{Bc} \leq \boldsymbol{\eta}}$ (resp., $CFP_{\mathbf{RC}_{Bc} \leq \boldsymbol{\eta}}$), thus we can limit ourselves to sign vectors $\boldsymbol{\eta}$ that are maximal in $sign(CFC_{\mathbf{RC}_{Bc}})$ (resp., $sign(CFP_{\mathbf{RC}_{Bc}})$). We saw that we could rewrite the Boolean constraint as a disjunction of two by two exclusive disjuncts, $Bc = \bigvee_k D_k$, decomposing thus the solution space $CFC_{\mathbf{RC}_{Bc} \leq \boldsymbol{\eta}}$ (resp., $CFP_{\mathbf{RC}_{Bc} \leq \boldsymbol{\eta}}$) into the disjoint solution spaces for each disjunct, $CFC_{\mathbf{RC}_{D_k} \leq \boldsymbol{\eta}}$ (resp., $CFP_{\mathbf{RC}_{D_k} \leq \boldsymbol{\eta}}$). The elementary flux vectors (i.e., faces of dimension one) of $FC_{\leq \boldsymbol{\eta}}$ that satisfy the constraint $\mathbf{RC}_{D_k}$ are obviously elementary flux vectors of $CFC_{\mathbf{RC}_{D_k} \leq \boldsymbol{\eta}}$ and the reciprocal is also true: if a flux vector of the semi-open polyhedral cone $CFC_{\mathbf{RC}_{D_k} \leq \boldsymbol{\eta}}$ is not an elementary flux vector of $FC_{\leq \boldsymbol{\eta}}$, i.e., is not a face of dimension one, then it belongs to the interior of a face of $CFC_{\mathbf{RC}_{D_k} \leq \boldsymbol{\eta}}$ of dimension at least two and is thus conformally decomposable in this face, i.e., is not elementary in $CFC_{\mathbf{RC}_{D_k} \leq \boldsymbol{\eta}}$. It follows that the elementary flux vectors of $CFC_{\mathbf{RC}_{Bc} \leq \boldsymbol{\eta}}$ are made up of all the elementary flux vectors of the $CFC_{\mathbf{RC}_{D_k} \leq \boldsymbol{\eta}}$'s. We can sum up the results regarding EFVs as:

$$\text{For } Bc = \bigvee_k D_k, \quad EFVs(CFC_{\mathbf{RC}_{Bc}}) = \bigcup_{\boldsymbol{\eta} \text{ maximal in } sign(CFC_{\mathbf{RC}_{Bc}})} EFVs(CFC_{\mathbf{RC}_{Bc} \leq \boldsymbol{\eta}})$$

$$= \bigcup_{\boldsymbol{\eta}} \bigcup_k EFVs(CFC_{\mathbf{RC}_{D_k} \leq \boldsymbol{\eta}}) = \bigcup_{\boldsymbol{\eta}} \bigcup_k EFVs(FC_{\leq \boldsymbol{\eta}}) \cap Sol_{\mathbf{RC}_{D_k}}. \tag{82}$$

This means that EFVs can be computed flux tope by flux tope and constraint-disjunct by constraint-disjunct. Moreover, the result holds also for EFPs (by considering vertices of $FP_{\leq \boldsymbol{\eta}}$) and EFVs of $CFP_{\mathbf{RC}_{Bc}}$.

However, for EFMs, we have to take care that a phenomenon similar to that described in the example above still arises and that, even in a given flux tope, an EFM of $CFC_{\mathbf{RC}_{D_k} \leq \boldsymbol{\eta}}$ is not necessarily an EFM of $CFC_{\mathbf{RC}_{Bc} \leq \boldsymbol{\eta}}$.

**Example 2.** *Consider the network comprising three irreversible reactions and one internal metabolite $A$: $R1 : \rightarrow A$, $R2 : \rightarrow A$, $R3 : A \rightarrow$, and the Boolean constraint $Bc = \neg R1 \vee R2$, decomposed as $Bc = D_1 \vee D_2$, with $D_1 = \neg R1$ and $D_2 = R1 \wedge R2$ (see Figure 5). Take $\boldsymbol{\eta} = +$. Then, $e^1 = (0, 1, 1)^T$, made up of R2, R3, which is the only EFM of $CFC_{\mathbf{RC}_{D_1} \leq \boldsymbol{\eta}}$, is also the only EFM of $CFC_{\mathbf{RC}_{Bc} \leq \boldsymbol{\eta}}$. However any $e^2 = (\lambda, 1, 1 + \lambda)^T$ with $\lambda > 0$, made up of R1, R2, R3, is an EFM of $CFC_{\mathbf{RC}_{D_2} \leq \boldsymbol{\eta}}$ and is not an EFM of $CFC_{\mathbf{RC}_{Bc} \leq \boldsymbol{\eta}}$. Note that the way the constraint is decomposed matters. With the decomposition $Bc = D'_1 \vee D'_2$ with $D'_1 = R2$ and $D'_2 = \neg R1 \wedge \neg R2$, the result for $D'_1$ is identical to that for $D_1$, but $CFC_{\mathbf{RC}_{D'_2} \leq \boldsymbol{\eta}} = \{\boldsymbol{0}\}$.*

$$Bc = \neg R1 \vee R2 = D1 \vee D2 \text{ with } D1 = \neg R1, D2 = R1 \wedge R2$$

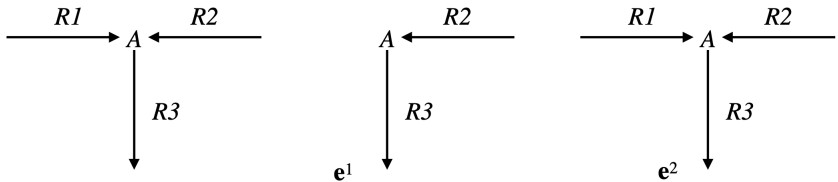

**Figure 5.** Simple network of Example 2. When we add the regulatory constraint $\neg R1 \vee R2$, $\mathbf{e}^1$ is the only EFM, and also the only one for the disjunct $D1 = \neg R1$ of the constraint, but no $\mathbf{e}^2$ combination is an EFM, while it is an EFM for the disjunct $D2 = R1 \wedge R2$.

This means that, if it is natural to study each $CFC_{\mathbf{RC}_{D_k} \leq \boldsymbol{\eta}}$ (resp., $CFP_{\mathbf{RC}_{D_k} \leq \boldsymbol{\eta}}$) separately in order to characterize the solution space and if EFVs are obtained in this way by collecting all the local EFVs, it is not the case for EFMs and, after collecting all local EFMs, we must only keep those with minimal support:

$$EFMs(CFC_{\mathbf{RC}_{Bc} \leq \boldsymbol{\eta}}) = Min_{supp-\subseteq} \{EFMs(CFC_{\mathbf{RC}_{D_k} \leq \boldsymbol{\eta}})\}_k. \tag{83}$$

**Proposition 8.** *Given an arbitrary regulatory constraint $\mathbf{RC}_{Bc}$ with $Bc = \bigvee_k D_k$, where the disjuncts $D_k$ are taken two by two inconsistent, and the associated constrained flux cone subset $CFC_{\mathbf{RC}_{Bc}}$ (resp., constrained flux polyhedron subset $CFP_{\mathbf{RC}_{Bc}}$), its EFVs (resp., EFPs and EFVs) are obtained by collecting these for each flux tope (i.e., in each r-orthant $O_{\leq \eta}$) and for each disjunct, i.e., for each $CFC_{\mathbf{RC}_{D_k} \leq \eta}$ (resp., $CFP_{\mathbf{RC}_{D_k} \leq \eta}$), and are nothing else than the EFVs (resp., EFPs and EFVs) of FC (resp., FP) satisfying the constraint $\mathbf{RC}_{Bc}$. This is not the case for EFMs. First, an EFM of $CFC_{\mathbf{RC}_{Bc} \leq \eta}$ is not necessarily an EFM of $CFC_{\mathbf{RC}_{Bc}}$, but actually the biologically relevant minimality concept being sign-minimality and not support-minimality, we will consider EFMs for each flux tope $CFC_{\mathbf{RC}_{Bc} \leq \eta}$. Second, an EFM of $CFC_{\mathbf{RC}_{D_k} \leq \eta}$ is not necessarily an EFM of $CFC_{\mathbf{RC}_{Bc} \leq \eta}$. EFMs of $CFC_{\mathbf{RC}_{Bc} \leq \eta}$ are actually obtained by collecting EFMs of $CFC_{\mathbf{RC}_{D_k} \leq \eta}$ for all k and keeping only those with minimal support.*

This being said, we will now focus on an arbitrary disjunct $D$ of the form $\bigwedge_{i \in I} \mathbf{v}_i \wedge \bigwedge_{j \in J} \neg \mathbf{v}_j$ with $I, J \subseteq \{1, \ldots, r\}, I \cap J = \varnothing$. Thus, $CFC_{\mathbf{RC}_D \leq \eta}$ (resp., $CFP_{\mathbf{RC}_D \leq \eta}$) is the semi-open face $F^\circ$ of $FC_{\leq \eta}$ (resp., $FP_{\leq \eta}$), obtained from the face $F$ defined by $\{\mathbf{v}_j = 0, j \in J\}$ (i.e., $F = FC_{\leq \eta} \cap \bigcap_{j \in J} \{\mathbf{v}_j = 0\}$, idem with $FP$) by removing its facets $\{\mathbf{v}_i = 0\}$ for all $i \in I$. Let's note EFMs $^{\mathbf{RC}_D} \triangleq$ EFMs $(FC_{\leq \eta}) \cap Sol_{\mathbf{RC}_D}$ those EFMs of $FC_{\leq \eta}$ that satisfy the constraint $\mathbf{RC}_D$ and EFMs $_{\mathbf{RC}_D} \triangleq$ EFMs $(CFC_{\mathbf{RC}_D \leq \eta})$ the EFMs of the part of the solution space in $O_\eta$. Obviously, EFMs $^{\mathbf{RC}_D} \subseteq$ EFMs $_{\mathbf{RC}_D}$. If $I = \varnothing$, $F^\circ = F$ and EFMs $^{\mathbf{RC}_D} =$ EFMs $_{\mathbf{RC}_D}$, so we will consider the case $I \neq \varnothing$. In this case, and contrary to what happens for thermodynamic and kinetic (as described in Proposition 5) constraints, there is generally no longer identity between EFMs $^{\mathbf{RC}_D}$ and EFMs $_{\mathbf{RC}_D}$ [42].

**Example 3.** *Consider the network of Example 1 (thus $D = T1 \wedge T4$) and let $\eta = +$ and $e^2 = (0, 1, 0, 1, 0)^T$ be the EFM of $FC_{\leq \eta}$ made up of T1 and T3. Then, for any $\lambda > 0$, $e^2 + \lambda e^3$, positive conformal combination of the two EFMs $e^2$ and $e^3$ of $FC_{\leq \eta}$, has support $\{T1, T2, T3, T4\}$ and belongs to $EFMs_{\mathbf{RC}_D} \backslash EFMs^{\mathbf{RC}_D}$ (see Figure 6).*

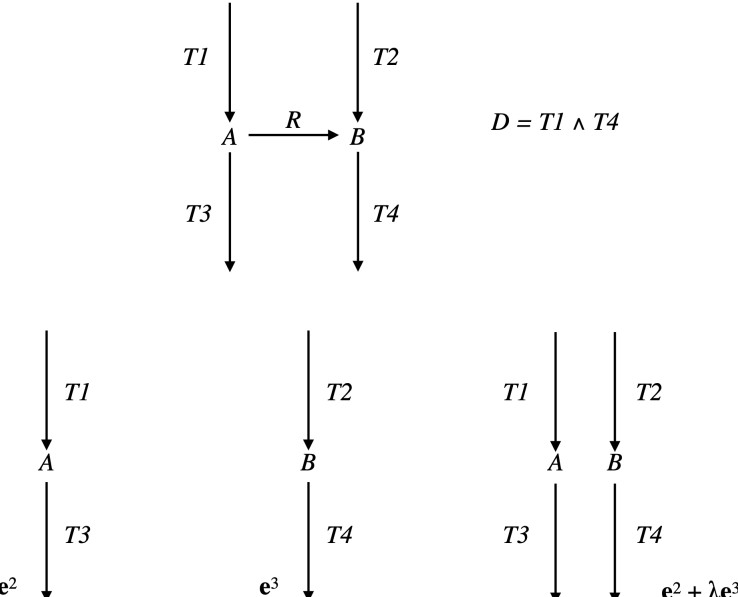

**Figure 6.** Simple network of Example 3. $\mathbf{e}^2$ and $\mathbf{e}^3$ are EFMs. When we add the regulatory constraint $T1 \wedge T4$, any positive combination of $\mathbf{e}^2$ and $\mathbf{e}^3$ is now an EFM, that is not obtained as an original EFM satisfying the constraint.

EFMs $^{\mathbf{RC}_D}$ correspond to the faces of dimension one (edges or extreme rays), if any, of the semi-open face $F^\circ$, i.e., the edges of the face $F$ that have not been removed, which means

that they are not included in any hyperplane of equation $\mathbf{v}_i = 0$, for a certain $i \in I$, or equivalently that their supports contain $I$. Now, consider a face $F'$ of $F$, of dimension at least two, such that $\mathring{F}' \subseteq F^\circ$ but no facet of $F'$ has its interior included in $F^\circ$ (i.e., any facet of $F'$ is included in an hyperplane of equation $\mathbf{v}_i = 0$, for a certain $i \in I$), if any. Note that several such $F'$ may exist, but none can be included in another one, i.e., they are not comparable for inclusion in the lattice of the faces of $F$. Then, all vectors of $\mathring{F}'$ have the same minimal support, i.e., $\mathring{F}' \subseteq \text{EFMs}_{\mathbf{RC}_D} \setminus \text{EFMs}^{\mathbf{RC}_D}$. If $\{\mathbf{e}^k\}_{k \in K}$ (with $|K| \geq 2$) are representatives of the extreme vectors of $F'$, we have $F' = cone_\oplus(\{\mathbf{e}^k\})$ (which is actually the same as $cone(\{\mathbf{e}^k\})$ as we are in orthant $O_{\leq \mathbf{\eta}}$) and thus $\mathring{F}' = \{\bigoplus_{k \in K} \beta_k \mathbf{e}^k \mid \beta_k > 0\} \overset{\triangle}{=} cone_\oplus^+(\{\mathbf{e}^k\})$ and the common minimal support of all vectors of $\mathring{F}'$ is $supp(\mathring{F}') = \bigcup_{k \in K} supp(\mathbf{e}^k)$. Conversely, if $\mathbf{v} \in \text{EFMs}_{\mathbf{RC}_D}$, let $F'$ be the minimal face of $F$ containing $\mathbf{v}$. If $F'$ has dimension one (extreme ray), then $F' \setminus \{\mathbf{0}\} \subseteq F^\circ$ and $\mathbf{v} \in \text{EFMs}^{\mathbf{RC}_D}$. If $F'$ has dimension at least two, then no facet of $F'$ has its interior included in $F^\circ$, because if it were the case for one facet, then any vector of its interior would have its support strictly included in the support of $\mathbf{v}$, which would contradict the minimality of the latter. Finally, $\mathbf{v} \in \mathring{F}' \subseteq F^\circ$ and all vectors of $\mathring{F}'$ have the same support as $\mathbf{v}$ and belong to $\text{EFMs}_{\mathbf{RC}_D} \setminus \text{EFMs}^{\mathbf{RC}_D}$. We have thus characterized both $\text{EFMs}^{\mathbf{RC}_D}$ and $\text{EFMs}_{\mathbf{RC}_D} \setminus \text{EFMs}^{\mathbf{RC}_D}$. We stipulate now, for the latter one, the decomposition of its vectors into $\mathbf{e}^k \in \text{EFMs}(F) \setminus \text{EFMs}^{\mathbf{RC}_D}$, in order to compute their supports, which is generally the only useful information (the precise decomposition into fluxes not often being very relevant). Therefore, we consider a face $F'$ of $F$, of dimension at least two, such that $\mathring{F}' \subseteq F^\circ$ but no facet of $F'$ has its interior included in $F^\circ$ and $\{\mathbf{e}^k\}_{1 \leq k \leq N}$ a minimal set of vectors in $\text{EFMs}(F')$ such that $supp(\mathring{F}') = \bigcup_{1 \leq k \leq N} supp(\mathbf{e}^k)$. Note that, for any $k$, $supp(\mathbf{e}^k) \cap I \neq \varnothing$ (and, as we have seen, $I \not\subseteq supp(\mathbf{e}^k)$), because if for a certain $k_0$ we had $supp(\mathbf{e}^{k_0}) \cap I = \varnothing$, then $\mathbf{e}^{k_0}$ would belong to all hyperplanes of equation $\mathbf{v}_i = 0$ for $i \in I$, thus to all facets of $F'$, which is impossible for a non-null vector. Let us note $S_1 = supp(\mathbf{e}^1) \cap I$ and, for any $k$, $2 \leq k \leq N$, $S_k = (supp(\mathbf{e}^k) \cap I) \setminus \bigcup_{1 \leq j \leq k-1} S_j$. Then, for any $k$, $S_k \neq \varnothing$, because if for a certain $k_0$ we had $S_{k_0} = \varnothing$, then the vectors of $cone_\oplus^+(\{\mathbf{e}^k\}_{k \neq k_0})$ would verify the constraint $\mathbf{RC}_D$ and have their supports included in $supp(\mathring{F}')$, thus equal to it as it is minimal for vectors in $CFC_{\mathbf{RC}_D \leq \mathbf{\eta}}$, which would contradict the minimality of $\{\mathbf{e}^k\}$. Finally, as by construction $I = \bigcup_{1 \leq k \leq N} S_k$ and $S_k \cap S_j = \varnothing$ for $k \neq j$, we obtain that $\{S_k\}_{1 \leq k \leq N}$ constitutes a partition of $I$ and $\{\mathbf{e}^k\}_{1 \leq k \leq N}$ is a set of vectors in $\text{EFMs}(F') \subseteq \text{EFMs}(F) \setminus \text{EFMs}^{\mathbf{RC}_D}$ such that $supp(\mathbf{e}^k) \supseteq S_k$ by construction and $supp(\mathbf{e}^k) \not\supseteq S_j$ for any $j \neq k$, otherwise, by the same reasoning as above, $\mathbf{e}^j$ could be suppressed from the set $\{\mathbf{e}^k\}$, contradicting its minimality. Finally, we obtain that the support of any vector in $\text{EFMs}_{\mathbf{RC}_D} \setminus \text{EFMs}^{\mathbf{RC}_D}$ can be written as $\bigcup_{1 \leq k \leq N} supp(\mathbf{e}^k)$, where $\{\mathbf{e}^k\}_{1 \leq k \leq N}$, $N \geq 2$, are vectors in $\text{EFMs}(F)$ verifying $supp(\mathbf{e}^k) \supseteq S_k$ and $supp(\mathbf{e}^k) \not\supseteq S_j$ for all $k$ and $j \neq k$, where $\{S_k\}_{1 \leq k \leq N}$ is a partition of $I$ (note that we have the same result for $\text{EFMs}^{\mathbf{RC}_D}$ by taking $N = 1$). Now, a given $\bigcup_{1 \leq k \leq N} supp(\mathbf{e}^k)$ is not necessarily minimal among the whole collection when we vary $N$, $\{\mathbf{e}^k\}$ and $\{S_k\}$. It is also possible that it strictly contains the support of a vector in $\text{EFMs}^{\mathbf{RC}_D}$. Therefore, to obtain exactly the supports of vectors in $\text{EFMs}_{\mathbf{RC}_D} \setminus \text{EFMs}^{\mathbf{RC}_D}$, we must only keep the minimal elements for inclusion w.r.t. the whole collection extended by $supp(\text{EFMs}^{\mathbf{RC}_D})$.

**Example 4.** *Let us continue with the network of Examples 1 and 3, so $D = T1 \wedge T4$ and $\mathbf{\eta} = +$. We have $EFMs(F) = \{e^1, e^2, e^3\}$ and $EFMs^{\mathbf{RC}_D} = \{e^1\}$ (see Figure 4). The only partition of $\{T1, T4\}$ with a size $\geq 2$ is given by: $S_1 = \{T1\}$ and $S_2 = \{T4\}$. The only vector in $EFMs(F)$ whose support contains $S_1$ and not $S_2$ is $e^2$ and the only one whose support contains $S_2$ and not $S_1$ is $e^3$. Thus, $supp(EFMs_{\mathbf{RC}_D} \setminus EFMs^{\mathbf{RC}_D}) = \{supp(e^2) \cup supp(e^3)\} = \{\{T1, T3\} \cup \{T2, T4\}\} = \{\{T1, T2, T3, T4\}\}$. Actually, we have from Example 3: $EFMs_{\mathbf{RC}_D} \setminus EFMs^{\mathbf{RC}_D} = \{e^2 + \lambda e^3 \mid \lambda > 0\}$ (see Figure 6).*

*Consider now the following network comprising two internal metabolites and seven irreversible reactions, six of which are transfer reactions, $R : A \to B$, $T1 :\to A$, $T2 : B \to$, $T3 :\to A$, $T4 : B \to$, $T5 : A \to$, $T6 :\to B$ (see Figure 7). Let $D = T1 \wedge T2$ and $\mathbf{\eta} = +$. Let $e^1 = (1, 1, 1, 0, 0, 0, 0)^T$*

made up of $T1, R, T2$, $e^2 = (1,0,0,1,1,0,0)^T$ made up of $T3, R, T4$, $e^3 = (1,1,0,0,1,0,0)^T$ made up of $T1, R, T4$, $e^4 = (1,0,1,1,0,0,0)^T$ made up of $T3, R, T2$, $e^5 = (0,1,0,0,0,1,0)^T$ made up of $T1, T5$, $e^6 = (0,0,0,1,0,1,0)^T$ made up of $T3, T5$, $e^7 = (0,0,1,0,0,0,1)^T$ made up of $T6, T2$ and $e^8 = (0,0,0,0,1,0,1)^T$ made up of $T6, T4$. We have $EFMs(F) = \{e^1, e^2, e^3, e^4, e^5, e^6, e^7, e^8\}$ and $EFMs^{RC_D} = \{e^1\}$. The only partition of $\{T1, T2\}$ with a size $\geq 2$ is given by $S_1 = \{T1\}$ and $S_2 = \{T2\}$. The vectors in $EFMs(F)$ whose support contains $S_1$ and not $S_2$ are $e^3$ and $e^5$ and those whose support contains $S_2$ and not $S_1$ are $e^4$ and $e^7$. We have $supp(e^3) \cup supp(e^4) = \{R, T1, T2, T3, T4\}$, $supp(e^3) \cup supp(e^7) = \{R, T1, T2, T4, T6\}$, $supp(e^5) \cup supp(e^4) = \{R, T1, T2, T3, T5\}$ and $supp(e^5) \cup supp(e^7) = \{T1, T2, T5, T6\}$. Each one of the four supports obtained is minimal in this collection, but the first three contain $supp(e^1) = \{R, T1, T2\}$. Thus, $supp(EFMs_{RC_D} \setminus EFMs^{RC_D}) = \{\{T1, T2, T5, T6\}\}$. Actually, $EFMs_{RC_D} \setminus EFMs^{RC_D} = \{e^5 + \lambda e^7 \mid \lambda > 0\}$.

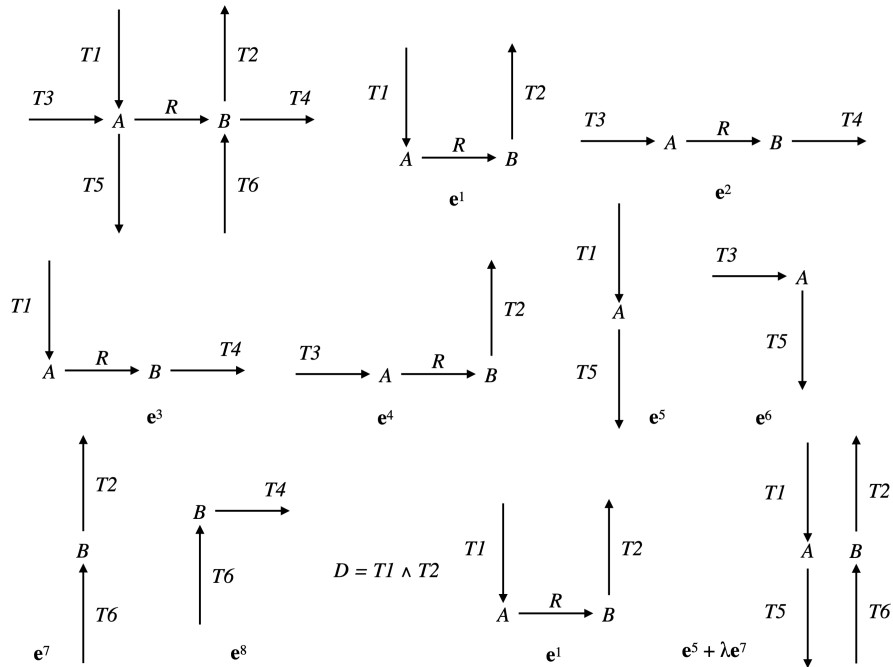

**Figure 7.** Second network of Example 4. $e^1$ to $e^8$ are all the EFMs. When we add the regulatory constraint $T1 \wedge T2$, only $e^1$ remains an EFM, with support $\{T1, T2, R\}$, but new EFMs appear, made up of any positive combination of $e^5$ and $e^7$, all with the same support $\{T1, T2, T5, T6\}$.

Finally, consider another network comprising three internal metabolites and seven irreversible reactions, five of which are transfer reactions, $R1 : B \to A$, $R2 : C \to A$, $T1 :\to B + C$, $T2 : A \to$, $T3 : A \to$, $T4 :\to B$, $T5 :\to C$. Let $D = R1 \wedge R2 \wedge T2$ and $\eta = +$ (see Figure 8). Let $e^1 = (1,1,1,2,0,0,0)^T$ made up of $T1, R1, R2, T2$, $e^2 = (1,1,1,0,2,0,0)^T$ made up of $T1, R1, R2, T3$, $e^3 = (1,0,0,1,0,1,0)^T$ made up of $T4, R1, T2$, $e^4 = (1,0,0,0,1,1,0)^T$ made up of $T4, R1, T3$, $e^5 = (0,1,0,1,0,0,1)^T$ made up of $T5, R2, T2$ and $e^6 = (0,1,0,0,1,0,1)^T$ made up of $T5, R2, T3$. We have $EFMs(F) = \{e^1, e^2, e^3, e^4, e^5, e^6\}$ and $EFMs^{RC_D} = \{e^1\}$. There are four partitions of $\{R1, R2, T2\}$ with a size $\geq 2$. The partition $\{\{R1\}, \{R2\}, \{T2\}\}$ gives nothing because there is no vector in $EFMs(F)$ whose support contains $\{T2\}$ but neither $\{R1\}$ nor $\{R2\}$. For the partition $S_1 = \{R1, T2\}$ and $S_2 = \{R2\}$, the vector in $EFMs(F)$ whose support contains $S_1$ and not $S_2$ is $e^3$ and those whose support contains $S_2$ and not $S_1$ are $e^2$, $e^5$ and $e^6$, providing $supp(e^3) \cup supp(e^2) = \{R1, R2, T1, T2, T3, T4\}$, $supp(e^3) \cup supp(e^5) = \{R1, R2, T2, T4, T5\}$ and $supp(e^3) \cup supp(e^6) = \{R1, R2, T2, T3, T4, T5\}$. For the partition $S_1 = \{R2, T2\}$ and $S_2 = \{R1\}$, the vector in $EFMs(F)$ whose support contains $S_1$ and not $S_2$ is $e^5$ and those whose support contains $S_2$ and not $S_1$ are $e^2$, $e^3$ and $e^4$, providing $supp(e^5) \cup supp(e^2) = \{R1, R2, T1, T2, T3, T5\}$, $supp(e^5) \cup supp(e^3) = \{R1, R2, T2, T4, T5\}$ and $supp(e^5) \cup supp(e^4) = \{R1, R2, T2, T3, T4, T5\}$. For the partition $S_1 = \{R1, R2\}$ and $S_2 = \{T2\}$,

*the vector in $EFMs(F)$ whose support contains $S_1$ and not $S_2$ is $e^2$ and those whose support contains $S_2$ and not $S_1$ are $e^3$ and $e^5$, providing $supp(e^2) \cup supp(e^3) = \{R1, R2, T1, T2, T3, T4\}$ and $supp(e^2) \cup supp(e^5) = \{R1, R2, T1, T2, T3, T5\}$. The minimal elements of this collection of supports are $\{R1, R2, T1, T2, T3, T4\}$, $\{R1, R2, T1, T2, T3, T5\}$ and $\{R1, R2, T2, T4, T5\}$. Minimizing also w.r.t. $supp(e^1) = \{R1, R2, T1, T2\}$ gives thus $supp(EFMs_{RC_D} \setminus EFMs^{RC_D}) = \{\{R1, R2, T2, T4, T5\}\}$. Actually, $EFMs_{RC_D} \setminus EFMs^{RC_D} = \{e^3 + \lambda e^5 \mid \lambda > 0\}$.*

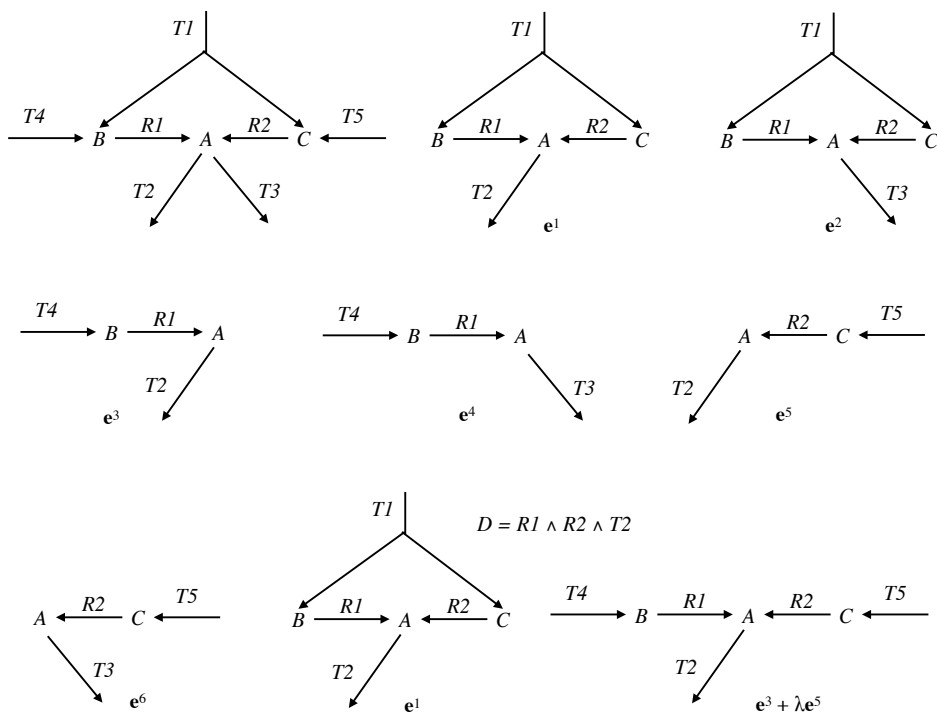

**Figure 8.** Third network of Example 4. $e^1$ to $e^6$ are all the EFMs. When we add the regulatory constraint $R1 \wedge R2 \wedge T2$, only $e^1$ remains an EFM, with support $\{T1, R1, R2, T2\}$, but new EFMs appear, made up of any positive combination of $e^3$ and $e^5$, all with the same support $\{T4, T5, R1, R2, T2\}$.

We have thus proved the following result.

**Proposition 9.** *Given a Boolean constraint $D$ as a conjunction of literals $\bigwedge_{i \in I} v_i \wedge \bigwedge_{j \in J} \neg v_j$ and an arbitrary closed r-orthant $O_{\leq \eta}$, let $CFC_{RC_D \leq \eta}$ be the constrained flux cone subset for the regulatory constraint $RC_D$ in $O_{\leq \eta}$. Let $F$ be the face of the flux cone $FC_{\leq \eta}$ defined as $FC_{\leq \eta} \cap \bigcap_{j \in J} \{v_j = 0\}$, then $CFC_{RC_D \leq \eta}$ is the semi-open flux cone $F^\circ$ obtained from $F$ by removing its facets $\{v_i = 0\}$ for all $i \in I$. Let $EFMs^{RC_D} \triangleq EFMs(FC_{\leq \eta}) \cap Sol_{RC_D}$ be the EFMs of $FC_{\leq \eta}$ that satisfy $RC_D$ and $EFMs_{RC_D} \triangleq EFMs(CFC_{RC_D \leq \eta})$ be the EFMs of $CFC_{RC_D \leq \eta}$. We get $EFMs^{RC_D} \subseteq EFMs_{RC_D}$ but, unlike the case of thermodynamic and kinetic (as in Proposition 5) constraints, there is generally no longer identity between $EFMs^{RC_D}$ and $EFMs_{RC_D}$. $EFMs^{RC_D}$ correspond to the extreme rays (faces of dimension one) of $F^\circ$, i.e., the edges of $F$ whose support contains $I$. $EFMs_{RC_D} \setminus EFMs^{RC_D}$ are the vectors of the $\mathring{F}'$'s for all $F'$ faces of $F$ of dimension at least two, such that $\mathring{F}' \subseteq F^\circ$ (i.e., $F'$ is not included in any hyperplane $\{v_i = 0\}$ with $i \in I$) but no facet of $F'$ has its interior included in $F^\circ$ (i.e., any facet of $F'$ is included in a certain hyperplane $\{v_i = 0\}$ with $i \in I$). The supports of those vectors in $EFMs_{RC_D} \setminus EFMs^{RC_D}$ are obtained as $\bigcup_{1 \leq k \leq N} supp(e^k)$, where $\{e^k\}_{1 \leq k \leq N}$, $N \geq 2$, are vectors in $EFMs(F)$ verifying $supp(e^k) \supseteq S_k$ and $supp(e^k) \not\supseteq S_j$ for all $k$ and $j \neq k$, where $\{S_k\}_{1 \leq k \leq N}$ is a partition of $I$ (they are actually the minimal elements, for subset inclusion and including the supports of $EFMs^{RC_D}$ when checking minimality, obtained like this, for any partition $\{S_k\}$ of $I$, of size at least two, and any choice of $\{e^k\}$ as above).*

This characterization of EFMs for flux cones in the presence of regulatory constraints does not hold as simply for flux polyhedra, because the added inhomogeneous linear constraints **ILC**, given by $\mathbf{Gv} \geq \mathbf{h}$, may not respect the structure of $CFC_{\mathbf{RC}_{Bc}}$ at all and may cut the interior of flux cones; we already mentioned that there is no direct relationship between EFMs and extreme or elementary fluxes for a flux polyhedron. Nevertheless, the ideas developed above for flux cones can be applied to flux polyhedra $FP_{\leq \boldsymbol{\eta}}$, where certain of their faces have to be removed due to the constraint $D$ because they are included in certain hyperplanes $\{\mathbf{v}_i = 0\}$, in order to determine EFMs $_{\mathbf{RC}_D} \setminus$ EFMs $^{\mathbf{RC}_D}$. Moreover, in practice, **ILC** is generally only used to bound (below and/or above) fluxes, in which case each extreme ray $\mathbb{R}_+\mathbf{e}$ of $FC_{\leq \boldsymbol{\eta}}$ is still partly present in $FP_{\leq \boldsymbol{\eta}}$ as for example an edge $[\alpha^-, \alpha^+]\mathbf{e}$, defined by the two vertices $\alpha^-\mathbf{e}$ and $\alpha^+\mathbf{e}$. The results above can then be transposed by using convex-conformal decomposition into vertices and conformal decomposition into elementary vectors to characterize EFMs $_{\mathbf{RC}_D} \setminus$ EFMs $^{\mathbf{RC}_D}$.

We are now interested in looking for vectors in the solution space that are (in some sense) non-decomposable while not being support-minimal, and in characterizing them, if any. We could think of EFVs (or EFPs) but, from the study above regarding EFMs, we note that, for a flux cone constrained by the regulatory constraint $D$, the vectors in EFMs $_{\mathbf{RC}_D} \setminus$ EFMs $^{\mathbf{RC}_D}$ are necessarily conformally decomposable, as they can be described as the interiors of polyhedral cones in $O_{\leq \boldsymbol{\eta}}$ of dimension at least two (for example, $\mathbf{e}^2 + \mathbf{e}^3$ in Example 3 can be decomposed as $(0.7\mathbf{e}^2 + 0.3\mathbf{e}^3) + (0.3\mathbf{e}^2 + 0.7\mathbf{e}^3)$). More straightforwardly, we can notice that EFMs $^{\mathbf{RC}_D}$ is equal to EFVs $(CFC_{\mathbf{RC}_D \leq \boldsymbol{\eta}})$, i.e., precisely the conformally non-decomposable vectors. As already pointed out, what matters more than the precise decomposition into fluxes is the decomposition of the supports (in the decomposition above of $\mathbf{e}^2 + \mathbf{e}^3$, the supports of the components are unchanged, i.e., equal to $supp(\mathbf{e}^2) \cup supp(\mathbf{e}^3) = \{T1, T2, T3, T4\}$). It follows that a relevant concept for a solution vector is not to be conformally non-decomposable but, less strictly, to be (conformally) support-wise non-decomposable, in the sense that the vector cannot be conformally decomposed into two vectors of different (necessarily not greater) supports. Now, for all faces $F'$ of $F$ of dimension at least two such that $\mathring{F}' \subseteq F°$, not covered by Proposition 9, i.e., owning at least one facet $F''$ with interior included in $F°$, it so happens that all vectors of $\mathring{F}'$ are actually support-wise decomposable, as each such vector can always be decomposed into a vector of $\mathring{F}'$ of same support and into a vector of $\mathring{F}''$ of smaller support. Therefore, in the decomposition, we do not authorize the same support for one of the component vectors. Thus the really relevant (less strict) concept is that the vector cannot be conformally decomposed into two vectors of smaller supports and we call a nonzero vector $\mathbf{x}$ of a convex polyhedral cone $C$ as (conformally) support-wise non-strictly-decomposable, if

$$
\begin{aligned}
\mathbf{x} = \mathbf{x}^1 \oplus \mathbf{x}^2, \text{ with nonzero } \mathbf{x}^1, \mathbf{x}^2 \in C, \text{ implies} \\
supp(\mathbf{x}^1) = supp(\mathbf{x}) \text{ or } supp(\mathbf{x}^2) = supp(\mathbf{x}).
\end{aligned}
\tag{84}
$$

The (conformally) support-wise non-strictly-decomposable vectors in $C$ (resp., flux vectors in a flux cone $FC$) will be noted swNSDVs (resp., swNSDFVs). Note that we could have defined support-wise non-strictly decomposability more generally without imposing a conformal decomposition, but, as already pointed out, this is not relevant for biological fluxes and we will only use this concept in a certain tope. It is obvious that, in such a tope (resp., flux tope), EMs, ExVs = EVs $\subseteq$ swNSDVs (resp., ExVs = EFVs = EFMs = swNSDFVs, so that all four definitions coincide). We will introduce a similar definition with a convex combination for a polyhedron $P$ (the previous definition and the associated relationships above being valid for its recession cone $C_P$). A vector $\mathbf{x}$ of a polyhedron $P$ is called support-wise convex(-conformally) non-strictly-decomposable, if

$$
\begin{aligned}
\mathbf{x} = \lambda\mathbf{x}^1 \oplus (1-\lambda)\mathbf{x}^2, \text{ with } \mathbf{x}^1, \mathbf{x}^2 \in P, \ 0 < \lambda < 1, \text{ implies} \\
supp(\mathbf{x}^1) = supp(\mathbf{x}) \text{ or } supp(\mathbf{x}^2) = supp(\mathbf{x}).
\end{aligned}
\tag{85}
$$

The support-wise convex(-conformally) non-strictly-decomposable vectors in $P$ (resp., flux vectors in a flux polyhedron $FP$) will be called support-wise non-strictly-decomposable points (resp., support-wise non-strictly-decomposable flux points) and noted swNSDPs (resp., swNSDFPs). It is obvious that, in any given tope (resp., flux tope), EMs, ExPs = EPs $\subseteq$ swNSDPs (resp., EFMs, ExPs = EFPs $\subseteq$ swNSDFPs).

With the notations of Proposition 9, let swNSDFVs $^{\mathbf{RC}_D}$ $\overset{\triangle}{=}$ swNSDFVs $(FC_{\leq \boldsymbol{\eta}}) \cap$ $Sol_{\mathbf{RC}_D}$ and swNSDFVs $_{\mathbf{RC}_D}$ $\overset{\triangle}{=}$ swNSDFVs $(CFC_{\mathbf{RC}_D \leq \boldsymbol{\eta}})$. We have thus EFMs $^{\mathbf{RC}_D} =$ swNSDFVs $^{\mathbf{RC}_D} \subseteq$ EFMs $_{\mathbf{RC}_D} \subseteq$ swNSDFVs $_{\mathbf{RC}_D}$ but, unlike the case of thermodynamic and kinetic (as in Proposition 5) constraints for flux cones, not only there is no longer identity between swNSDFVs $^{\mathbf{RC}_D}$ and swNSDFVs $_{\mathbf{RC}_D}$ (consequence of the non-identity between EFMs $^{\mathbf{RC}_D}$ and EFMs $_{\mathbf{RC}_D}$), but we will now see that there is generally no longer identity between EFMs $_{\mathbf{RC}_D}$ and swNSDFVs $_{\mathbf{RC}_D}$.

**Example 5.** *Continuing with the network of Example 3, $e^1 + e^3 \in swNSDFVs_{\mathbf{RC}_D} \backslash EFMs_{\mathbf{RC}_D}$. More precisely, we have (by considering only a representative for each ray) $EFMs^{\mathbf{RC}_D} = swNSDFVs^{\mathbf{RC}_D} = \{e^1\} \subset EFMs_{\mathbf{RC}_D} = EFMs^{\mathbf{RC}_D} \cup \{e^2 + \lambda e^3 \mid \lambda > 0\} \subset swNSDFVs_{\mathbf{RC}_D} = EFMs_{\mathbf{RC}_D} \cup \{e^1 + \lambda e^2 \mid \lambda > 0\} \cup \{e^1 + \lambda e^3 \mid \lambda > 0\}$ (see Figure 9).*

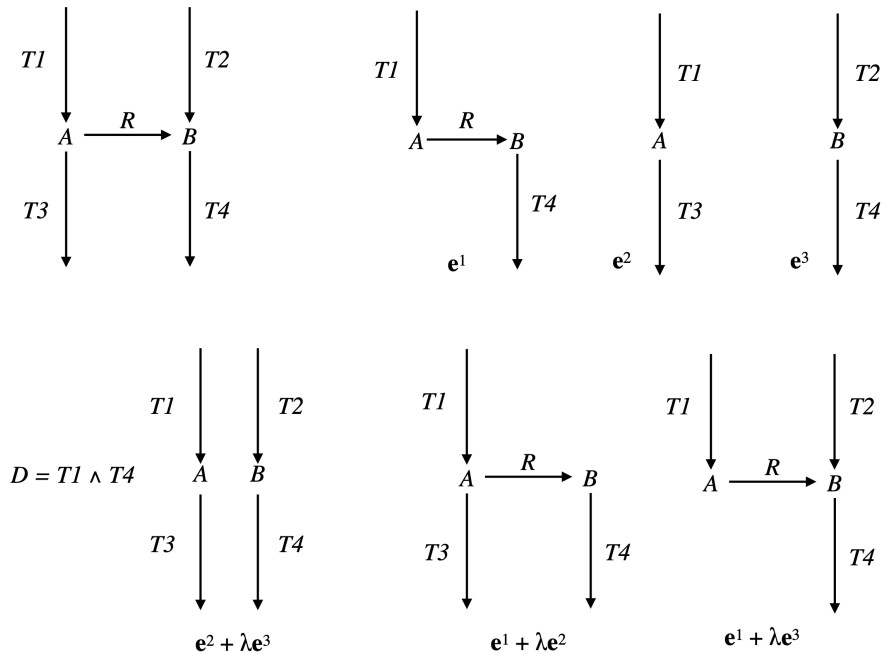

**Figure 9.** Simple network of Examples 3 and 5. The EFMs are $\mathbf{e}^1, \mathbf{e}^2, \mathbf{e}^3$. When we add the regulatory constraint $T1 \wedge T4$, only $\mathbf{e}^1$ remains an EFM, with support $\{T1, R, T4\}$, but new EFMs appear, made up of any positive combination of $\mathbf{e}^2$ and $\mathbf{e}^3$, all with the same support $\{T1, T2, T3, T4\}$. Moreover, new swNSDFVs also appear, which are not EFMs: any positive combination of $\mathbf{e}^1$ and $\mathbf{e}^2$, all with the same support $\{T1, R, T3, T4\}$, and any positive combination of $\mathbf{e}^1$ and $\mathbf{e}^3$, all with the same support $\{T1, R, T2, T4\}$.

Therefore, we extend the results of Proposition 9 by refining the structure in $F$ of swNSDFVs $_{\mathbf{RC}_D}$. Let $\{\mathbf{e}^m\}_{m \in M}$ be representatives of the extreme vectors of $F$, thus $F = cone_\oplus(\{\mathbf{e}^m\}_{m \in M})$. We get the following structure for $F^\circ$ regarding EFMs and swNSDFVs, given in the form of an algorithm.

- Let $R = \{m \in M \mid \mathbf{e}^m \in F^\circ\} = \{m \in M \mid \forall i \in I \, \mathbf{e}^m_i \neq 0\}$. Then EFMs $^{\mathbf{RC}_D} = \{\mathbf{e}^m\}_{m \in R}$. Note that $R$ can vary from $\varnothing$ to $M$, thus EFMs $^{\mathbf{RC}_D}$ from $\varnothing$ to EFMs $(F)$. If $R = M$, then EFMs $^{\mathbf{RC}_D} =$ EFMs $_{\mathbf{RC}_D} =$ swNSDFVs $_{\mathbf{RC}_D} =$ EFMs $(F)$ and the analysis

is done (if $\boldsymbol{\eta}$ has been chosen maximal in $sign(CFC_{\mathbf{RC}_D})$, this case corresponds to $I = \varnothing$). We consider the case $R \subset M$ here below.

- Let us consider successively all faces $F'$ of $F$ of dimension at least two, such that $\mathring{F}' \subseteq F^\circ$, i.e., $F'$ is not included in any hyperplane $\{\mathbf{v}_i = 0\}$ with $i \in I$ (the lattice of faces of $F$ can be explored for example in a way such that a sub-face is visited before a super-face; once such an $F'$ has been found, all faces of $F$ containing it are also suitable). Let $\{\mathbf{e}^k\}_{k \in K}$, with $K \subseteq M$, be representatives of the extreme vectors of $F'$. Thus, $\mathring{F}' = cone_{\oplus}^{+}(\{\mathbf{e}^k\}_{k \in K})$ and $I \subseteq \bigcup_{k \in K} supp(\mathbf{e}^k)$. For each of these $F'$, three exclusive cases can now be distinguished.

- If no facet of $F'$ has its interior included in $F^\circ$, i.e., any facet of $F'$ is included in a certain hyperplane $\{\mathbf{v}_i = 0\}$ with $i \in I$ (a necessary but insufficient condition is $K \subseteq M \backslash R$), then $\mathring{F}' \subseteq \text{EFMs}_{\mathbf{RC}_D} \backslash \text{EFMs}^{\mathbf{RC}_D}$ and $\bigcup_{k \in K} supp(\mathbf{e}^k)$ is a minimal support for vectors in $F^\circ$.

- If exactly one facet of $F'$ has its interior included in $F^\circ$, i.e., it is the only facet not included in any hyperplane $\{\mathbf{v}_i = 0\}$ with $i \in I$, then $\mathring{F}' \subseteq \text{swNSDFVs}_{\mathbf{RC}_D} \backslash \text{EFMs}_{\mathbf{RC}_D}$ and $\bigcup_{k \in K} supp(\mathbf{e}^k)$ is a non-strictly-decomposable non-minimal support for vectors in $F^\circ$ (non-strictly-decomposable support means that any vector which is a conical sum of the $\mathbf{e}^k$'s having this support is support-wise non-strictly-decomposable, independently of the choice of the non-negative coefficients fixing the contribution of each $\mathbf{e}^k$ in the distribution of the fluxes). This result follows immediately from the facts that one facet is not enough to decompose a certain vector in $\mathring{F}'$ strictly in terms of supports and that the support of the vectors in the interior of the facet in question is strictly included in the support of the vectors in $\mathring{F}'$.

- If at least two facets of $F'$ have their interior included in $F^\circ$, i.e., these facets are not included in any hyperplane $\{\mathbf{v}_i = 0\}$ with $i \in I$, then let $\{\mathbf{e}^l\}_{l \in L}$, with $L \subseteq K$, be representatives of the extreme vectors of all these facets (note that we have then necessarily $K \cap R \subseteq L$, i.e., $K \backslash L \subseteq K \backslash R$). Thus, the strict conical sum of the interiors of these facets, which is equal to $cone_{\oplus}^{+}(\{\mathbf{e}^l\}_{l \in L})$, is not empty in $\mathring{F}'$ (as there are at least two such facets) and is made up of the support-wise strictly-decomposable vectors of $\mathring{F}'$ (by construction): $cone_{\oplus}^{+}(\{\mathbf{e}^l\}_{l \in L}) = \mathring{F}' \backslash \text{swNSDFVs}_{\mathbf{RC}_D}$. Two subcases must therefore be distinguished.

- If $L = K$, i.e., the strict conical sum of the interiors of these facets is equal to $\mathring{F}'$, then $\mathring{F}' \subseteq F^\circ \backslash \text{swNSDFVs}_{\mathbf{RC}_D}$ and $\bigcup_{k \in K} supp(\mathbf{e}^k)$ is a strictly-decomposable support for vectors in $F^\circ$ (which means that any vector which is a conical sum of the $\mathbf{e}^k$'s having this support is support-wise strictly-decomposable, independently of the choice of the non-negative coefficients fixing the contribution of each $\mathbf{e}^k$ in the distribution of the fluxes).

- If $L \subset K$, then $\mathring{F}'$ is split into two nonempty subsets: $cone_{\oplus}^{+}(\{\mathbf{e}^l\}_{l \in L}) \subseteq F^\circ \backslash \text{swNSDFVs}_{\mathbf{RC}_D}$ and $\mathring{F}' \backslash cone_{\oplus}^{+}(\{\mathbf{e}^l\}_{l \in L}) \subseteq \text{swNSDFV}_{\mathbf{RC}_D} \backslash \text{EFMs}_{\mathbf{RC}_D}$ (note that $\mathring{F}' \backslash cone_{\oplus}^{+}(\{\mathbf{e}^l\}_{l \in L})$ is made up of the vectors of $cone_{\oplus}^{+}(\{\mathbf{e}^k\}_{k \in K})$ the conical decomposition of which on the $\mathbf{e}^k$'s requires at least one $\mathbf{e}^k$ with $k \in K \backslash L$). This means that part of the vectors of $\mathring{F}'$ are support-wise non-strictly-decomposable and part are support-wise strictly-decomposable, while having the same non-minimal support $\bigcup_{k \in K} supp(\mathbf{e}^k)$ (still equal to $\bigcup_{l \in L} supp(\mathbf{e}^l)$). This proves that support-wise strict decomposability generally depends not only on the support, but also on the positive values of the fluxes and that, unlike the particular cases above, we cannot speak of a strictly-decomposable or non-strictly-decomposable support. Note that, in this subcase, the support-wise non-strictly-decomposable vectors of $\mathring{F}'$ constitute the complementary, in the open cone $cone_{\oplus}^{+}(\{\mathbf{e}^k\}_{k \in K})$, of the open sub-cone $cone_{\oplus}^{+}(\{\mathbf{e}^l\}_{l \in L})$ which is thus a finite disjoint union of semi-open cones, each of which is conically generated (with positive or non-negative coefficients according to faces that are present or not) by extreme vectors $\mathbf{e}^k$'s with either $k \in K \backslash L \subseteq K \backslash R$ or $k \in L \backslash R$, thus in any case with $k \in K \backslash R$, i.e., by extreme vectors $\mathbf{e}^k \in \text{EFMs}(F) \backslash \text{EFMs}^{\mathbf{RC}_D}$.

We have finally proved the following result (keeping the notations of Proposition 9).

**Proposition 10.** *Let $swNSDFVs_{RC_D} \triangleq swNSDFVs(CFC_{RC_D \leq \eta})$ be the support-wise non-strictly-decomposable vectors of $CFC_{RC_D \leq \eta}$. We obtain $EFMs_{RC_D} \subseteq swNSDFVs_{RC_D}$ but, unlike the case of thermodynamic and kinetic (as in Proposition 5) constraints, there is in general no longer identity between $EFMs_{RC_D}$ and $swNSDFVs_{RC_D}$. Consider all faces $F'$ of $F$ of dimension at least two, such that $\mathring{F}' \subseteq F^\circ$ (i.e., $F'$ is not included in any hyperplane $\{v_i = 0\}$ with $i \in I$), and let $D(F') \triangleq cone^+_\oplus(\{\mathring{F}'' \mid F'' \text{ facet of } F' \text{ with } \mathring{F}'' \subseteq F^\circ\})$. Result of Proposition 9 can be stated as $EFMs_{RC_D} \backslash EFMs^{RC_D} = \bigcup_{\{F'|D(F')=\varnothing\}} \mathring{F}'$. Now, we have $swNSDFVs_{RC_D} \backslash EFMs_{RC_D} = \bigcup_{\{F'|\varnothing \neq D(F') \not\supseteq \mathring{F}'\}} \mathring{F}' \backslash D(F')$. Note that the $F''$'s considered here necessarily own at least one facet $F''$ that is not included in any hyperplane $\{v_i = 0\}$ with $i \in I$. There are actually two cases: For those $F'$'s which own exactly one such facet $F''$ (thus $D(F') = \mathring{F}'' \subseteq F^\circ \backslash \mathring{F}'$), we obtain $\mathring{F}' \subseteq swNSDFVs_{RC_D} \backslash EFMs_{RC_D}$ and the common support of vectors in $\mathring{F}'$ is thus a non-strictly-decomposable (independently of the choice of the distribution of the fluxes) non-minimal support for vectors in $F^\circ$. For those $F'$'s which own at least two such facets $F''$, we obtain $\mathring{F}' \backslash D(F') \subseteq swNSDFVs_{RC_D} \backslash EFMs_{RC_D}$ and $D(F') \subseteq F^\circ \backslash swNSDFVs_{RC_D}$, thus part of the vectors of $\mathring{F}'$ (consisting of a finite disjoint union of semi-open cones) are support-wise non-strictly-decomposable and part (consisting of an open cone) are support-wise strictly-decomposable (depending on the choice of the distribution of the fluxes), while having the same non-minimal support.*

**Example 6.** *Let us continue with the network of Examples 3 and 5 (see Figure 9). FC is a pointed cone of dimension 3 included in the positive orthant of $\mathbb{R}^5$ with axes $\{R, T3, T2, T1, T4\}$ in this order. We have $FC = \{(x \ y \ z \ x+y \ x+z)^T \mid x, y, z \geq 0\}$. FC has 3 extreme rays (EFMs $e^1, e^2, e^3$ seen above) and 3 facets (see Figure 10). When we add the Boolean constraint $D = T1 \wedge T4$, then $F = FC$ and the solution space is the semi-open cone $CFC_{RC_D} = F^\circ = \{(x \ y \ z \ x+y \ x+z)^T \mid x, y, z \geq 0, x+y > 0, x+z > 0\}$, i.e., $F$ without its two edges corresponding to $e^2$ and $e^3$. The only edge remaining in $F^\circ$ provides $EFMs^{RC_D} = \{e^1\}$ with support $\{T1, R, T4\}$. There are three faces of $F$ of dimension two with their interiors included in $F^\circ$: $F'_1 = cone_\oplus(\{e^2, e^3\})$, $F'_2 = cone_\oplus(\{e^1, e^3\})$ and $F'_3 = cone_\oplus(\{e^1, e^2\})$. $F'_1$ is the only one to have no facet with its interior included in $F^\circ$, i.e., such that $D(F'_1) = \varnothing$, thus $EFMs_{RC_D} \backslash EFMs^{RC_D} = \mathring{F}'_1 = cone^+_\oplus(\{e^2, e^3\})$. This means that the new EFMs that appear are all the positive combinations of $e^2$ and $e^3$, all with the same minimal support $\{T1, T3, T2, T4\}$. Regarding $F'_2$ and $F'_3$, they both have $\mathbb{R}_+ e^1$ as their only facet with interior included in $F^\circ$, thus $D(F'_2) = D(F'_3) = \mathbb{R}^*_+ e^1$ and $\mathring{F}'_2 = cone^+_\oplus(\{e^1, e^3\})$ and $\mathring{F}'_3 = cone^+_\oplus(\{e^1, e^2\})$ are both included in $swNSDFVs_{RC_D} \backslash EFMs_{RC_D}$. $\{T1, R, T2, T4\}$ and $\{T1, R, T3, T4\}$ are thus non-strictly-decomposable (independently of the respective values of the fluxes in T1 and T2 or in T3 and T4, respectively) non-minimal supports. The last face of $F$ with its interior included in $F^\circ$ is $F$ itself, of dimension three, with $\mathring{F} = cone^+_\oplus(\{e^1, e^2, e^3\})$. As all the facets $F'_1, F'_2, F'_3$ of $F$ have their interior included in $F^\circ$, we obtain $D(F) = \mathring{F}$. Thus, the pathways with nonzero fluxes in all five reactions, i.e., with support $\{T1, T3, R, T2, T4\}$, are exactly the support-wise strictly-decomposable ones.*

**Example 7.** *Let us consider the simple network comprising one reaction $R : A \rightarrow B$, where A and B are the two internal metabolites, and four transfer reactions $T1 : \rightarrow A$, $T2 : \rightarrow A$, $T3 : B \rightarrow$, $T4 : B \rightarrow$, and assume the five reactions irreversible (see Figure 11). FC is a pointed cone of dimension 3 included in the positive orthant of $\mathbb{R}^5$ with axes $\{T1, T2, T3, R, T4\}$ in this order. We have: $FC = \{(x \ y \ z \ x+y \ x+y-z)^T \mid x, y, z \geq 0, x+y \geq z\}$. FC has 4 facets and 4 extreme rays. Representatives of these extreme rays (EFMs) are $e^1 = (1 \ 0 \ 1 \ 1 \ 0)^T$, $e^2 = (1 \ 0 \ 0 \ 1 \ 1)^T$, $e^3 = (0 \ 1 \ 1 \ 1 \ 0)^T$ and $e^4 = (0 \ 1 \ 0 \ 1 \ 1)^T$, defined by their supports: $\{T1, R, T3\}$, $\{T1, R, T4\}$, $\{T2, R, T3\}$ and $\{T2, R, T4\}$, respectively.*

*Let us consider the Boolean constraint $D = T1 \wedge T3$. Then, $F = FC$ and the solution space is the semi-open cone $CFC_{RC_D} = F^\circ = \{(x \ y \ z \ x+y \ x+y-z)^T \mid x, z > 0, y \geq 0, x+y \geq z\}$, i.e., $F$ without its two facets $\{x = 0\}$ and $\{z = 0\}$.*

*The only EFM still present in $F^\circ$ is $e^1$, thus $EFMs^{RC_D} = \{e^1\}$ with support $\{T1, R, T3\}$.*

*There are two faces of F of dimension two with their interiors included in $F°$: $F'_1 = cone_\oplus(\{e^1, e^2\})$ and $F'_2 = cone_\oplus(\{e^1, e^3\})$, both with $\mathbb{R}_+ e^1$ as their only facet with interior included in $F°$. Thus, $D(F'_1) = D(F'_2) = \mathbb{R}^*_+ e^1$, $EFMs_{RC_D} \setminus EFMs^{RC_D} = \varnothing$, $\mathring{F}'_1 = cone^+_\oplus(\{e^1, e^2\}) \subseteq swNSDFVs_{RC_D} \setminus EFMs_{RC_D}$ and $\mathring{F}'_2 = cone^+_\oplus(\{e^1, e^3\}) \subseteq swNSDFVs_{RC_D} \setminus EFMs_{RC_D}$. $\{T1, R, T3, T4\}$ and $\{T1, T2, R, T3\}$ are thus non-strictly-decomposable (independently of the respective values of the fluxes in T3 and T4 or in T1 and T2, respectively) non-minimal supports.*

*The last face of F with its interior included in $F°$ is F itself, of dimension three, with $\mathring{F} = cone^+_\oplus(\{e^1, e^2, e^3, e^4\})$ defined in $F°$ by $\{y > 0, x + y > z\}$. F has exactly two facets with their interiors included in $F°$, namely, $F'_1$ and $F'_2$ and thus $D(F) = cone^+_\oplus(\{e^1, e^2, e^3\}) = F° \setminus swNSDFVs_{RC_D}$. The support-wise strictly-decomposable vectors constitute the open sub-cone of $\mathring{F}$ defined by $\{y < z\}$: $F° \setminus swNSDFVs_{RC_D} = \{(x\ y\ z\ x + y\ x + y - z)^T \mid x, y, z > 0, y < z < x + y\}$. They are the pathways of support $\{T1, T2, R, T3, T4\}$ such that the flux in T2 is smaller than the flux in T3; actually any vector $(x\ y\ z\ x + y\ x + y - z)^T$ with $x, y, z > 0, y < z < x + y$ can be decomposed as $(k(z - y)e^1 + (x + y - z)e^2) \oplus ((1 - k)(z - y)e^1 + ye^3)$, with arbitrary $k$, $0 < k < 1$, i.e., into two support-wise non-strictly-decomposable vectors respectively in $\mathring{F}'_1$ with support $\{T1, R, T3, T4\}$ and in $\mathring{F}'_2$ with support $\{T1, T2, R, T3\}$.*

*We thus have $\mathring{F} \setminus D(F) = cone^+_\oplus(\{e^2, e^3, e^4\}) \cup cone^+_\oplus(\{e^2, e^3\}) \subseteq swNSDFVs_{RC_D} \setminus EFMs_{RC_D}$. The support-wise non-strictly-decomposable vectors constitute the semi-open sub-cone of $\mathring{F}$ defined by $\{z \le y\}$: $\{(x\ y\ z\ x + y\ x + y - z)^T \mid x, y, z > 0, z \le y\}$. They are the pathways of support $\{T1, T2, R, T3, T4\}$ such that the flux in T2 is not smaller than the flux in T3. Any vector $(x\ y\ z\ x + y\ x + y - z)^T$ with $x, y, z > 0, z \le y$ is equal to $xe^2 + ze^3 + (y - z)e^4$ and belongs to $cone^+_\oplus(\{e^2, e^3, e^4\})$ if $z < y$ and to $cone^+_\oplus(\{e^2, e^3\})$ if $z = y$.*

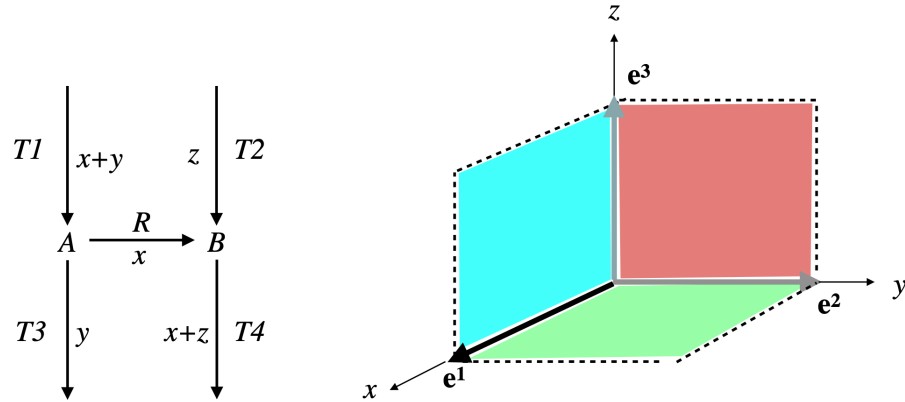

**Figure 10.** Simple network of Examples 3, 5 and 6. The flux cone *FC* of dimension 3 has 3 extreme rays, which are EFMs $e^1, e^2, e^3$, and 3 facets (in color in the picture that represents the projection of $\mathbb{R}^5$ on the subspace spanned by $x, y, z$, the fluxes in $R, T3, T2$, respectively). When we add the regulatory constraint $T1 \wedge T4$, the solution space becomes *FC* without its two edges corresponding to $e^2$ and $e^3$ (in gray), thus a disjoint union of the following open cones: extreme ray $e^1$, the interiors of the red, blue and green facets, and the interior of *FC*. Thus, only $e^1$ remains an EFM, with support $\{T1, R, T4\}$, but there are new EFMs, namely all positive combinations of $e^2$ and $e^3$ (the interior of the red facet), all with the same minimal support $\{T1, T3, T2, T4\}$. Moreover, there are new swNSDFVs with non-minimal support: all positive combinations of $e^1$ and $e^3$ (the interior of the blue facet), all with the same non-strictly-decomposable support $\{T1, R, T2, T4\}$, and all positive combinations of $e^1$ and $e^2$ (the interior of the green facet), all with the same non-strictly-decomposable support $\{T1, R, T3, T4\}$. All vectors in the interior of *FC*, which have $\{T1, R, T3, T2, T4\}$ as a support, are support-wise strictly-decomposable.

However, in the case of a general constraint $Bc = \bigvee_k D_k$, take care that, if the support-wise strictly-decomposable vectors of each $CFC_{\mathbf{RC}_{D_k} \le \eta}$ are support-wise strictly-decomposable in $CFC_{\mathbf{RC}_{Bc} \le \eta}$, it is not true for support-wise non-strictly-decomposable vectors and some of them for $CFC_{\mathbf{RC}_{D_k} \le \eta}$ may be decomposable in $CFC_{\mathbf{RC}_{Bc} \le \eta}$.

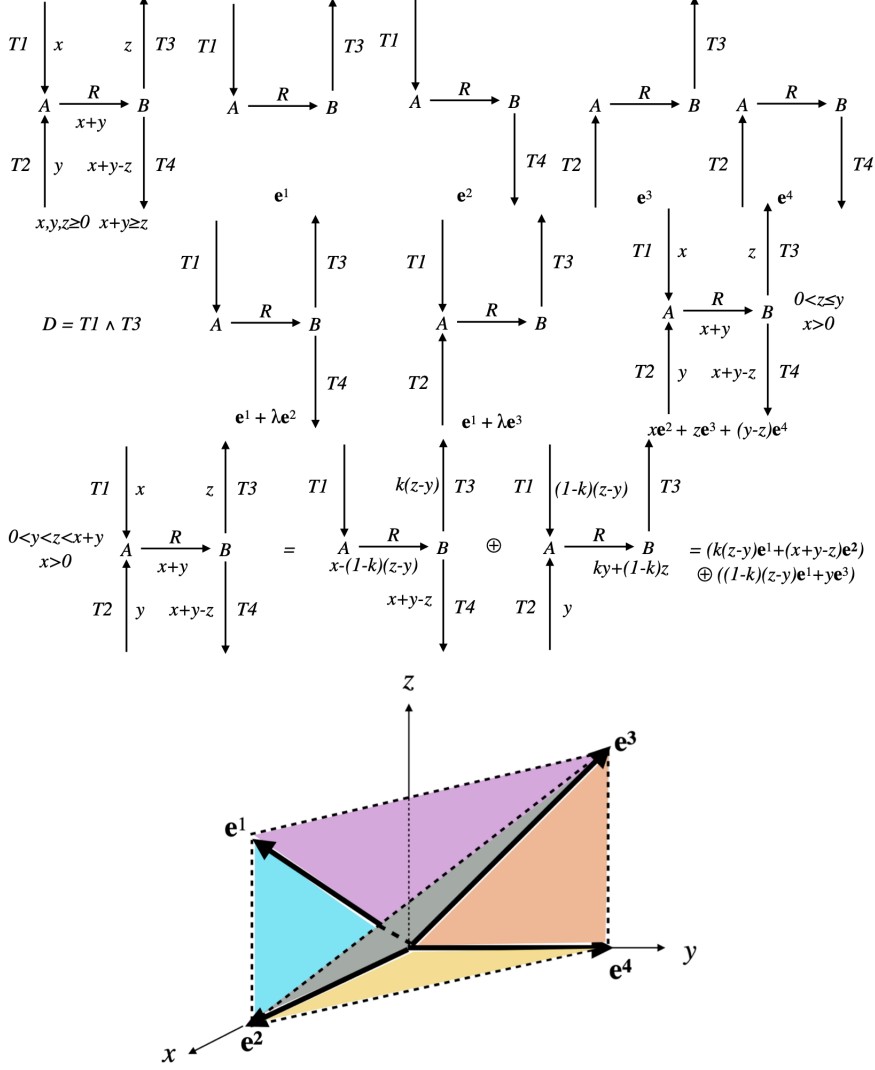

**Figure 11.** Simple network of Example [7]. Flux cone *FC* of dimension 3 has 4 extreme rays, which are EFMs $\mathbf{e}^1, \mathbf{e}^2, \mathbf{e}^3, \mathbf{e}^4$, and 4 facets (in color in the picture that represents the projection of $\mathbb{R}^5$ on the subspace spanned by $x, y, z$, the fluxes in $T1, T2, T3$, respectively). When we add the regulatory constraint $T1 \wedge T3$, the solution space becomes *FC* without its two facets in yellow and orange, thus a disjoint union of the following open cones: the extreme ray $\mathbf{e}^1$, the interiors of the blue and purple facets, and the interior of *FC*. Thus, only $\mathbf{e}^1$ remains an EFM, with support $\{T1, R, T3\}$ and there are no new EFMs, but the following new swNSDFVs occur. First, any positive combination of $\mathbf{e}^1$ and $\mathbf{e}^2$ (the interior of the blue facet), all with the same support $\{T1, R, T3, T4\}$, and any positive combination of $\mathbf{e}^1$ and $\mathbf{e}^3$ (the interior of the purple facet), all with the same support $\{T1, T2, R, T3\}$, these supports being thus non-strictly-decomposable (independently of the positive values of the fluxes) though not minimal. Second, the complementary in the interior of *FC* of the open sub-cone generated by the positive combinations of $\mathbf{e}^1, \mathbf{e}^2, \mathbf{e}^3$ (i.e., the interior of the cone with blue, purple and grey facets), that is the original network with nonzero fluxes in all five reactions, thus with support $\{T1, T2, R, T3, T4\}$, in the case where input flux $y$ in $T2$ is not smaller than output flux $z$ in $T3$, obtained as a positive combination of $\mathbf{e}^2, \mathbf{e}^3, \mathbf{e}^4$ (the interior of the cone with yellow, orange and grey facets) if $y > z$, or of $\mathbf{e}^2, \mathbf{e}^3$ (the interior of the gray cone) if $y = z$. On the other hand, when $y$ is smaller than $z$, the global pathway can be decomposed into two swNSDFVs of non-strictly-decomposable supports $\{T1, R, T3, T4\}$ and $\{T1, T2, R, T3\}$ in an infinite number of ways depending on a parameter $k, 0 < k < 1$ (decomposition of a vector in the interior of the cone with blue, purple and gray facets into two vectors in the interiors of the blue and purple facets respectively). This shows that in general support-wise strict-decomposability does not depend only on the support of the pathway but on the distribution of the fluxes.

This characterization of support-wise non-strictly-decomposable vectors in the presence of regulatory constraints does not extend directly from flux cones to flux polyhedra. The reason is that, due to inhomogeneous linear constraints **ILC**, certain faces of $FP_{\leq \eta}$ are not defined by equalities of the form $\mathbf{v}_k = 0$ and thus play no role in the definition of the support of the vectors they contain. Consequently, the basis of the reasoning above, namely, that the support of vectors of the interior of any face is larger than the supports of vectors of the interior of any facet of this face, does not hold. Nevertheless, the ideas developed for flux cones for dealing with faces included in a certain hyperplane $\{\mathbf{v}_i = 0\}$ with $i \in I$, and thus removed from the solution space due to the Boolean constraint considered, can be applied. However, it is necessary at each step, when considering an arbitrary face, to distinguish its facets resulting from **ILC**, not involved in the definition of the support, its facets included in a certain hyperplane $\{\mathbf{v}_k = 0\}$ with $k \notin I$ that are still present and contribute to the definition of the support, and its facets included in a certain hyperplane $\{\mathbf{v}_i = 0\}$ with $i \in I$ that are removed.

### 2.2.4. Case of Several Types of Constraints

In general, when analyzing a metabolic pathway, all known biological constraints will have to be taken into account together, typically kinetic constraints and regulatory constraints. For two such constraints $\mathbf{C}_1$ (say, a kinetic constraint **KC**, equivalent to **TC**, or $\mathbf{KC}^b$ in the absence of bounds on enzyme concentrations, equivalent to $\mathbf{TC}^b$) and $\mathbf{C}_2$ (say a regulatory constraint $\mathbf{RC}_{Bc}$), the solution space, in the case of a flux cone $FC$ (the reasoning would be similar for a flux polyhedron $FP$) is given by $CFC_{\mathbf{C}_1 \wedge \mathbf{C}_2} = FC \cap Sol_{\mathbf{C}_1 \wedge \mathbf{C}_2} = FC \cap Sol_{\mathbf{C}_1} \cap Sol_{\mathbf{C}_2} = CFC_{\mathbf{C}_1} \cap Sol_{\mathbf{C}_2}$. Now, from Propositions 2 and 5, $CFC_{\mathbf{C}_1}$ is a finite union of flux cones $FC_i$, which are certain particular faces of each flux tope of $FC$ and the constraint $\mathbf{C}_2$ can be applied to each one as to an original flux cone: $CFC_{\mathbf{C}_1 \wedge \mathbf{C}_2} = \bigcup_i FC_i \cap Sol_{\mathbf{C}_2} = \bigcup_i CFC_{i\,\mathbf{C}_2}$. From Proposition 7, each $CFC_{i\,\mathbf{C}_2}$ is a disjoint union of particular open faces of $FC_i$. In all, the solution space is a disjoint union of particular open faces of the flux topes of $FC$. From propositions above and Proposition 8, we can also conclude that the EFVs of $CFC_{\mathbf{C}_1 \wedge \mathbf{C}_2}$ are exactly the EFVs of $FC$ that satisfy both constraints $\mathbf{C}_1$ and $\mathbf{C}_2$.

**Proposition 11.** *The space of flux vectors in FC (resp., FP) satisfying both the kinetic constraint KC (or $KC^b$ in the absence of bounds on enzyme concentrations) and the regulatory constraint $RC_{Bc}$ is a finite disjoint union of open polyhedral cones (resp., open polyhedra) which are certain open faces of the flux topes of FC (resp., FP). The elementary flux vectors (resp., elementary flux points and vectors) are those of FC (resp., FP) that satisfy both constraints.*

Note that Proposition 10 applies also, by starting from each $FC_i$ instead of each flux tope of $FC$, to determine elementary flux modes and support-wise non-strictly-decomposable vectors.

### 3. General Case of Sign-Compatible Constraints

We are interested in determining, for general biological constraints **C**, what can be said about the structure of the solution spaces $CFC_{\mathbf{C}}$ or $CFP_{\mathbf{C}}$. More precisely, in identifying certain general features regarding constraints **C** (some of which are present in particular in biological constraints Equations (42), (47) and (50) we have considered so far), allowing us to clarify the mathematical and geometrical structure of the solution space, to determine the EFs (conformal non-decomposable fluxes) or EFMs (support-minimal fluxes) for this space and whether they characterize it and lastly how to compute them efficiently, in particular by checking the integration of this computation into the DD method. We will focus on deducing pertinent geometrical characteristics of the solution space only from general properties of the compatibility of the constraints with vector signs (i.e., with vector supports in each closed $r$-orthant or flux tope) and we will see that part of the results

obtained above for thermodynamic, kinetic and regulatory constraints actually depends only on these general global properties.

### 3.1. Sign-Invariant Constraints

A constraint $\mathbf{C_x}(\mathbf{v})$ (resp., $\exists \mathbf{x}\mathbf{C_x}(\mathbf{v})$), is said to be sign-invariant if it depends only on $sign(\mathbf{v})$, i.e., on the signs of the $\mathbf{v}_i$'s but not on their values (in the formulas below, free variables are assumed universally quantified):

$$\mathbf{C_x}(\mathbf{v}), sign(\mathbf{v}') = sign(\mathbf{v}) \Rightarrow \mathbf{C_x}(\mathbf{v}') \tag{86}$$

$$\exists \mathbf{x}\mathbf{C_x}(\mathbf{v}), sign(\mathbf{v}') = sign(\mathbf{v}) \Rightarrow \exists \mathbf{x}\mathbf{C_x}(\mathbf{v}'). \tag{87}$$

It follows from these definitions that, if the $\mathbf{C_x}(\mathbf{v})$'s are sign-invariant for all $\mathbf{x}$, then $\exists \mathbf{x}\mathbf{C_x}(\mathbf{v})$ is sign-invariant. Note that the property for a constraint to be support-invariant (i.e., to depend only on $supp(\mathbf{v})$) is stronger than the property to be sign-invariant. Actually, in any flux tope (or in any given closed orthant), having the same sign for two vectors is equivalent to having the same support and thus being sign-invariant for a constraint means being support-invariant in each flux tope independently.

**Example 8.** *It follows then from definitions Equations (41), (42) or (43) that the thermodynamic constraint $TC_M(v)$ or $\underline{TC}_M(v)$ is sign-invariant. Thus this is also the case for $TC(v)$ and $\underline{TC}(v)$ Equation (60) and for $TC^b(v)$ and $\underline{TC}^b(v)$ Equation (61).*

*On the other hand, the kinetic constraint $KC_{E,M}(v)$ Equation (47) is not sign-invariant, and this is also the case of $KC_E(v)$ Equation (78), because $\kappa_i(M)$ is bounded below and above (e.g., for Michaelis–Menten kinetics, $-k_i^- < \kappa_i < k_i^+$) and thus, for a given nonzero enzyme concentration $E_i$, there is no metabolite concentrations vector $M$ allowing an arbitrary flux value $v_i$ in reaction i (as soon as the value of $v_i/E_i$ is outside the $\kappa_i$ bounds).*

*The kinetic constraint $KC_M(v)$ Equation (80) is however sign-invariant. This follows from the linear dependency of $v_i$ on $E_i$. Actually, if $KC_M(v)$ is satisfied (for a certain $E$) and $sign(v') = sign(v)$, then, for all $i \in supp(v')$, $v_i$ and $\kappa_i(M)$ have the same sign; thus, it is also the case for $v_i'$ and $\kappa_i(M)$. Therefore, by taking $E_i' = v_i'/\kappa_i(M) = E_i v_i'/v_i$ when $v_i' \neq 0$, and $E_i' = 0$ when $v_i' = 0$, $KC_M(v')$ is satisfied (for $E'$). This sign-invariant property thus holds also for $KC(v)$ Equation (76) and for $KC^b(v)$ Equation (77) if the only bounds are on metabolite concentrations. However, it does not hold for $KC_M^b(v)$ (81) and for $KC^b(v)$ (77) in the presence of bounds on enzyme concentrations.*

*Finally, by definition, the regulatory constraint $RC_{Bc}(v)$ Equation (50) depends only on $supp(v)$, so is support-invariant and thus sign-invariant.*

**Lemma 4.** *All thermodynamic constraints are sign-invariant. Only kinetic constraints $KC_M$ and $KC$ are sign-invariant, as well as $KC^b$ with bounds only on metabolite concentrations (but not on enzyme concentrations, thus in particular without enzymatic resource constraint). The regulatory constraints are support-invariant, thus sign-invariant.*

The structure of the solution space of a sign-invariant constraint follows directly from its definition. Actually, if $\mathbf{v}$ satisfies a given sign-invariant constraint $\mathbf{C} = \mathbf{C_x}(\mathbf{v})$ (resp., $\mathbf{C} = \exists \mathbf{x}\mathbf{C_x}(\mathbf{v})$), then $\mathring{O}_{sign(\mathbf{v})} = \{\mathbf{x} \in \mathbb{R}^r \mid sign(\mathbf{x}) = sign(\mathbf{v})\}$ is included in $Sol_{\mathbf{C}}$ and thus $Sol_{\mathbf{C}} = \bigcup_{\{\mathbf{v}|\mathbf{C}(\mathbf{v})\}} \mathring{O}_{sign(\mathbf{v})}$ is a disjoint union of open orthants. This result can be compared to Lemma 2. More precisely, if we consider a support-invariant constraint $\mathbf{C}$, the same reasoning applies and gives $Sol_{\mathbf{C}} = \bigcup_{\{\mathbf{v}|\mathbf{C}(\mathbf{v})\}} S_{supp(\mathbf{v})}$, where $S_{\boldsymbol{\rho}} = \{\mathbf{x} \in \mathbb{R}^r \mid supp(\mathbf{x}) = \boldsymbol{\rho}\}$ denotes the set of vectors of support $\boldsymbol{\rho}$, where we code a support as a binary vector $\boldsymbol{\rho}$ (1 coding support membership and 0 non-membership). A support-invariant constraint $\mathbf{C}$ is thus equivalent (in extension) to a family $\{S_{\boldsymbol{\rho}^i}\}_i$ of such support sets and, as there are $2^r$ possible binary vectors, there are thus $2^{2^r}$ different support-invariant constraints. Now, a given $S_{\boldsymbol{\rho}}$ is equivalent to the Boolean constraint $D_{\boldsymbol{\rho}} = \bigwedge_{\{i|\boldsymbol{\rho}_i=1\}} \mathbf{v}_i \wedge \bigwedge_{\{j|\boldsymbol{\rho}_j=0\}} \neg\mathbf{v}_j$ and thus an arbitrary support-invariant constraint $\mathbf{C}$ is equivalent to the Boolean constraint

$Bc = \bigvee_i D_{\rho^i}$, thus to the regulatory constraint $\mathbf{RC}_{Bc}$. Thus support-invariant constraints identify with regulatory constraints and all properties we have demonstrated for the latter apply to the first. For example, by associating with any sign vector $\boldsymbol{\eta}$ its support $\boldsymbol{\rho}$ (i.e., the support of any vector having the given sign), given by $\rho_i = 1$ if $\eta_i = +$ or $-$ and $\rho_i = 0$ if $\eta_i = 0$, we deduce that, for any binary vector $\boldsymbol{\rho}$, there are $2^{|supp(\boldsymbol{\rho})|}$ sign vectors $\boldsymbol{\eta}$ with support $\boldsymbol{\rho}$, given by $\eta_i = +$ or $-$ if $\rho_i = 1$ and $0$ else, and we obtain $S_{\boldsymbol{\rho}} = \bigcup_{\{\boldsymbol{\eta}^i | supp(\boldsymbol{\eta}^i) = \boldsymbol{\rho}\}} \mathring{O}_{\boldsymbol{\eta}^i}$ and $Sol_{\mathbf{C}} = \bigcup_{\{\boldsymbol{\eta}^i | supp(\boldsymbol{\eta}^i) = supp(\mathbf{v}) \wedge \mathbf{C}(\mathbf{v})\}} \mathring{O}_{\boldsymbol{\eta}^i}$, i.e., that $Sol_{\mathbf{C}}$ is a disjoint union of open orthants, which is precisely the statement of Lemma 2. Note that a support-invariant constraint $\mathbf{C}$ is entirely defined by its restriction to an arbitrary closed $r$-orthant $O_{\boldsymbol{\eta}}$, i.e., for $\boldsymbol{\eta}$ an arbitrary full support sign vector, as $Sol_{\mathbf{C}}$ can be reconstituted from $Sol_{\mathbf{C}} \cap O_{\boldsymbol{\eta}}$. Moreover, as to be sign-invariant and to be support-invariant coincide in any closed $r$-orthant, a sign-invariant constraint is nothing but a constraint whose restriction to any closed $r$-orthant is support-invariant, i.e., which coincides in any closed $r$-orthant with a well-defined regulatory constraint (but such regulatory constraints differ in general from one $r$-orthant to another, while obviously coinciding on their intersection). A sign-invariant constraint $\mathbf{C}$ is equivalent to a family $\{\mathring{O}_{\boldsymbol{\eta}^i}\}_i$ of open orthants and, as there are $3^r$ possible sign vectors, there are thus $2^{3^r}$ different sign-invariant constraints. Applying the merging method described in the proof of Lemma 2, these open orthants can be grouped together in order to obtain a family of semi-open orthants, without any possible further merging between any two of them.

**Proposition 12.** *Support-invariant constraints coincide with Boolean constraints, i.e., regulatory constraints. They are completely characterized by their restriction to any closed $r$-orthant and number $2^{2^r}$. Sign-invariant constraints coincide with constraints whose restriction to any closed $r$-orthant is given by a regulatory constraint (with identity of such regulatory constraints on the intersections of any two of these orthants). They number $2^{3^r}$. The set $Sol_{\mathbf{C}}$ of vectors in $\mathbb{R}^r$ satisfying the sign-invariant constraint $\mathbf{C} = \mathbf{C}_x(v)$ (resp., $\mathbf{C} = \exists x \mathbf{C}_x(v)$) is a disjoint union of open orthants, which can be grouped together according to Lemma 2 to provide a disjoint union of semi-open orthants without any possible further merging between any two of them.*

An important consequence is that, if we reason for each closed $r$-orthant separately, i.e., flux tope by flux tope in the solution space, then all results demonstrated in Section 2.2.3 for regulatory constraints apply to sign-invariant constraints. With the previous definitions of open or semi-open polyhedral cones and open or semi-open polyhedra and, more generally, of open or semi-open faces of polyhedral cones or polyhedra, we get the following result.

**Theorem 1.** *Let $\mathbf{C} = \mathbf{C}_x(v)$ (resp., $\mathbf{C} = \exists x \mathbf{C}_x(v)$) be a sign-invariant constraint and $CFC_{\mathbf{C}}$ (resp., $CFP_{\mathbf{C}}$) be the associated constrained flux cone subset (resp., the associated constrained flux polyhedron subset). Then, $CFC_{\mathbf{C}}$ (resp., $CFP_{\mathbf{C}}$) is a finite disjoint union of open polyhedral cones (resp., open polyhedra), which are certain open faces of the flux topes $FC_{\leq \boldsymbol{\eta}}$ (resp., $FP_{\leq \boldsymbol{\eta}}$) for all $\boldsymbol{\eta}$ maximal sign vectors in $sign(FC)$ (resp., $sign(FP)$). They can be grouped together according to Lemma 2 to provide a disjoint union of semi-open faces of the $FC_{\leq \boldsymbol{\eta}}$'s (resp., $FP_{\leq \boldsymbol{\eta}}$'s) without any possible further merging between any two of them. Elementary fluxes are obtained by collecting those for each flux tope (which is not the case for elementary flux modes where, after collecting them, only those with minimal support have to be kept) and are nothing but the elementary fluxes of $FC$ (resp., $FP$) that satisfy the constraint $\mathbf{C}$ (which again is not the case for elementary flux modes):*

$$EFVs(CFC_{\mathbf{C}}) = EFVs(FC) \cap Sol_{\mathbf{C}} = EFMs(FC) \cap Sol_{\mathbf{C}} \subseteq EFMs(CFC_{\mathbf{C}}) \qquad (88)$$

$$EFPs(CFP_{\mathbf{C}}) = EFPs(FP) \cap Sol_{\mathbf{C}} \qquad EFVs(CFP_{\mathbf{C}}) = EFVs(C_{FP}) \cap Sol_{\mathbf{C}}. \qquad (89)$$

**Proof of Theorem 1.** The disjoint decomposition of the solution space into open faces of the flux topes $FC_{\leq \boldsymbol{\eta}}$ (resp., $FP_{\leq \boldsymbol{\eta}}$) results from Proposition 12 or directly from the fact that all vectors of the strict conical hull $cone^+(\mathbf{v}^1, \ldots, \mathbf{v}^n) = \{\beta_1 \mathbf{v}^1 + \ldots + \beta_n \mathbf{v}^n \mid \beta_1, \ldots, \beta_n > 0\}$ and of the strict convex hull $conv^+(\mathbf{v}^1, \ldots, \mathbf{v}^n) = \{\alpha_1 \mathbf{v}^1 + \ldots + \alpha_n \mathbf{v}^n \mid \alpha_1, \ldots, \alpha_n > 0,$

$\alpha_1 + \ldots + \alpha_n = 1$} of given vectors $\mathbf{v}^1, \ldots, \mathbf{v}^n$ in a flux tope, thus in a closed orthant (so the sum is conformal), have the same support $supp(\mathbf{v}^1) \cup \ldots \cup supp(\mathbf{v}^n)$, thus the same sign, and from the fact that an open face of a polyhedron is the Minkowski sum of the strict convex hull of its vertices and the strict conical hull of its extreme vectors. Thus, an open face of a flux tope $FC_{\leq \eta}$ (resp., $FP_{\leq \eta}$ ) either does not intersect the solution space or is completely included in it. We have seen (55) that any elementary flux or any elementary flux mode of $FC$ (resp., $FP$) that satisfies the constraint is an elementary flux or an elementary flux mode of the solution space (the conditions of validity of (55) for elementary fluxes are satisfied from the structure of the solution space). These elementary fluxes or elementary flux modes actually identify with the extreme vectors of the $FC_{\leq \eta}$'s (resp., extreme points of the $FP_{\leq \eta}$'s and extreme vectors of the $C_{FP \leq \eta}$'s) that satisfy the constraint, i.e., whose sign satisfies the constraint. Reciprocally, any vector of the solution space that is not in this case necessarily belongs to an open face of a certain $FC_{\leq \eta}$ (or $C_{FP \leq \eta}$) of dimension at least two (resp., of a certain $FP_{\leq \eta}$ of dimension at least one) and is thus conformally (resp., convex-conformally) decomposable in this open face and consequently cannot be an elementary vector (resp., elementary point) of the solution space (obviously, the decomposition involves two vectors of same support, thus nothing can be deduced regarding elementary flux modes). This gives the result for elementary fluxes. □

Theorem 1 applies in particular to all thermodynamic constraints, regulatory constraints and those kinetic constraints described in Lemma 4. Especially, we directly obtain the results of Propositions 7 and 8 for regulatory constraints.

Note that our only knowledge of the EFVs for $CFC_{\mathbf{C}}$ (resp., the EFPs and EFVs for $CFP_{\mathbf{C}}$) is really a long way from characterizing the solution space $CFC_{\mathbf{C}}$ (resp., $CFP_{\mathbf{C}}$): actually they are just the ExVs of each tope $CFC_{\mathbf{C} \leq \eta}$ (resp., the ExPs and ExVs of each $CFP_{\mathbf{C} \leq \eta}$), i.e., the only one-dimension open polyhedral cones, i.e., edges (resp., zero-dimension polyhedra, i.e., vertices, and edges for the recession cones) among all the open cones (resp., open polyhedra) of any dimension that constitute $CFC_{\mathbf{C}}$ (resp., $CFP_{\mathbf{C}}$).

As in each closed $r$-orthant, i.e., in each flux tope, a sign-invariant constraint identifies with a regulatory constraint, Propositions 9 and 10 demonstrated for regulatory constraints apply thus to sign-invariant constraints. Nevertheless, they are stated for constraints that are conjunctions of literals and, even if a decomposition into such disjuncts is always possible from the identity above, it is not necessarily natural for an arbitrary sign-invariant constraint and a global characterization for each flux tope is preferable, in particular for what concerns support-wise non-strictly-decomposable vectors. The difference is that one has to deal for $CFC_{\mathbf{C} \leq \eta}$ with an arbitrary family of open faces of $FC_{\leq \eta}$ instead of a single semi-open face of $FC_{\leq \eta}$, obtained specifically as a face of $FC_{\leq \eta}$ without certain of its facets, for $CFC_{\mathbf{RC}_D \leq \eta}$.

Let us adopt notations similar to those used in Propositions 9 and 10, i.e., let EFMs $^{\mathbf{C}} \overset{\triangle}{=}$ EFMs $(FC_{\leq \eta}) \cap Sol_{\mathbf{C}}$ be the elementary flux modes of $FC_{\leq \eta}$ that satisfy $\mathbf{C}$ (i.e., the elementary flux vectors of $CFC_{\mathbf{C} \leq \eta}$), EFMs $_{\mathbf{C}} \overset{\triangle}{=}$ EFMs $(CFC_{\mathbf{C} \leq \eta})$ be the elementary flux modes of $CFC_{\mathbf{C} \leq \eta}$ and swNSDFVs $_{\mathbf{C}} \overset{\triangle}{=}$ swNSDFVs $(CFC_{\mathbf{C} \leq \eta})$ be the support-wise non-strictly-decomposable vectors of $CFC_{\mathbf{C} \leq \eta}$. We have thus EFMs $^{\mathbf{C}} \subseteq$ EFMs $_{\mathbf{C}} \subseteq$ swNSDFVs $_{\mathbf{C}}$. The proof of Proposition 9 adapts straightforwardly: EFMs $^{\mathbf{C}}$ correspond to the open faces of dimension one (extreme rays) of $FC_{\leq \eta}$ in $CFC_{\mathbf{C} \leq \eta}$ and EFMs $_{\mathbf{C}} \setminus$ EFMs $^{\mathbf{C}}$ are made up of all the open faces $\mathring{F}$ in $CFC_{\mathbf{C} \leq \eta}$ where $F$ is a face of dimension at least two of $FC_{\leq \eta}$ but not any proper (i.e., with positive dimension, less than the dimension of $F$) face $F'$ of $F$ is such that $\mathring{F}'$ is in $CFC_{\mathbf{C} \leq \eta}$. The result is based on the properties that all vectors of an open face $\mathring{F}$ in $CFC_{\mathbf{C} \leq \eta}$ have the same support, and the common support of vectors of $\mathring{F}'$, where $F'$ is a face of $F$, is strictly included in that of vectors of $\mathring{F}$. In the same way, the proof of Proposition 10 is easily adapted for what concerns swNSDFVs $_{\mathbf{C}} \setminus$ EFMs $_{\mathbf{C}}$. For this we consider successively all the open faces $\mathring{F}$ in $CFC_{\mathbf{C} \leq \eta}$ where $F$ is a face of dimension at least two of $FC_{\leq \eta}$ which owns at least one proper face $F'$ such that $\mathring{F}'$ is in $CFC_{\mathbf{C} \leq \eta}$. For each such given $\mathring{F}$, let $\{F'_j\}_{j \in J}$ be the family of all such $F'$. Let $\{\mathbf{e}^k\}_{k \in K}$ be representatives

of the extreme vectors of $F$ and $\{\mathbf{e}^l\}_{l\in L}$, with $L \subseteq K$, be representatives of the extreme vectors of all these $F_j'$. Thus, $cone_\oplus^+(\{\mathbf{e}^k\}_{k\in K}) = \mathring{F}$ and $cone_\oplus^+(\{\mathbf{e}^l\}_{l\in L}) = cone_\oplus^+(\{\mathring{F}_j'\}_{j\in J})$ and we obtain $\mathring{F} \cap cone_\oplus^+(\{\mathbf{e}^l\}_{l\in L}) = \mathring{F}\setminus$ swNSDFVs $_{\mathbf{C}}$, as by construction those vectors of $\mathring{F}$ which belong to $cone_\oplus^+(\{\mathring{F}_j'\}_{j\in J})$ are precisely those in $\mathring{F}$ which are support-wise strictly-decomposable in $CFC_{\mathbf{C}\leq\boldsymbol{\eta}}$, the decomposition being achieved along vectors in the $\mathring{F}_j''$'s, whose support is strictly included in the common support of vectors in $\mathring{F}$. Therefore, three different exclusive cases can be distinguished for $\mathring{F}$:

- $cone_\oplus^+(\{\mathbf{e}^l\}_{l\in L}) \cap \mathring{F} = \varnothing$, i.e., the $F_j''$'s are all included in a same facet of $F$, that is in a same hyperplane $\{\mathbf{v}_i = 0\}$ for a certain coordinate $i$ of the affine span of $F$, which is still equivalent to $\bigcup_{l\in L} supp(\mathbf{e}^l) \subset \bigcup_{k\in K} supp(\mathbf{e}^k)$ (this is for example the case if $|J| = 1$). Then, $\mathring{F} \subseteq$ swNSDFVs $_{\mathbf{C}}\setminus$ EFMs $_{\mathbf{C}}$ is entirely made up of support-wise non-strictly-decomposable vectors and $\bigcup_{k\in K} supp(\mathbf{e}^k)$ is a non-strictly-decomposable non-minimal support for vectors in $CFC_{\mathbf{C}\leq\boldsymbol{\eta}}$.
- $cone_\oplus^+(\{\mathbf{e}^l\}_{l\in L}) = \mathring{F}$, i.e., $L = K$. Then, $\mathring{F} \subseteq CFC_{\mathbf{C}\leq\boldsymbol{\eta}}\setminus$ swNSDFVs $_{\mathbf{C}}$ is entirely made up of support-wise strictly-decomposable vectors and $\bigcup_{k\in K} supp(\mathbf{e}^k)$ is a strictly-decomposable support for vectors in $CFC_{\mathbf{C}\leq\boldsymbol{\eta}}$.
- $cone_\oplus^+(\{\mathbf{e}^l\}_{l\in L}) \subset \mathring{F}$, i.e., $L \subset K$ and not all $F_j''$'s are included in a same facet of $F$, which means that, for all $i$ coordinates of the affine span of $F$, there exists $j \in J$ such that $F_j'$ is not included in the hyperplane $\{\mathbf{v}_i = 0\}$, or equivalently, there exists $l \in L$ such that $\mathbf{e}_i^l \neq 0$, which is still equivalent to $\bigcup_{l\in L} supp(\mathbf{e}^l) = \bigcup_{k\in K} supp(\mathbf{e}^k)$. Then, $\mathring{F}$ is split into two nonempty subsets: $cone_\oplus^+(\{\mathbf{e}^l\}_{l\in L}) = \mathring{F}\setminus$ swNSDFVs $_{\mathbf{C}}$, made up of support-wise strictly-decomposable vectors, and $\mathring{F}\setminus cone_\oplus^+(\{\mathbf{e}^l\}_{l\in L}) \subseteq$ swNSDFVs $_{\mathbf{C}}\setminus$ EFMs $_{\mathbf{C}}$, made up of support-wise non-strictly-decomposable vectors (note that $\mathring{F}\setminus cone_\oplus^+(\{\mathbf{e}^l\}_{l\in L})$ is made up of those vectors of $\mathring{F}$ the conical decomposition of which on the $\mathbf{e}^k$'s requires at least one $\mathbf{e}^k$ with $k \in K\setminus L$. As all vectors of $\mathring{F}$ have the same non-minimal support $\bigcup_{k\in K} supp(\mathbf{e}^k)$, this proves that support-wise strict decomposability does not generally only depend on the support but also on the positive values of the fluxes and that, unlike the two particular cases above, we cannot speak of a strictly-decomposable or non-strictly-decomposable support. Note that the support-wise non-strictly-decomposable vectors of $\mathring{F}$ are obtained as the complementary, in this open cone, of the open sub-cone $cone_\oplus^+(\{\mathbf{e}^l\}_{l\in L})$, which is thus a disjoint union of semi-open cones (its connected components), each of which being conically generated both by certain extreme vectors $\mathbf{e}^k$ with $k \in K\setminus L$ (that necessarily do not belong to EFMs $^{\mathbf{C}}$), with non-negative coefficients, and by certain extreme vectors $\mathbf{e}^k$ with $k \in L$ (that may belong or not to EFMs $^{\mathbf{C}}$), with positive coefficients (in order to keep the concerned facets common with $cone_\oplus(\{\mathbf{e}^l\}_{l\in L})$).

We have finally proved the following result (while keeping the notations of Theorem [1]).

**Theorem 2.** *For $\mathbf{C}$ a sign-invariant constraint, $FC_{\leq\boldsymbol{\eta}}$ a flux tope and $CFC_{\mathbf{C}\leq\boldsymbol{\eta}}$ the associated subset of the solution space (disjoint union of open faces of $FC_{\leq\boldsymbol{\eta}}$ from Theorem [1]), let $EFMs^{\mathbf{C}} \overset{\triangle}{=} EFMs(FC_{\leq\boldsymbol{\eta}}) \cap Sol_{\mathbf{C}}$ be the elementary flux modes of $FC_{\leq\boldsymbol{\eta}}$ that satisfy $\mathbf{C}$ (equal to $EFVs(CFC_{\mathbf{C}\leq\boldsymbol{\eta}})$), $EFMs_{\mathbf{C}} \overset{\triangle}{=} EFMs(CFC_{\mathbf{C}\leq\boldsymbol{\eta}})$ be the elementary flux modes of $CFC_{\mathbf{C}\leq\boldsymbol{\eta}}$, and $swNSDFVs_{\mathbf{C}} \overset{\triangle}{=} swNSDFVs(CFC_{\mathbf{C}\leq\boldsymbol{\eta}})$ be the support-wise non-strictly-decomposable vectors of $CFC_{\mathbf{C}\leq\boldsymbol{\eta}}$, with $EFMs^{\mathbf{C}} \subseteq EFMs_{\mathbf{C}} \subseteq swNSDFVs_{\mathbf{C}}$. $EFMs^{\mathbf{C}}$ correspond to the open faces of dimension one (extreme rays) of $FC_{\leq\boldsymbol{\eta}}$ belonging to $CFC_{\mathbf{C}\leq\boldsymbol{\eta}}$. Consider now successively all the open faces $\mathring{F}$ in $CFC_{\mathbf{C}\leq\boldsymbol{\eta}}$ where $F$ is a face of dimension at least two of $FC_{\leq\boldsymbol{\eta}}$, and for each such given $F$, let $\{\mathbf{e}^k\}_{k\in K}$ be representatives of the extreme vectors of $F$ and $\{\mathbf{e}^l\}_{l\in L}$, with $L \subseteq K$, be representatives of the extreme vectors of all proper (i.e., with positive dimension, less than the dimension of $F$) faces $F'$ of $F$ such that $\mathring{F}'$ belongs to $CFC_{\mathbf{C}\leq\boldsymbol{\eta}}$. Thus, $cone_\oplus^+(\{\mathbf{e}^k\}_{k\in K}) = \mathring{F}$ and let $D(F) \overset{\triangle}{=} cone_\oplus^+(\{\mathbf{e}^l\}_{l\in L})$. Consequently, if $L = \varnothing$, then $\mathring{F} \subseteq EFMs_{\mathbf{C}}\setminus EFMs^{\mathbf{C}}$; if $L \neq \varnothing$ and $\bigcup_{l\in L} supp(\mathbf{e}^l) \subset \bigcup_{k\in K} supp(\mathbf{e}^k)$, then $\mathring{F} \subseteq swNSDFVs_{\mathbf{C}}\setminus EFMs_{\mathbf{C}}$ and $\bigcup_{k\in K} supp(\mathbf{e}^k)$*

*is a non-strictly-decomposable non-minimal support for vectors in $CFC_{C\leq\eta}$; if $L = K$, then $\mathring{F} \subseteq CFC_{C\leq\eta}\backslash swNSDFVs_C$ and $\bigcup_{k\in K} supp(e^k)$ is a strictly-decomposable support for vectors in $CFC_{C\leq\eta}$; if $L \subset K$ and $\bigcup_{l\in L} supp(e^l) = \bigcup_{k\in K} supp(e^k)$, then $D(F) = \mathring{F}\backslash swNSDFVs_C$, a non-empty open cone, constitutes the support-wise strictly-decomposable vectors in $\mathring{F}$ and $\mathring{F}\backslash D(F) \subseteq swNSDFVs_C\backslash EFMs_C$, a non-empty finite disjoint union of semi-open cones, constitutes the support-wise non-strictly-decomposable vectors in $\mathring{F}$, all having the same non-minimal support (decomposability depending in this case not only on the support but also on the distribution of the fluxes). We finally obtain $EFMs_C\backslash EFMs^C$ and $swNSDFVs_C\backslash EFMs_C$ by collecting these vectors for each $\mathring{F}$.*

Note that even the knowledge of swNSDFVs $_C$ is not enough to completely reconstruct the tope $CFC_{C\leq\eta}$ of the solution space.

In order to deal with the usual enzymatic resource constraint (more generally capacity constraints) in the kinetic constraints, we generalize the sign-invariance criterion. A constraint $\mathbf{C_x(v)}$ (resp., $\exists\mathbf{xC_x(v)}$), is said to be contracting-sign-invariant if, when satisfied by one vector, it is satisfied by any vector having the same sign which is not greater (on each component), i.e., that belongs to the open rectangle parallelepiped defined by the null vector and the given vector:

$$\mathbf{C_x(v)}, sign(\mathbf{v'}) = sign(\mathbf{v}), \forall i |\mathbf{v}'_i| \leq |\mathbf{v}_i| \Rightarrow \mathbf{C_x(v')} \tag{90}$$

$$\exists\mathbf{xC_x(v)}, sign(\mathbf{v'}) = sign(\mathbf{v}), \forall i |\mathbf{v}'_i| \leq |\mathbf{v}_i| \Rightarrow \exists\mathbf{xC_x(v')}. \tag{91}$$

Obviously a sign-invariant constraint is contracting-sign-invariant and, if the $\mathbf{C_x(v)}$'s are contracting-sign-invariant for all $\mathbf{x}$, then $\exists\mathbf{xC_x(v)}$ is contracting-sign-invariant.

**Example 9.** *The kinetic constraints $KC_M^b(v)$ Equation (81) and $KC^b(v)$ Equation (77) are contracting-sign-invariant in the absence of positive lower bounds on enzyme concentrations (i.e., when $\mathbf{E}^- = \mathbf{0}$). Actually, if $KC_M^b(v)$ is satisfied (for a certain $\mathbf{E}$ verifying $\mathbf{c}^T\mathbf{E} \leq W$ and $\mathbf{E} \leq \mathbf{E}^+$) and $sign(\mathbf{v'}) = sign(\mathbf{v})$ with $|\mathbf{v}'_i| \leq |\mathbf{v}_i|$ for all i, we saw that, by taking $\mathbf{E}'_i = \mathbf{v}'_i/\kappa_i(\mathbf{M}) = \mathbf{E}_i\mathbf{v}'_i/\mathbf{v}_i$ when $\mathbf{v}'_i \neq 0$, and $\mathbf{E}'_i = 0$ when $\mathbf{v}'_i = 0$, then $KC_M(v')$ is satisfied (for $\mathbf{E}'$). Now, as $|\mathbf{v}'_i| \leq |\mathbf{v}_i|$, we have $\mathbf{E}' \leq \mathbf{E}$ and thus $\mathbf{c}^T\mathbf{E}' \leq \mathbf{c}^T\mathbf{E} \leq W$ and $\mathbf{E}' \leq \mathbf{E}^+$, so $KC_M^b(v')$ is satisfied.*

**Lemma 5.** *The kinetic constraints $KC_M^b$ and $KC^b$ are contracting-sign-invariant in the absence of positive lower bounds on enzyme concentrations (i.e., when $\mathbf{E}^- = \mathbf{0}$).*

We will call **0**-star domain a subset $SD$ of $\mathbb{R}^r$ which has the property that the whole open segment joining **0** to any element of $SD$ is included in $SD$: $\mathbf{v} \in SD, 0 < \lambda < 1 \Rightarrow \lambda\mathbf{v} \in SD$. Any cone is a **0**-star domain.

**Theorem 3.** *If $C = C_x(v)$ (or $C = \exists xC_x(v)$) is contracting-sign-invariant, then $CFC_C$ is a **0**-star domain and there exists a neighborhood $N = ]-\delta, +\delta[^r$ of **0** for a certain $\delta > 0$ such that $CFC_C \cap N$ is a finite disjoint union of N-truncated open polyhedral cones, which are the intersection with N of open faces of the flux topes $FC_{\leq\eta}$. In addition, results (88) regarding EFVs and EFMs hold in N, i.e., for sufficiently small fluxes. The same holds for $CFP_C$ with the flux topes $FP_{\leq\eta}$ if **0** is an interior point of $FP$, and in this case results (89) regarding EFVs and EFPs hold in N.*

**Proof of Theorem 3.** The property of being a **0**-star domain is a direct consequence of the definition: if a contracting-sign-invariant constraint is satisfied by a vector $\mathbf{v}$, it is satisfied by $\lambda\mathbf{v}$ for all $0 < \lambda < 1$. From the proof of Theorem 1, it follows that if a nonzero vector $\mathbf{v} \in FC_{\leq\eta}$ verifies a given contracting-sign-invariant constraint, then any vector $\mathbf{v'}$ of the minimal open face of $FC_{\leq\eta}$ containing $\mathbf{v}$ (thus having the same sign as $\mathbf{v}$) and belonging to $]-\delta_{\mathbf{v}}, +\delta_{\mathbf{v}}[^r$ with $\delta_{\mathbf{v}} = min_{i\in supp(\mathbf{v})}|\mathbf{v}_i|$ (contracting condition) also verifies the constraint. The result is obtained by taking for $\delta$ the minimum of the $\delta_{\mathbf{v}}$'s on all open faces of the $FC_{\leq\eta}$'s (whose number is smaller than the number of possible signs, i.e., $3^r$). The proof

is still valid for $FP$ but the result is meaningful only if $N$ is included in $FP$, i.e., if $\mathbf{0}$ is an interior point of $FP$, which means that the inhomogeneous linear constraints defining $FP$, given by $\mathbf{Gv} \geq \mathbf{h}$, verify $\mathbf{h} < \mathbf{0}$. $\quad\square$

Theorem 3 tells us that the result of Theorem 1 for sign-invariant constraints regarding the geometrical structure of the solution space applies identically for contracting-sign-invariant constraints locally in a neighborhood of $\mathbf{0}$, i.e., when considering only pathways with sufficiently small amounts of fluxes. This applies in particular to those kinetic constraints described in Lemma 5.

*3.2. Sign-Monotone Constraints*

We now consider a property of compatibility of a constraint with signs that is stronger than sign-invariance. A constraint $\mathbf{C_x}(\mathbf{v})$ (resp., $\exists\mathbf{x}\mathbf{C_x}(\mathbf{v})$), is said to be sign-monotone if, when satisfied by a vector, it is satisfied by any other vector with a smaller or equal sign (for the partial order on signs):

$$\mathbf{C_x}(\mathbf{v}), sign(\mathbf{v}') \leq sign(\mathbf{v}) \Rightarrow \mathbf{C_x}(\mathbf{v}') \tag{92}$$

$$\exists\mathbf{x}\mathbf{C_x}(\mathbf{v}), sign(\mathbf{v}') \leq sign(\mathbf{v}) \Rightarrow \exists\mathbf{x}\mathbf{C_x}(\mathbf{v}'). \tag{93}$$

It follows from these definitions that, if the $\mathbf{C_x}(\mathbf{v})$'s are sign-monotone for all $\mathbf{x}$, then $\exists\mathbf{x}\mathbf{C_x}(\mathbf{v})$ is sign-monotone and that any sign-monotone constraint is sign-invariant. Note that, in any given closed orthant, thus in any flux tope, $sign(\mathbf{v}') \leq sign(\mathbf{v})$ is equivalent to $supp(\mathbf{v}') \subseteq supp(\mathbf{v})$ and thus being sign-monotone for a constraint means being support-monotone in each flux tope. Note also that if a sign-monotone constraint has a solution, then the null vector $\mathbf{0}$ is a solution.

**Example 10.** *It then follows from definitions (41), (42) or (43) that the thermodynamic constraint $TC_M(v)$ or $\underline{TC}_M(v)$ is sign-monotone. This is thus also the case for $TC(v)$ and $\underline{TC}(v)$ Equation (60) and for $TC^b(v)$ and $\underline{TC}^b(v)$ Equation (61).*

*The argument given previously to establish that the kinetic constraint $KC_M(v)$ Equation (80) is sign-invariant proves in fact that it is sign-monotone. This sign-monotone property holds thus also for $KC(v)$ Equation (76) and for $KC^b(v)$ Equation (77) in the absence of bounds on enzyme concentrations.*

*The regulatory constraint $RC_{Bc}(v)$ Equation (50) is not generally sign-monotone: consider for example Bc reduced to a positive literal. Actually, $RC_{Bc}(v)$ is sign-monotone if and only if Bc in DNF contains no positive literal, which is a very special case, requiring only certain fluxes to be zero but unable to express that a given flux is nonzero (in this particular case, it is support-monotone, which is stronger than sign-monotone).*

**Lemma 6.** *All thermodynamic constraints are sign-monotone. Only those kinetic constraints $KC_M$ and $KC$ are sign-monotone, as well as $KC^b$ in the absence of bounds on enzyme concentrations. The regulatory constraint $RC_{Bc}$ is not sign-monotone, except if Bc in DNF contains only negative literals (i.e., assigns only zero fluxes, but no nonzero fluxes), in which case it is support-monotone.*

The structure of the solution space of a sign-monotone constraint follows directly from its definition.

**Lemma 7.** *The set $Sol_C$ of vectors in $\mathbb{R}^r$ satisfying the sign-monotone constraint $C = C_x(v)$ (resp., $C = \exists x C_x(v)$) is a union of closed orthants.*

Actually, if $\mathbf{v}$ satisfies $\mathbf{C}$, then $O_{sign(\mathbf{v})} = \{\mathbf{x} \in \mathbb{R}^r \mid sign(\mathbf{x}) \leq sign(\mathbf{v})\}$ is included in $Sol_\mathbf{C}$ and thus $Sol_\mathbf{C} = \bigcup_{\{\mathbf{v}|\mathbf{C}(\mathbf{v})\}} O_{sign(\mathbf{v})}$. Obviously, we can keep only those $\mathbf{v}$'s for which $sign(\mathbf{v})$ is maximal, in order to avoid any inclusion between the closed orthants. This result can be compared to Lemmas 1, 3 and first part of Proposition 5, which are obviously more precise but it shows that the mathematical structure of the solution space

of these thermodynamic and kinetic constraints is mainly the only consequence of their sign-monotonicity.

**Theorem 4.** *Consider indifferently a sign-monotone constraint $C_x(v)$ or $\exists x C_x(v)$ (e.g., if $C_x(v)$ is sign-monotone for any $x$), and let us name it $\mathbf{C}$ and the associated constrained flux cone subset $CFC_{\mathbf{C}}$ (resp., the associated constrained flux polyhedron subset $CFP_{\mathbf{C}}$). Then, $CFC_{\mathbf{C}}$ (resp., $CFP_{\mathbf{C}}$) is a finite union of polyhedral cones (resp., polyhedra), which are faces of the flux topes $FC_{\leq \boldsymbol{\eta}}$ (resp., $FP_{\leq \boldsymbol{\eta}}$) for all $\boldsymbol{\eta}$ maximal sign vectors in $sign(FC)$ (resp., $sign(FP)$) and we obtain*

$$EFVs(CFC_{\mathbf{C}}) = EFMs(CFC_{\mathbf{C}}) = EFVs(FC) \cap Sol_{\mathbf{C}} = EFMs(FC) \cap Sol_{\mathbf{C}} \qquad (94)$$

$$EFPs(CFP_{\mathbf{C}}) = EFPs(FP) \cap Sol_{\mathbf{C}} \qquad EFVs(CFP_{\mathbf{C}}) = EFVs(C_{FP}) \cap Sol_{\mathbf{C}}. \qquad (95)$$

**Proof of Theorem 4.** The (non-disjoint) decomposition of the solution space into faces of the flux topes $FC_{\leq \boldsymbol{\eta}}$ (resp., $FP_{\leq \boldsymbol{\eta}}$) follows from Lemma 7 or directly from the fact that all vectors of the conical hull $cone(\mathbf{v}^1, \ldots, \mathbf{v}^n) = \{\beta_1 \mathbf{v}^1 + \ldots + \beta_n \mathbf{v}^n \mid \beta_1, \ldots, \beta_n \geq 0\}$ and of the convex hull $conv(\mathbf{v}^1, \ldots, \mathbf{v}^n) = \{\alpha_1 \mathbf{v}^1 + \ldots + \alpha_n \mathbf{v}^n \mid \alpha_1, \ldots, \alpha_n \geq 0, \alpha_1 + \ldots + \alpha_n = 1\}$ of given vectors $\mathbf{v}^1, \ldots, \mathbf{v}^n$ in a flux tope, thus in a closed orthant (so the sum is conformal), have their supports included in $supp(\mathbf{v}^1) \cup \ldots \cup supp(\mathbf{v}^n)$, which is the support of any vector $\mathbf{v}$ in $cone^+(\mathbf{v}^1, \ldots, \mathbf{v}^n)$ or $conv^+(\mathbf{v}^1, \ldots, \mathbf{v}^n)$, thus their signs being less than or equal to the sign of $\mathbf{v}$, and from the fact that a face of a polyhedron is the Minkowski sum of the convex hull of its vertices and the conical hull of its extreme vectors. Thus, if a vector $\mathbf{v}$ of a flux tope $FC_{\leq \boldsymbol{\eta}}$ (resp., $FP_{\leq \boldsymbol{\eta}}$) belongs to the solution space, the minimal face of $FC_{\leq \boldsymbol{\eta}}$ (resp., $FP_{\leq \boldsymbol{\eta}}$) containing $\mathbf{v}$ is completely included in it. As a sign-monotone constraint is sign-invariant, the results of Theorem 1 apply and provide formulas for EFVs and EFPs. Thus remains only the case of EFMs. Now, if an elementary flux mode $\mathbf{v}$ of the solution space were not support-minimal in $FC$, a nonzero vector $\mathbf{v}'$ would exist in $FC$ with $supp(\mathbf{v}') \subset supp(\mathbf{v})$. Moreover, we could choose $\lambda \in \mathbb{R}$ such that $\mathbf{v}'' = \mathbf{v} - \lambda \mathbf{v}'$ is nonzero, belongs to $FC$ and verifies $supp(\mathbf{v}'') \subset supp(\mathbf{v})$ and $sign(\mathbf{v}'') \leq sign(\mathbf{v})$. From the sign-monotonicity property of the constraint, $\mathbf{v}''$ would also satisfy the constraint and would thus belong to the solution space, which would contradict the fact that $\mathbf{v}$ is an elementary flux mode in this space. Therefore, $\mathbf{v}$ is an elementary flux mode in $FC$ and we get the result for EFMs. $\square$

Of course, in the decomposition of $CFC_{\mathbf{C}}(\mathbf{x})$ or $CFC_{\mathbf{C}}$ (resp., $CFP_{\mathbf{C}}(\mathbf{x})$ or $CFP_{\mathbf{C}}$) as a union of certain faces of the $FC_{\leq \boldsymbol{\eta}}$'s (resp., $FP_{\leq \boldsymbol{\eta}}$'s), we can only keep those faces which are maximal. Theorem 4 applies in particular to all thermodynamic constraints, to those kinetic constraints and to those very few regulatory constraints described in Lemma 6. In particular, we directly obtain (except of course the reference to $\mathbf{ts_M}$) Proposition 2 for thermodynamic constraints $\mathbf{TC}$ and $\mathbf{TC}^b$, as well as the structure of the solution space as a union of flux cones (resp., flux polyhedra) for kinetic constraints $\mathbf{KC}$ and $\mathbf{KC}^b$ (in the absence of bounds on enzyme concentrations) and the characterization of elementary fluxes and elementary flux modes given by Proposition 5, which proves that these results depend only on the fact that these constraints are sign-monotone.

Note that the knowledge of EFVs, equal to EFMs, of $CFC_{\mathbf{C}}$ (resp., and of EFPs of $CFP_{\mathbf{C}}$) is not good enough to reconstruct the structure of the solution space $CFC_{\mathbf{C}}$ (resp., $CFP_{\mathbf{C}}$) as a union of polyhedral cones (resp., polyhedra). For this, it is necessary to know the (non-disjoint) decomposition $\{E_i\}$ of these EFVs (resp., and EFPs) as extreme vectors (resp., and extreme points) of the faces $F_i$ of the flux topes $FC_{\leq \boldsymbol{\eta}}$ (resp., $FP_{\leq \boldsymbol{\eta}}$) that constitute the solution space. The $E_i$'s are exactly the maximal subsets of EFVs (resp., and EFPs) whose conformal conical hull (resp., conformal convex hull) is entirely contained in the solution space: $cone_{\oplus}(E_i) \subseteq CFC_{\mathbf{C}}$ (resp., and $conv_{\oplus}(E_i) \subseteq CFP_{\mathbf{C}}$). By analogy with LTCS, we will call each such $E_i$ a largest $\mathbf{C}$-consistent set of EFVs or EFMs (resp., and of EFPs), noted $LCS_{\mathbf{C}}$.

In order to deal with enzymatic capacity constraints, we generalize the sign-monotonicity criterion exactly in the same way we generalized the sign-invariance criterion. A constraint, $\mathbf{C_x(v)}$ (resp., $\exists \mathbf{xC_x(v)}$), is said to be contracting-sign-monotone if, when satisfied by one vector, it is satisfied by any vector with a smaller or equal sign which is not greater (on each component) than the vector itself, i.e., by any vector that belongs to the rectangle parallelepiped defined by the null vector and the given vector:

$$\mathbf{C_x(v)}, sign(\mathbf{v'}) \leq sign(\mathbf{v}), \forall i |\mathbf{v}'_i| \leq |\mathbf{v}_i| \Rightarrow \mathbf{C_x(v')} \tag{96}$$

$$\exists \mathbf{xC_x(v)}, sign(\mathbf{v'}) \leq sign(\mathbf{v}), \forall i |\mathbf{v}'_i| \leq |\mathbf{v}_i| \Rightarrow \exists \mathbf{xC_x(v')}. \tag{97}$$

Obviously a sign-monotone constraint is contracting-sign-monotone and, if the $\mathbf{C_x(v)}$'s are contracting-sign-monotone for all $\mathbf{x}$, then $\exists \mathbf{xC_x(v)}$ is contracting-sign-monotone. Furthermore, any contracting-sign-monotone constraint is contracting-sign-invariant.

**Example 11.** *The argument given in Example 9 to establish that the kinetic constraints $KC_M^b(v)$ Equation (81) and $KC^b(v)$ Equation (77) are contracting-sign-invariant in the absence of positive lower bounds on enzyme concentrations (i.e., when $E^- = 0$), proves in fact that they are contracting-sign-monotone.*

**Lemma 8.** *The kinetic constraints $KC_M^b$ and $KC^b$ are contracting-sign-monotone in the absence of positive lower bounds on enzyme concentrations (i.e., when $E^- = 0$).*

**Theorem 5.** *If $C = C_x(v)$ (or $C = \exists xC_x(v)$) is contracting-sign-monotone, then $CFC_C$ is a $\mathbf{0}$-star domain and there exists a neighborhood $N = ]-\delta, +\delta[^r$ of $\mathbf{0}$ for a certain $\delta > 0$ such that $CFC_C \cap N$ is a finite union of N-truncated polyhedral cones, which are the intersection with N of faces of the flux topes $FC_{\leq \eta}$. In addition, results (94) regarding EFVs and EFMs hold in N, i.e., for sufficiently small fluxes. The same holds for $CFP_C$ with flux topes $FP_{\leq \eta}$ if $\mathbf{0}$ is an interior point of FP, and in this case results (95) regarding EFVs and EFPs hold in N.*

**Proof of Theorem 5.** The proof becomes identical to that of Theorem 3 by using the proof of Theorem 4. $\square$

Theorem 5 tells us that the result of Theorem 4 for sign-monotone constraints regarding the geometrical structure of the solution space and the determination of elementary fluxes and elementary flux modes applies identically for contracting-sign-monotone constraints locally in a neighborhood of $\mathbf{0}$, i.e., when considering only pathways with sufficiently small amounts of fluxes. It applies in particular to those kinetic constraints described in Lemma 8, providing the result for the structure of $CFC_{\mathbf{KC}_M^b}$ and $CFC_{\mathbf{KC}^b}$ quoted in the last part of Proposition 5.

Finally, Proposition 11 extends straightforwardly to the case where we have to deal with both one sign-monotone constraint and one sign-invariant constraint. As a sign-monotone constraint is sign-invariant the conjunction of the two constraints is sign-invariant and Theorem 1 applies.

**Theorem 6.** *The space of flux vectors in FC (resp., FP) satisfying both a sign-monotone constraint and a sign-invariant constraint is a finite disjoint union of open polyhedral cones (resp., open polyhedra) which are certain open faces of the flux topes of FC (resp., FP). The elementary flux vectors (resp., elementary flux points and vectors) are those of FC (resp., FP) that satisfy both constraints.*

Note that, in the case of *FC*, Theorem 2 applies also to determine elementary flux modes and support-wise non-strictly-decomposable vectors.

### 3.3. Consequences on the Computation of Elementary Fluxes

From the results above, we will see that the computation of elementary fluxes in the presence of a sign-monotone constraint can be efficiently performed with the Double Description (DD) method.

First let us briefly remember the principle of the DD method [11]. This is an algorithm that takes as input the implicit description of a pointed convex polyhedral cone $C$ as its representation matrix, i.e., a finite set of homogeneous linear inequalities defining $C$ as the intersection of the corresponding vector half-spaces, and produces as output the explicit description of $C$ as a (minimal) generating matrix, i.e., the set of extreme rays of $C$. More generally, it can deal in the same way with a pointed convex polyhedron $P$ producing, from a finite set of linear inequalities defining $P$ as the intersection of the corresponding affine half-spaces, the explicit description of $P$ as two generating matrices providing respectively the vertices of $P$ and the extreme rays of $C_P$. As the first are obtained as the extreme rays of the pointed cone obtained from $P$ by adding one dimension to the space and considering the conical hull of $P$ from an origin of this extended vector space, it is enough to explain how the DD method works on pointed convex polyhedral cones. The DD method is an incremental algorithm that processes one by one each of the $n$ homogeneous linear inequalities defining $C$. At each step $i, 1 \leq i \leq n$, it builds the intermediate extreme rays of the intermediate current cone $C_i$ defined by the $i$ first linear inequalities from the knowledge of the extreme rays of $C_{i-1}$ built at previous steps and of the $i$th linear inequality. At the end, for $i = n$, $C_n = C$ and the extreme rays are thus obtained. The $i$th linear inequality defines a half-space $H_i^+$ with a vector hyperplane $H_i$ as a frontier. All the extreme rays of $C_{i-1}$ that are on the right side of $H_i$, i.e., belong to $H_i^+$, are extreme rays of $C_i$ and thus kept; all those that are on the wrong side do not belong to $C_i$ and will be discarded; the new extreme rays at step $i$ appear from the intersection of $C_{i-1}$ with $H_i$ and are obtained as the intersection with $H_i$ of the 2-faces of $C_{i-1}$ defined by two adjacent extreme rays, one on each side of $H_i$. We therefore just need to keep and update at each step this list of pairs of adjacent extreme rays and, when the $i$th linear inequality is processed, each such pair with elements on both sides of $H_i$ determines with $H_i$ an extreme ray for $C_i$. The key fact is that each new extreme ray built at step $i$ is the conical sum of two extreme rays existing at step $i - 1$ and, as we will proceed by flux tope, i.e., in a given closed $r$-orthant, this sum is conformal and the support of this new extreme ray is the union of the supports of two previously existing extreme rays. Moreover, similarly, this new extreme ray, if involved in the next steps as a member of a relevant pair of adjacent rays, in order to build a new extreme ray, will have its support included in the support of the latter one. Finally, all we need to do is initialize $C_1$. For a flux cone $FC$ defined by Equation (6), what has been shown as the most efficient initialization is to start from a basis of the nullspace of $\mathbf{S}$ (the authors of [43] proposed a method to compute EFMs directly as linear combinations of the vectors of a basis of this nullspace but it turned out to be less efficient than DD), and this basis (of size $r - m$ by assuming $\mathbf{S}$ of full rank) can be chosen so that $r - m$ of the $r$ linear inequalities are satisfied (we consider here the worst case corresponding to the highest number of such inequalities, i.e., $r_I = r$, which happens in particular if each reversible reaction is split into two irreversible ones). There thus remain $m$ linear inequalities to satisfy, i.e., there are $n = m$ steps in the DD algorithm.

Let us now consider a sign-monotone constraint $\mathbf{C}$. From Theorem 4, the EFVs (identical to EFMs) of the constrained flux cone subset $CFC_{\mathbf{C}}$ are simply those of $FC$, i.e., of all its flux topes $FC_{\leq\boldsymbol{\eta}}$ successively, that satisfy constraint $\mathbf{C}$ (and the same is true for EFPs of $CFP_{\mathbf{C}}$). Therefore, we just have to build the extreme rays of each $FC_{\leq\boldsymbol{\eta}}$ by using the DD algorithm as presented above and filter those that satisfy $\mathbf{C}$. Obviously a filtering at the end, once all extreme rays built, will not at all improve the efficiency. However, by exploiting the key fact that any newly built extreme ray at a certain step, if used in other next steps to build other new extreme rays will have necessarily its support included in each support of the latter ones, and exploiting the sign-monotonicity of $\mathbf{C}$, we can conclude that, each time a newly built extreme ray does not satisfy the constraint $\mathbf{C}$, it can be immediately

discarded because not only it has no use at the step of its discovery but also no use at the future steps as all extreme rays, in the building of which it could be involved, would have a larger support and thus would thus not satisfy **C** either. Thus, the computation of elementary fluxes or elementary flux modes in the presence of a sign-monotone constraint can be fully integrated into the incremental DD algorithm, the filtering of the extreme rays satisfying the constraint being achieved at each step when they are newly created. The extra-cost is having to check all intermediate extreme rays built for satisfiability by **C** (for most constraints, this can be practically instantaneous) and the gain is that all intermediate extreme rays for which this checking gives a negative answer are discarded.

**Proposition 13.** *Given a sign-monotone constraint **C**, the EFVs (or EFMs) of the constrained flux cone subset $CFC_C$ are obtained by the DD algorithm as the extreme rays of each flux tope $FC_{\leq \eta}$ by filtering out at each step all those newly built extreme rays that do not satisfy **C**.*

This result applies in particular to thermodynamic constraints **TC** and **TC**$^b$ [44,45] and to kinetic constraints **KC** and **KC**$^b$ (in the absence of bounds on enzyme concentrations). Actually, for **TC** (or **KC** which is identical from Proposition 5), Proposition 3 demonstrated that the criterion for an extreme ray to be thermodynamically satisfiable is to belong to a given fixed open half-space delimited by a vector hyperplane. Checking this satisfiability thus boils down to computing the scalar product of the extreme ray with a fixed vector (normal to the hyperplane and oriented towards the half-space) and verifying it is positive. In this case, as it has been shown, it is much simpler to integrate this thermodynamic constraint as one supplementary $(n + 1)$th homogeneous linear inequality and we can choose for example that it is processed first by the DD algorithm. However, as it has been already pointed out, providing the list of the extreme flux vectors of $CFC_C$ does not provide its structure in terms of a union of polyhedral cones. For this, we have to identify all the $LCS_C$'s, i.e., the maximal subsets of those extreme flux vectors whose conformal conical hull is entirely contained in $CFC_C$. For each flux tope $FC_{\leq \eta}$, this amounts to computing all maximal upper bounds for sets of signs of those extreme rays built from this flux tope such that any vector of this tope with a sign equal to such an upper bound satisfies constraint **C**, as the vectors of the strict conical hull of a subset of extreme rays have as a common sign the upper bound for the signs of these extreme rays. To each such upper bound identified is associated the $LCS_C$ made up of all the extreme rays whose signs are smaller than, or equal to, this upper bound.

For sign-invariant constraints, and thus for regulatory constraints, no such incremental filtering is possible as a newly built intermediate extreme ray may have a sign that does not satisfy the constraint but may participate at some later stage in the construction of other new extreme rays (with necessarily larger supports) whose sign will satisfy the constraint. Thus, such an extreme ray cannot be discarded. In addition, we saw in any case there is no longer identity between those EFMs of $FC$ that satisfy the constraint, the EFMs of $CFC_C$ and the support-wise non-strictly-decomposable vectors in $CFC_C$ and, as only the last two are biologically relevant, computing only the first ones would be of limited interest.

## 4. Conclusions

The analysis of metabolic networks in a steady state takes classically into account only stoichiometric and reactions irreversibility constraints. In this context, well-known pathways have been introduced, such as elementary flux modes EFMs, expressing the property of minimality, or elementary flux vectors EFVs, expressing the property of non-decomposability (which coincide in this standard framework), that are biologically relevant in their own and from which all pathways, whose structure is a convex polyhedral flux cone $FC$, can be reconstructed. However, first, the computation of these EFMs or EFVs is hampered by the combinatorial explosion of their number in genome-scale metabolic models, and, second, most of them are biologically irrelevant, because other important biological constraints are not taken into account. With the objective of both enumerating

only biologically feasible EFMs or EFVs (without filtering them afterwards) and, as there are considerably fewer of them, improving the scalability of this computation, we took into consideration in this paper on one side thermodynamic and, more generally, kinetic constraints and on the other side regulatory constraints, and we tackled the problem of revisiting in this new extended framework the concept of EFMs and EFVs and, more largely, of characterizing the geometry of the solution space. Actually, we considered a more general conceptual framework for constraints, namely their property to be compatible with vector signs (i.e., with vector supports separately in each closed *r*-orthant), because most properties of the geometrical structure of the solution space happen to depend only on this very general sign-compatibility of constraints. This is how we demonstrated, for constraints which are sign-monotone (i.e., once satisfied by a certain vector are satisfied by any vector with smaller or equal sign), which is the case of thermodynamic constraints and of kinetic constraints in the absence of bounds on enzyme concentrations, that the solution space is a union of convex polyhedral cones, which are certain faces of the flux topes of *FC*, and that the EFMs, which still coincide with the EFVs, are simply those of *FC* that satisfy the constraint. In addition, we showed that their computation can be efficiently integrated into the classical double description algorithm, as each newly built intermediate extreme ray that does not satisfy the constraint can be filtered out at each incremental step. For the specific case of thermodynamic constraints or of kinetic constraints in the absence of bounds on enzyme concentrations, we demonstrated that their solution spaces are identical and, when there are also no bounds on metabolite concentrations, made up of those maximal faces of the flux topes of *FC* which are entirely contained in a fixed open vector half-space and thus that the EFMs are simply those of *FC* belonging to this half-space and thus computable by adding one single homogeneous linear inequality to the initial (stoichiometric and reactions irreversibility) ones. The situation is more complex, and challenging from a computational perspective, for constraints which are only sign-invariant (i.e., once satisfied by a certain vector are satisfied by any vector with the same sign), which is the case of regulatory constraints. We showed that in fact sign-invariant constraints are constraints that are support-invariant in each closed *r*-orthant separately and that support-invariant constraints coincide with regulatory (Boolean) constraints, therefore regulatory constraints are prototypical of sign-invariant constraints. For such sign-invariant constraints, we demonstrated that the solution space is a finite disjoint union of semi-open convex polyhedral cones, obtained from certain faces of the flux topes of *FC* by removing certain of their own faces of lesser dimension, and that the EFVs are simply those of *FC* (equal to the EFMs of *FC*) that satisfy the constraint. However, these cannot be efficiently computed by the double description algorithm because they cannot be filtered out during the incremental process. In addition it is no longer true that EFVs identify with EFMs, as it exists in general minimal solutions that are decomposable, and that EFMs identify with support-wise non-strictly-decomposable vectors (a new property that we introduced and which is the proper one to express non-decomposability in this general framework, in terms of support non-decomposability), as there are in general such support-wise non-strictly-decomposable solutions that are not minimal: there are usually strict inclusions between these three sets and we provided again a complete characterization of the two latter ones. Finally, we extended all these results to the case where inhomogeneous linear constraints (expressing for example capacity constraints or bounds on fluxes) exist, dealing thus with a convex flux polyhedron *FP* instead of a flux cone *FC*. Basically, most of the results regarding the geometrical structure of the solution space in the presence of the above biological constraints remain the same with cones replaced by polyhedra.

Future work will be carried out along two paths. First, the present theoretical work will be extended to minimal cut sets (MCSs). Such MCSs are defined as minimal (for set inclusion) sets of reactions whose deletion will block the operation of given objectives or target reactions (as, e.g., those producing some toxic or undesirable product), i.e., removal of an MCS (that is, the knockout of its reactions) implies a zero flux for the target reactions in a steady state. MCSs are important for computing intervention strategies, e.g., for

metabolic engineering. It is obvious from this definition that, for a metabolic network modeled in a steady state by a flux cone *FC*, the MCSs are the minimal hitting sets of the set of target EFMs (identical to EFVs of the given metabolic network (i.e., the set of EFMs that comprise at least one of the target reactions), where a hitting set of the target EFMs is a set (of reactions) that has a nonempty intersection with each one of these EFMs. Moreover, this generalizes to a metabolic network modeled by a flux polyhedron *FP* by using EFPs. This gives an indirect method for computing MCSs [46,47] from the preliminary computation of EFVs (or EFPs) that can be applied in the presence of biological constraints by using the results obtained in this paper. However, it is known that there is also a method for computing MCSs directly as the EFVs of a dual network [48,49], obtained basically by transposing the stoichiometric matrix of the original matrix (thus in some sense, exchanging reactions and metabolites). We will therefore study how to define this dual operation properly for metabolic networks in the presence of biological constraints in order to preserve this result (or most of it). Second, the algorithms described in this paper will be implemented and testing conducted on metabolic networks described in the literature for which biological (thermodynamic, kinetic or regulatory) data are known. As it has been shown, the computation of elementary flux vectors (or elementary flux points) boils down to filtering those of extreme rays (or vertices) in each flux tope which satisfy the biological constraints. Moreover, for sign-monotone constraints as thermodynamic constraints or kinetic constraints in the absence of bounds on enzyme concentrations, this filtering can be easily integrated at each step of the incremental double description method, so it is natural to rely first on this method which benefits from very efficient implementations. In the absence of bounds on metabolite concentrations, the advantage is obvious as it is just necessary to add one step (i.e., one homogeneous linear constraint to deal with) in the DD algorithm. It has nevertheless been shown that at most 50% of the EFVs can be ruled out that way, which is clearly insufficient to scale up to GSMMs. Adding bounds on metabolite concentrations has already proved capable of ruling out a higher percentage of EFVs. This is achieved by checking the extreme rays at each step of the DD algorithm thanks to a call to a linear programming solver and it will be interesting to compare its efficiency with the method given by Proposition 4 which does not need using an LP solver but has the disadvantage of requiring reasoning in a higher dimension. As the structure of the solution space cannot be deduced from merely the knowledge of the EFVs but requires the identification of the largest consistent (w.r.t. the constraints) sets of EFVs, efficient computation of these $LCS_C$'s will be studied. We can reasonably think that scaling up will be obtained only by dealing with all biological constraints together. However, as it has been shown, handling regulatory constraints poses serious problems as these constraints are not sign-monotone and thus filtering of the EFVs that satisfy them cannot be integrated incrementally into the DD algorithm. In addition the structure of the solution space as union of semi-open polyhedral cones (or semi-open polyhedra) is more complex and pathways of interest for biologists, such as elementary flux modes or support-wise non-strictly-decomposable vectors, no longer coincide with EFVs and require more complex computations to be identified. Algorithms will be carefully studied for maximal efficiency and novel ways of using the DD method or the use of other methods recently proposed such as local reverse search or satisfiability based methods [42,50] will be investigated

**Funding:** This research received no external funding.

**Data Availability Statement:** Data sharing not applicable.

**Acknowledgments:** I would like to express my sincere thanks to my colleagues Sabine Peres, who introduced me a few years ago to bioinformatics and more particularly to metabolic pathway analysis, that was a field completely unknown to me, and Antoine Deza, who explained to me all I had to know about convex geometry and the double description algorithm. Without their help and the huge number of hours we spent working and discussing together, this work could never have been accomplished. I also warmly thank Rosemary Patricot for having proofread the English.

**Conflicts of Interest:** The authors declare no conflict of interest.

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
