# Peer review of "Metabolic Pathway Analysis in the Presence of Biological Constraints"

_computation, doi:10.3390/computation9100111_

Round 1
Reviewer 1 Report
1. Main comments:
The past years have seen a lot of advances with dealing with the understanding of metabolic pathways from both the computational and theoretical aspects. Being able survey and analyze the interactions between the computational framework and geometric and combinatorial structure of the input is certainly a useful and timely contribution.
The exposition is relatively clear and self-contained, the bibliography is thoughtful, and the notation helps to present a coherent overview of different approaches. Thus, the reviewer recommends accepting this manuscript.
2. Other comments:
Another round of style / typo check to polish the submission would be useful. For example, but they are other occurrences, in lines 347-349 some words and/or punctuations are probably missing, e.g “As was the case” or " In the same way we replaced …. , we will"
Maybe reducing the numbers of "(….)" or “….” could help to clarify the exposition as, for instance, in lines 309-110
Propositions, for instance Proposition 2.8, might be easier to parse if what is assumed and what is derived was clearly delineated. The current presentation might expect too much from the reader, especially from someone not knowledgeable in some aspects of the computational and theoretical aspects of metabolic pathways.
line 27: “historical” might be unnecessary
Title of Section 1.7 (and elsewhere): every field has its own conventions so this comment might be wrong, but the reviewer is wondering whether non-homogenous would be more appropriate than inhomogeneous?
Lines 304-306: everything is correct there. Just one remark that is just a question of taste
"cannot be written as a convex combination of distinct vectors of P:”
could be replaced by
"cannot be written as the midpoint of two distinct points of P:”
(and then replace \lambda by 1/2)
Lines 455-456: maybe add a reference or clarity the first statement. It is rather odd to have references for a direct corollary and not for the original general case. In addition, the abstract of [2] states "The complexity of enumerating all elementary modes remains open”. Note that NP-completeness is stated as a decision problem. This part should be either clarified or removed.
The author identifies the vector v and the point v. It is fine but a bit confusing when both are used in the same sentence.
The contribution of the part of Section 1.8 dealing with McMullen upper bound is a bit unclear as this discussion is about the upper bound while the actual number of extreme rays is likely much smaller that these bounds.
Section 2 is certainly a key one, for instance Proposition 2.9, and adding some illustrations and more examples could be beneficial to the impact of the paper. Examples 2.1, 2,2, etc could be illustrated by some drawings.
Line 761: Farkas’ work precedes Gale’s work by over 50 years so maybe Farkas’ Lemma is a more appropriate terminology
Section 3 would also benefit from illustrations and examples.In particular would it be possible to illustrate Theorem 3.3?
The double description method is known to be highly sensitive to the ordering of the input. Did the author tried different ordering to gain some insights?
Section 4 would benefit from a clearer hierarchy of the results. Overall, given the length of the paper, a clearer segmentation / hierarchy could strengthen the paper. The main statements should be better highlighted maybe by having a first section summing up the key findings.
Author Response
Thanks very much for your comments and global evaluation.
I join an updated pdf file with revisions in blue.
>> Another round of style / typo check to polish the submission would be useful.
I do not know what version of my paper you read but a few days after my initial submission I uploaded on the site of the editor a new version, proofread by an English friend.
>> For example, in lines 347-349 some words and/or punctuations are probably missing, e.g “As was the case” or " In the same way we replaced …. , we will"
I rephrased the first sentence and modified punctuations in the second one (see lines 356-361).
>> Maybe reducing the numbers of "(….)" or “….” could help to clarify the exposition as, for instance, in lines 309-110.
I agree, I suppressed the parentheses (see lines 317-318).
>> Propositions, for instance Proposition 2.8, might be easier to parse if what is assumed and what is derived was clearly delineated. The current presentation might expect too much from the reader, especially from someone not knowledgeable in some aspects of the computational and theoretical aspects of metabolic pathways.
Ok, I will try to make propositions more readable.
>> line 27: “historical” might be unnecessary
Ok, I suppressed it (see line 35).
>> Title of Section 1.7 (and elsewhere): every field has its own conventions so this comment might be wrong, but the reviewer is wondering whether non-homogenous would be more appropriate than inhomogeneous?
Both terms are used in mathematics and bioinformatics. I kept inhomogeneous, which is largely used in reference [24] from Klamt et al., one of the references I used for the state of the art (actually my title appears almost identically in this paper).
>> Lines 304-306: everything is correct there. Just one remark that is just a question of taste
>> "cannot be written as a convex combination of distinct vectors of P:”
>> could be replaced by
>> "cannot be written as the midpoint of two distinct points of P:”
>> (and then replace \lambda by 1/2)
I added it (see line 314) but did not replaced as it depends on the space considered (e.g., each non-rational number is a convex (non-rational) combination of two distinct rational numbers, but is never the midpoint of two such numbers).
>> Lines 455-456: maybe add a reference or clarity the first statement. It is rather odd to have references for a direct corollary and not for the original general case. In addition, the abstract of [2] states "The complexity of enumerating all elementary modes remains open”. Note that NP-completeness is stated as a decision problem. This part should be either clarified or removed.
Thanks for having noticed the error in my first line, where enumerating had to be replaced by counting. I clarified the results about enumeration and added reference [23] from Khachiyan et al. (see lines 470-488).
>> The author identifies the vector v and the point v. It is fine but a bit confusing when both are used in the same sentence.
I added a footnote regarding this identification in page 7 (line 294).
>> The contribution of the part of Section 1.8 dealing with McMullen upper bound is a bit unclear as this discussion is about the upper bound while the actual number of extreme rays is likely much smaller that these bounds.
I separated clearly enumeration from counting and, concerning counting, added reference [8] from Dyer (see lines 489-491). I also added a sentence to point out that the actual number of solutions is in practice much smaller than the upper bound (see lines 604-605).
>> Section 2 is certainly a key one, for instance Proposition 2.9, and adding some illustrations and more examples could be beneficial to the impact of the paper. Examples 2.1, 2,2, etc could be illustrated by some drawings.
I completely agree with you. I planned to add drawings if the paper was accepted, but I am not familiar with this and asked help to the editor or at least more time than the 5 days given for the revision.
>> Line 761: Farkas’ work precedes Gale’s work by over 50 years so maybe Farkas’ Lemma is a more appropriate terminology
I added it (see line 789).
>> Section 3 would also benefit from illustrations and examples. In particular would it be possible to illustrate Theorem 3.3?
Yes, I will try to provide hand-made drawings if the editor may produce figures from them.
>> The double description method is known to be highly sensitive to the ordering of the input. Did the author tried different ordering to gain some insights?
Yes, it is one of the defaults of this method. It is a future work to investigate if a general efficient ordering strategy exists that takes into account the peculiarities of metabolic networks (as long as only thermodynamical or kinetic constraints are handled). However, anyway, filtering extreme rays satisfying regulatory constraints cannot be integrated incrementally in the DD method and some new methods have to be developed.
>> Section 4 would benefit from a clearer hierarchy of the results. Overall, given the length of the paper, a clearer segmentation / hierarchy could strengthen the paper. The main statements should be better highlighted maybe by having a first section summing up the key findings.
I added some sentences (see lines 1909-1910, 1915-1916, 1936-1941, 1945-1946, 1949-1953) to emphasize the results. I can try to segment these contributions with bullets.
I also clarified the contributions in the abstract by adding several sentences (see new abstract).

Reviewer 2 Report
The submitted manuscript describes and discusses the results of an original research project carried out to metabolic pathway analysis as a key method to study metabolism. The manuscript describes and discusses logically designed experiments and presents results that are expected to be of large interest for the scientific community. The Author provided results about the mathematical structure and geometrical characterization of the solutions space in presence of such biological constraints and revisit the concept of elementary fluxes and elementary flux modes in this framework. It is an interesting study with a novel approach.
Author Response
Thank you for appreciation.
Best regards,
Philippe